# Consistent Sampling and Simulation: Molecular Dynamics with Energy-Based Diffusion Models

**Michael Plainer**[† 1,2,3,4]

**Hao Wu**[† 5]

**Leon Klein** [1]         **Stephan Günnemann** [6]         **Frank Noé**[† 1,7,8]

[1]Freie Universität Berlin    [2]Zuse School ELIZA    [3]Technische Universität Berlin
[4]Berlin Institute for the Foundations of Learning and Data [5]Shanghai Jiao Tong University
[6]Technische Universität München    [7]Rice University    [8]Microsoft Research AI4Science

## Abstract

In recent years, diffusion models trained on equilibrium molecular distributions have proven effective for sampling biomolecules. Beyond direct sampling, the score of such a model can also be used to derive the forces that act on molecular systems. However, while classical diffusion sampling usually recovers the training distribution, the corresponding energy-based interpretation of the learned score is often inconsistent with this distribution, even for low-dimensional toy systems. We trace this inconsistency to inaccuracies of the learned score at very small diffusion timesteps, where the model must capture the correct evolution of the data distribution. In this regime, diffusion models fail to satisfy the Fokker–Planck equation, which governs the evolution of the score. We interpret this deviation as one source of the observed inconsistencies and propose an energy-based diffusion model with a Fokker–Planck-derived regularization term to enforce consistency. We demonstrate our approach by sampling and simulating multiple biomolecular systems, including fast-folding proteins, and by introducing a state-of-the-art transferable Boltzmann emulator for dipeptides that supports simulation and achieves improved consistency and efficient sampling. Our code, model weights, and self-contained JAX and PyTorch notebooks are available at `https://github.com/noegroup/ScoreMD`.

## 1  Introduction

Understanding biochemical systems requires modeling not only static molecular structures but also their temporal evolution and interactions. Molecular dynamics (MD) simulations offer a principled framework to obtain such information from atomistic force fields and, with recent methodological and computational advances, can reach biologically relevant timescales (Lindorff-Larsen et al., 2011; Wolf et al., 2020). Nonetheless, fully atomistic simulations of large systems or slow conformational transitions remain computationally challenging (Kmiecik et al., 2016; Plattner et al., 2017). Coarse-graining (CG) methods address this limitation by grouping atoms into beads to accelerate simulations at the cost of physical resolution. In this reduced space, interactions and forces cannot be described accurately with traditional MD methods, requiring learning-based approaches to approximate forces and recover correct dynamics (Clementi, 2008; Noid, 2013; Husic et al., 2020; Charron et al., 2025).

Diffusion models (Ho et al., 2020; Song et al., 2021) have emerged as powerful generative tools capable of learning high-dimensional molecular distributions (Abramson et al., 2024; Watson et al., 2023; Corso et al., 2023; Plainer et al., 2023b; Lewis et al., 2025). They are trained to reverse

---

[†]Corresponding Authors:
    michael.plainer@fu-berlin.de, hwu81@sjtu.edu.cn, franknoe@microsoft.com

39th Conference on Neural Information Processing Systems (NeurIPS 2025).

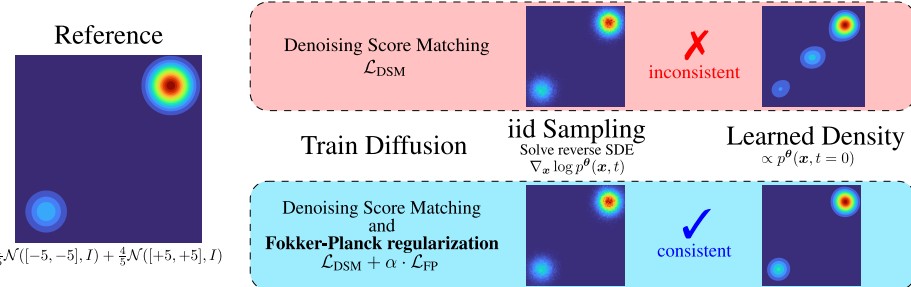

Figure 1: Training diffusion models on a 2D toy example reveals inconsistencies. While classical iid diffusion sampling (i.e., denoising) correctly reproduces both modes, evaluating the score at $t = 0$ to estimate the unnormalized density yields a third mode and an incorrect mass distribution. Such a diffusion model would produce incorrect dynamics while producing correct samples. A model trained with Fokker–Planck regularization is self-consistent, and aligns the learned score at $t = 0$ with the distribution recovered by diffusion sampling.

a stochastic diffusion process in which data is progressively perturbed with Gaussian noise over diffusion time $t$, starting from clean samples at $t = 0$ and approaching pure noise at $t = 1$. Generation then proceeds in reverse: random noise is iteratively denoised using the learned time-dependent *score* function $\nabla_{\boldsymbol{x}} \log p(\boldsymbol{x}, t)$, producing molecular configurations that approximate the equilibrium training distribution. While such models can efficiently generate independent equilibrium structures (Lewis et al., 2025), they lack temporal structure and cannot capture kinetic properties (Wang and Hou, 2011a,b) or transition mechanisms (Henkelman et al., 2000; Bolhuis et al., 2002). In contrast, MD yields both, thermodynamic and kinetic properties but has a high computational cost.

In principle, the score function at $t = 0$ of a diffusion model trained on Boltzmann-distributed molecular samples can serve as a surrogate for the physical forces. In an ideal energy-based diffusion model, this correspondence would ensure thermodynamic consistency between the learned energy landscape and the equilibrium distribution. In practice, however, diffusion models rarely behave as proper energy-based models, especially as system complexity increases (Arts et al., 2023). In this work, we address these limitations and propose a consistent energy-based diffusion model that can generate realistic equilibrium samples while providing physically consistent forces for simulation.

To address this, we leverage the Fokker–Planck equation, which governs the evolution of probability densities under diffusion processes (Särkkä and Solin, 2019), but is generally violated by existing diffusion models (Lai et al., 2023). By introducing a loss term during training that penalizes deviations from this equation, we improve model alignment and empirically enhance consistency between generated samples and simulated dynamics, as shown in Figure 1. In practice, we parameterize the score as the gradient of an energy function, providing access to a potential energy and conservative forces, which is the prerequisite to enable the use of established physical sampling methods (Torrie and Valleau, 1977; Laio and Parrinello, 2002) with coarse-grained diffusion models. Furthermore, by decomposing the diffusion timeline into smaller intervals and assigning a separate model to each, this regularization can be selectively applied to small diffusion timesteps (i.e., where it is needed most).

Our main contributions in this work are as follows:

1. We include a Fokker–Planck-based regularization when training energy-based diffusion models, which substantially improves simulation stability and enforces consistency between diffusion sampling and molecular dynamics simulations recovered from the learned forces. This enables accurate sampling and simulation of fast-folding proteins such as Chignolin and BBA.

2. We show that training on a small sub-interval of the diffusion time suffices for stable simulation. Combining this with smaller models trained on complementary intervals can yield more efficient training and inference while preserving high sampling quality.

3. We develop a state-of-the-art transferable Boltzmann emulator for dipeptides capable of fast high-quality independent sampling, and consistent long-time simulation.

## 2 Background

### 2.1 Generative Score-based Modeling

**Diffusion models** (Song and Ermon, 2019; Ho et al., 2020; Song et al., 2021) are self-supervised generative models that gradually corrupt the training data with noise and learn to reverse this stochastic

process. The forward process is defined by a stochastic differential equation (SDE)

$$\mathrm{d}\boldsymbol{x} = \boldsymbol{f}(\boldsymbol{x}, t)\,\mathrm{d}t + g(t)\,\mathrm{d}\boldsymbol{w}, \qquad (1)$$

where $\boldsymbol{w}$ denotes the standard Wiener process, and $\boldsymbol{f}$ and $g$ define the drift and diffusion coefficient respectively. To generate samples, diffusion models simulate the corresponding reverse-time SDE

$$\mathrm{d}\boldsymbol{x} = \left[\boldsymbol{f}(\boldsymbol{x}, t) - g^2(t)\nabla_{\boldsymbol{x}}\log p_t(\boldsymbol{x})\right]\mathrm{d}t + g(t)\,\mathrm{d}\bar{\boldsymbol{w}}. \qquad (2)$$

Starting from Gaussian noise at time $t = 1$, they iteratively denoise until a sample is produced at $t = 0$. Here, $p_t(\boldsymbol{x})$ denotes the density of the sample $\boldsymbol{x}$ at time $t$, which the model aims to approximate by learning the score $\nabla_{\boldsymbol{x}}\log p_t(\boldsymbol{x}, t)$. $\bar{\boldsymbol{w}}$ denotes the time-reversed Wiener process.

As for $\boldsymbol{f}$ and $g$, the choice depends on the specific diffusion formulation. In this work, we adopt the variance preserving (VP) SDE formulation introduced by Song et al. (2021) (compare Appendix A.1).

**Denoising score matching** (Vincent, 2011; Song et al., 2021) is a common way to train diffusion models by minimizing the squared error between a time-dependent learned score function $\boldsymbol{s}^{\boldsymbol{\theta}}(\boldsymbol{x}(t), t)$ and the true score of the transition kernel $p_{0t}(\boldsymbol{x}(t)\,|\,\boldsymbol{x}(0))$ conditioned on the training data $\boldsymbol{x}(0)$:

$$\boldsymbol{\theta}^* = \operatorname*{argmin}_{\boldsymbol{\theta}} \mathbb{E}_{t\sim\mathcal{U}(0,1)}\,\mathbb{E}_{\boldsymbol{x}(0)}\,\mathbb{E}_{\boldsymbol{x}(t)\,|\,\boldsymbol{x}(0)}\left[\lambda(t)\left\|\boldsymbol{s}^{\boldsymbol{\theta}}(\boldsymbol{x}(t), t) - \nabla_{\boldsymbol{x}(t)}\log p_{0t}(\boldsymbol{x}(t)\,|\,\boldsymbol{x}(0))\right\|_2^2\right], \quad (3)$$

where $\lambda(t)$ is a time-dependent weighting function. For brevity, we will denote the denoising diffusion loss as $\mathcal{L}_{\mathrm{DSM}}[\boldsymbol{s}^{\boldsymbol{\theta}}](\boldsymbol{x}, t) = \lambda(t)\left\|\boldsymbol{s}^{\boldsymbol{\theta}}(\boldsymbol{x}(t), t) - \nabla_{\boldsymbol{x}(t)}\log p_{0t}(\boldsymbol{x}(t)\,|\,\boldsymbol{x}(0))\right\|_2^2$.

**Parameterization and instabilities.** With an affine drift $\boldsymbol{f}$, we can write the closed-form solution of $p_{0t}$ as a Gaussian (Särkkä and Solin, 2019), and can efficiently train diffusion models such that

$$\boldsymbol{\theta}^* = \operatorname*{argmin}_{\boldsymbol{\theta}} \mathbb{E}_{t\sim\mathcal{U}(0,1)}\,\mathbb{E}_{\boldsymbol{x}(0)}\,\mathbb{E}_{\epsilon\sim\mathcal{N}(0,I)}\left[\lambda(t)\left\|\boldsymbol{s}^{\boldsymbol{\theta}}(\mu(\boldsymbol{x}(0), t) + \sigma(t)\epsilon, t) + \frac{\epsilon}{\sigma(t)}\right\|_2^2\right], \quad (4)$$

where $\mu(\boldsymbol{x}(0), t), \sigma(t)$ depend on the concrete choices for $\boldsymbol{f}$ and $g$. By construction, $\sigma(0) = 0$, which ensures correct interpolation between data and noise. Minimizing the denoising loss yields a neural network with parameters $\boldsymbol{\theta}$ that can approximate the unconditional score $\boldsymbol{s}^{\boldsymbol{\theta}}(\boldsymbol{x}, t) \approx \nabla_{\boldsymbol{x}}\log p_t(\boldsymbol{x})$.

As $t \to 0$, this parameterization introduces numerical instability, where $\sigma(t)$ vanishes and the loss explodes. This instability makes training difficult at small timescales (Kim et al., 2022) and is typically mitigated by truncating the training interval to $(\varepsilon, 1)$ for some $\varepsilon > 0$. While effective for training stability, this inherent instability in training limits the model's accuracy at small $t$, which is critical for applications requiring reliable scores close to the data manifold, as targeted in this work.

## 2.2 Boltzmann Distribution

**Molecular dynamics simulations** numerically integrate Newton's equations of motion to describe the time evolution of a molecular system. A widely used formulation is Langevin dynamics (Leimkuhler and Matthews, 2015), which corresponds to integrating the following set of SDEs

$$\mathrm{d}\boldsymbol{x} = \boldsymbol{v}\,\mathrm{d}t, \qquad \boldsymbol{M}\,\mathrm{d}\boldsymbol{v} = -\nabla_{\boldsymbol{x}}U(\boldsymbol{x})\,\mathrm{d}t - \gamma\boldsymbol{M}\boldsymbol{v}\,\mathrm{d}t + \sqrt{2\gamma k_B T}\,\mathrm{d}\boldsymbol{w}_t. \qquad (5)$$

$\boldsymbol{M}$ denotes the particle masses, $\boldsymbol{v}$ the velocities, $\gamma$ is a friction constant, $k_B T$ a constant, and $\boldsymbol{w}_t$ is the standard Brownian motion. Note that $t$ here refers to the physical time instead of the diffusion time used earlier. Integration of this system requires access to the forces $-\nabla_{\boldsymbol{x}}U$. However, in settings where direct force evaluation is not feasible, such as in CG models, a surrogate force function is required. In this work, we propose using the score $\boldsymbol{s}^{\boldsymbol{\theta}}(\boldsymbol{x}, t = 0)$ for this purpose, as we describe next.

**Extracting forces.** After a long, equilibrated MD simulation, samples follow the *Boltzmann distribution* (Boltzmann, 1868) such that $p(\boldsymbol{x}) = \exp(-\frac{U(\boldsymbol{x})}{k_B T})/Z$, where $U$ is the potential energy and $Z$ the normalization constant. Training a diffusion model on data from such a distribution yields at $t = 0$

$$\boldsymbol{s}^{\boldsymbol{\theta}}(\boldsymbol{x}, t = 0) \approx \nabla_{\boldsymbol{x}}\log p_{t=0}(\boldsymbol{x}) \qquad (6)$$

$$= \nabla_{\boldsymbol{x}}\log\exp\left(-\frac{U(\boldsymbol{x})}{k_B T}\right) - \nabla_{\boldsymbol{x}}\log Z$$

$$= -\nabla_{\boldsymbol{x}}\frac{U(\boldsymbol{x})}{k_B T} - 0.$$

Hence, the score at diffusion time $t = 0$ is proportional to $-\nabla_{\boldsymbol{x}} U(\boldsymbol{x})$, the forces acting on the system. This equivalence implies that any diffusion model trained on Boltzmann-distributed data can not only be used for independent sampling but also for molecular simulation by using the learned score as a force estimator with the SDEs of Equation (5). Unlike previous methods that rely on explicit force labels for training (Husic et al., 2020; Durumeric et al., 2023; Charron et al., 2025; Krämer et al., 2023), this relation allows us to learn a model directly from equilibrium samples. This is particularly useful for systems where potential energy and force labels are unavailable, as in CG modeling, which is the setting we consider. The central question we will answer next is whether and how this identity can be realized in practice and if the resulting model is sufficiently accurate.

## 3 Method

Diffusion models trained on Boltzmann-distributed molecular data provide a bridge between generative sampling and physical energy landscapes. At the diffusion endpoint $t = 0$, the score represents the forces acting on the system (Equation (6)), suggesting that a single model can capture the sampling process and the underlying potential energy. These two perspectives correspond to reconstructing the data distribution through denoising, $p(\boldsymbol{x})$, or evaluating the energy-based form at $t = 0$, $p_0(\boldsymbol{x})$:

$$p(\boldsymbol{x}) \sim \text{Denoising}[\boldsymbol{s}^{\boldsymbol{\theta}}](\boldsymbol{x}, t)$$
$$p_0(\boldsymbol{x}) \sim \exp\left(-\frac{U(\boldsymbol{x})}{k_B T}\right) \qquad \text{with } \boldsymbol{s}^{\boldsymbol{\theta}}(\boldsymbol{x}, 0) = -\nabla_{\boldsymbol{x}} \frac{U(\boldsymbol{x})}{k_B T}. \tag{7}$$

Ideally, these two formulations should agree such that $p(\boldsymbol{x}) = p_0(\boldsymbol{x})$. In practice, however, diffusion models often display inconsistencies (Koehler et al., 2023; Li et al., 2023; Bortoli et al., 2024). Prior attempts to use the score–force relation beyond static sampling can reproduce realistic ensembles under classical diffusion sampling $p(\boldsymbol{x})$ but fail to recover these distributions in simulation $p_0(\boldsymbol{x})$ (Arts et al., 2023), indicating that the learned model does not behave as a proper energy-based model.

We show that consistency can be recovered directly from unlabeled equilibrium data by enforcing the Fokker–Planck equation. This equation links energy functions across different diffusion times, effectively transferring accuracy from stable, large-$t$ regions to the small-$t$ regime. The result is a self-consistent diffusion model that learns physically meaningful forces without ever observing them.

### 3.1 Improving Consistency with the Fokker–Planck Equation

The Fokker–Planck equation (Øksendal, 2003; Särkkä and Solin, 2019) is a partial differential equation that describes how probability densities evolve in stochastic processes, including diffusion models. For the diffusion SDE introduced in Equation (1), the log-density formulation of the Fokker–Planck equation (Lai et al., 2023; Hu et al., 2025) can be written as

$$\partial_t \log p_t(\boldsymbol{x}) = \mathcal{F}_p(\boldsymbol{x}, t) \triangleq \frac{1}{2} g^2(t) \left[ \text{div}_{\boldsymbol{x}}(\nabla_{\boldsymbol{x}} \log p_t) + \|\nabla_{\boldsymbol{x}} \log p_t\|_2^2 \right] - \langle \boldsymbol{f}, \nabla_{\boldsymbol{x}} \log p_t \rangle - \text{div}_{\boldsymbol{x}}(\boldsymbol{f}), \tag{8}$$

where $\text{div}_{\boldsymbol{x}}$ denotes the divergence operator $\text{div}_{\boldsymbol{x}} \boldsymbol{F} = \text{tr}(\partial_{\boldsymbol{x}} \boldsymbol{F})$. This equation is a fundamental property of the diffusion process, and any well-trained model should satisfy Equation (8). However, as shown in Section 5, we observe that diffusion models violate this equation, particularly at small $t$, which is consistent with findings in prior work (Lai et al., 2023). Since the Fokker–Planck equation links the evolution of the score to that of the underlying density, such violations imply that the learned score does not evolve consistently with the density over diffusion time. Consequently, small deviations can accumulate, possibly leading to inconsistencies between the distribution $p(\boldsymbol{x})$ recovered through denoising and the instantaneous density $p_0(\boldsymbol{x})$ we want to recover.

**Fokker–Planck regularization.** To correct this, we introduce a regularization term that enforces compliance with Equation (8) during training, similar to Lai et al. (2023). Alongside the standard diffusion objective, we minimize the residual error

$$\|R(\boldsymbol{x}, t)\|_2^2 = \left\| \mathcal{F}_{p^{\boldsymbol{\theta}}}(\boldsymbol{x}, t) - \partial_t \log p_t^{\boldsymbol{\theta}}(\boldsymbol{x}) \right\|_2^2, \tag{9}$$

and define the corresponding loss as $\mathcal{L}_{\text{FP}}[\log p^{\boldsymbol{\theta}}](\boldsymbol{x}, t) = \lambda_{FP}(t) D^{-2} \|R(\boldsymbol{x}, t)\|_2^2$, where $\boldsymbol{x} \in \mathbb{R}^D$ and $\lambda_{FP}$ is a time-dependent weighting function, which we set to be the same as $\lambda$. As the regularization can be evaluated for all $\boldsymbol{x}, t$, we combine it with the score-matching objective

$$\underset{\boldsymbol{\theta}}{\arg\min} \, \mathbb{E}_{t \sim \mathcal{U}(0,1)} \, \mathbb{E}_{\boldsymbol{x}(0)} \, \mathbb{E}_{\boldsymbol{x}(t) \,|\, \boldsymbol{x}(0)} \left[ \mathcal{L}_{\text{DSM}}[\nabla_{\boldsymbol{x}} \log p^{\boldsymbol{\theta}}](\boldsymbol{x}(t), t) + \alpha \cdot \mathcal{L}_{\text{FP}}[\log p^{\boldsymbol{\theta}}](\boldsymbol{x}(t), t) \right], \tag{10}$$

where $\alpha$ is a hyperparameter that determines the regularization strength. The regularization $\mathcal{L}_{\text{FP}}$ term provides an additional training signal that stabilizes learning in the small-time regime, where the denoising loss becomes ill-conditioned and gradients are dominated by noise. By explicitly linking the temporal evolution of densities, the regularization encourages the model to remain consistent with the Fokker–Planck dynamics throughout diffusion time.

**Weak residual formulation.** The exact computation of the residual $R$ (and thus the loss $\mathcal{L}_{\text{FP}}$) involves costly higher-order derivatives, especially the divergence term $\text{div}_{\boldsymbol{x}}(\nabla_{\boldsymbol{x}} \log p^{\boldsymbol{\theta}})$ can be challenging to compute for high-dimensional data. To reduce this overhead, we introduce a series of approximations and begin by using a residual in the weak formulation similar to the one used by Guo et al. (2022)

$$\widetilde{R}(\boldsymbol{x}, t) = E_{\boldsymbol{v}} \left[ R(\boldsymbol{x} + \boldsymbol{v}, t) \right] \tag{11}$$

with $\boldsymbol{v} \sim \mathcal{N}(0, \sigma^2 I)$ and a small $\sigma > 0$. As $\sigma$ approaches 0, $\widetilde{R}(\boldsymbol{x}, t)$ will be equal to $R(\boldsymbol{x}, t)$. Using the weak residual formulation, $\widetilde{R}(\boldsymbol{x}, t)$ can be estimated by the following unbiased estimator, which only requires the computation of first-order derivatives (see Appendix A.2 for derivation)

$$\widetilde{R}(\boldsymbol{x}, t; \boldsymbol{v}) = \frac{1}{2} g^2(t) \left[ \left( \frac{\boldsymbol{v}}{\sigma} \right)^{\top} \frac{\boldsymbol{s}^{\boldsymbol{\theta}}(\boldsymbol{x} + \boldsymbol{v}, t) - \boldsymbol{s}^{\boldsymbol{\theta}}(\boldsymbol{x} - \boldsymbol{v}, t)}{2\sigma} + \left\| \boldsymbol{s}^{\boldsymbol{\theta}}(\boldsymbol{x} + \boldsymbol{v}, t) \right\|_2^2 \right] \tag{12}$$
$$- \langle \boldsymbol{f}(\boldsymbol{x} + \boldsymbol{v}, t), \boldsymbol{s}^{\boldsymbol{\theta}}(\boldsymbol{x} + \boldsymbol{v}, t) \rangle - \text{div}_{\boldsymbol{x}}(\boldsymbol{f}(\boldsymbol{x} + v, t)) - \partial_t \log p_t^{\boldsymbol{\theta}}(\boldsymbol{x} + \boldsymbol{v}),$$

where $\boldsymbol{s}^{\boldsymbol{\theta}} = \nabla_{\boldsymbol{x}} \log p^{\boldsymbol{\theta}}$. This yields an unbiased estimator of the squared residual

$$\mathcal{L}_{\text{FP}}(\boldsymbol{x}, t) \approx \left\| \widetilde{R}(\boldsymbol{x}, t) \right\|_2^2 \approx \widetilde{R}(\boldsymbol{x}, t; \boldsymbol{v}) \cdot \widetilde{R}(\boldsymbol{x}, t; \boldsymbol{v}'), \tag{13}$$

with $\boldsymbol{v}, \boldsymbol{v}' \sim \mathcal{N}(0, \sigma^2 I)$, yielding a computationally manageable estimate of $\mathcal{L}_{\text{FP}}$.

We further reduce computational cost by estimating $\partial_t \log p_t^{\boldsymbol{\theta}}$ using finite differences (Fornberg, 1988)

$$\partial_t \log p_t^{\boldsymbol{\theta}} \approx \frac{h_s^2 \log p_t^{\boldsymbol{\theta}}(\boldsymbol{x}, t + h_d) + (h_d^2 - h_s^2) \log p_t^{\boldsymbol{\theta}}(\boldsymbol{x}, t) - h_d^2 \log p_t^{\boldsymbol{\theta}}(\boldsymbol{x}, t - h_s)}{h_s h_d (h_s + h_d)}. \tag{14}$$

Since both $t$ and $\log p_t^{\boldsymbol{\theta}}$ are one-dimensional, finite differences serve as a robust estimate. This allows for computationally feasible approximation of the loss $\mathcal{L}_{\text{FP}}$, using multiple forward passes.

## 3.2 Physically Consistent Model Design

To model biochemical systems accurately and support Fokker–Planck regularization, we must enforce known physical constraints through an appropriate parameterization and choice of architecture.

**Conservative model parameterization.** In diffusion models, it is common to parameterize the score directly as $\boldsymbol{s}^{\boldsymbol{\theta}} = \nabla_{\boldsymbol{x}} \log p_t^{\boldsymbol{\theta}} = NNET(\boldsymbol{x}, t)$ rather than defining an explicit energy function and taking its gradient, $\boldsymbol{s}^{\boldsymbol{\theta}} = \nabla_{\boldsymbol{x}} NNET'(\boldsymbol{x}, t)$. While energy-based formulations have been explored previously for diffusion models (Song and Ermon, 2019), they are less common in practice (Du et al., 2023), since most applications require only the score and report no difference in sampling quality (Salimans and Ho, 2021). In our setting, however, we explicitly aim to construct a consistent energy-based diffusion model, ensuring the correspondence between the learned score and the underlying energy landscape as formalized in Equation (7). This gradient-based formulation provides access to the energy $\log p_t$, which is essential for enforcing physical consistency through the Fokker–Planck equation. Moreover, for molecular simulations, a conservative formulation, where the forces are derived from a well-defined energy, is critical for numerical stability and accurate force estimation (Schütt et al., 2017; Batzner et al., 2022; Arts et al., 2023), as further demonstrated in Appendix C.1.

**Architecture.** Our choice for the score is conservative, translation invariant, and learns $SO(3)$ equivariance. Similarly to Arts et al. (2023), we use a graph transformer (Shi et al., 2021), making the score permutation equivariant and achieving translation invariance by using pairwise distances instead of absolute coordinates. For the rotation equivariance, recent work (Abramson et al., 2024) has shown that the architecture itself does not need to enforce this property. Hence, we apply random rotations during training so that the network learns rotational equivariance via data augmentation.

The main part of the architecture can be summarized by describing the nodes $\boldsymbol{n}$ and edges $\boldsymbol{e}$ such that

$$\boldsymbol{e}_{ij} = \boldsymbol{x}_i - \boldsymbol{x}_j, \qquad \boldsymbol{n}_i^{(0)} = [\boldsymbol{a}_i, t], \qquad \boldsymbol{n}^{(l+1)} = \phi^{(l)}(\boldsymbol{n}^{(l)}, \boldsymbol{e}), \tag{15}$$

where $\boldsymbol{x}$ are the coarse-grained positions, $\boldsymbol{e}$ are the edge features, $\boldsymbol{a}$ are atom features, $t$ is the diffusion time, $\boldsymbol{n}^{(l)}$ are the node embeddings of layer $l$, and $\phi$ is one layer of the graph transformer. When training on a single molecule, we use one-hot atom types; for the transferable model, we use the atom identity, atom number, residue index, and amino acid type, following Klein and Noé (2024).

Finally, to achieve conservativeness, we map the last node embeddings $\boldsymbol{n}_i^{(L)} \in \mathbb{R}^K$ to scalar energies via $\psi : \mathbb{R}^K \to \mathbb{R}$, and compute the score as $\nabla_{\boldsymbol{x}} \sum_i \psi(\boldsymbol{n}_i^{(L)})$. Overall, this yields a translation-invariant, approximately rotation-equivariant, conservative architecture that also avoids issues caused by mirror symmetries (Trippe et al., 2023; Klein and Noé, 2024).

### 3.3 Mixture of Experts

Classical diffusion models are trained across the full diffusion timeline to enable generative sampling. In some applications, learning the score only at $t = 0$ can be sufficient, as it gives access to the potential. To support this, we introduce a time-based mixture of experts (MoE) approach, inspired by ideas from the image domain to improve efficiency (Balaji et al., 2023; Ganjdanesh et al., 2025). Rather than training a single model across the entire range $t \in (0, 1)$, we partition the interval into disjoint subintervals $\mathcal{I}_0, \mathcal{I}_1, \dots$ with $\bigcup_i \mathcal{I}_i = (0, 1)$, and assign a separate expert $\boldsymbol{s}_i^{\boldsymbol{\theta}}$ to each interval. The overall score can then be written by evaluating the correct model

$$\boldsymbol{s}^{\boldsymbol{\theta}}(\boldsymbol{x}, t) = \boldsymbol{s}_i^{\boldsymbol{\theta}}(\boldsymbol{x}, t), \quad \text{for} \quad t \in \mathcal{I}_i. \tag{16}$$

Thus, only one expert is active at any given $t$, which simplifies memory management and allows independent and parallel training of all experts. For simulation, only the expert corresponding to $t = 0 \in \mathcal{I}_i$ is needed, while iterative diffusion sampling sequentially loads every expert as $t$ decreases.

This design provides several advantages. For simulation, training only on the relevant small-$t$ range (e.g., $(0, 0.1)$) avoids wasting resources on unnecessary diffusion times that are never used. For generative sampling, dividing the timeline still offers clear benefits. Applying the Fokker–Planck loss from Equation (10) across all diffusion times can lead to overregularization at large $t$, degrading iid sampling performance. Since large-$t$ models do not require Fokker–Planck regularization or a conservative parameterization, using simpler unconstrained models improves generative quality and prevents unnecessary constraints. Moreover, the experts specialize in distinct temporal regions: small-$t$ experts capture fine structural details, while large-$t$ experts focus on coarse features (Ganjdanesh et al., 2025). This specialization stabilizes training and enhances the consistency between sampling and simulation beyond what could be achieved by simply increasing model capacity (see Appendix C.2). Finally, because only one expert is loaded into memory at a time, we can overall train more parameters without exceeding GPU limits. We will explore these benefits in Section 5.

## 4 Related Work

In recent years, a variety of deep learning methods have been proposed to enhance or replace molecular simulation. The work most closely related to ours is that of Arts et al. (2023), who employ an energy-based diffusion model for coarse-grained systems. However, their approach fails to maintain consistency between sampling and the learned score. They mitigate some of the inconsistencies we describe by evaluating the diffusion model at larger timesteps $t > 0$, which introduces additional noise and reduces structural fidelity. We compare against this model in Section 5. Raja et al. (2025) also use the score–force relation but for transition path sampling and could benefit from our approach to have more accurate forces. Daigavane et al. (2025) use the inherent noise of the diffusion process to converge to an equilibrium distribution faster, but they do not learn accurate scores either. Several works instead learn coarse-grained force fields via a force-matching objective (Husic et al., 2020; Köhler et al., 2023; Charron et al., 2025; Durumeric et al., 2024). Rather than training a model to represent the data distribution, these methods approximate the target forces. However, they typically need system-specific energy priors, relying heavily on domain knowledge.

Other approaches bypass MD sampling entirely, generating Boltzmann-distributed configurations either sequentially, by conditioning each sample on its predecessor (Dibak et al., 2022; Plainer et al., 2023a; Schreiner et al., 2023; Tamagnone et al., 2024), or completely independently (Noé et al., 2019; Wirnsberger et al., 2020; Köhler et al., 2020; Midgley et al., 2023; Klein et al., 2023b; Abdin and Kim, 2024; Kim et al., 2024; Wu and Noé, 2024; Schebek et al., 2024; Diez et al., 2024; Tan et al., 2025). These methods frequently leverage diffusion- or flow-based architectures. In the latter

case, for all-atom systems with known energy functions, they can guarantee asymptotically unbiased sampling via reweighting or integration into an MCMC scheme. Although this is often a desirable property, extending it to larger systems remains challenging, as CG is not directly possible.

The inaccurate behavior of the score function has also been studied in low-dimensional settings by Koehler et al. (2023); Li et al. (2023), who demonstrated inconsistencies and derived error bounds. Bortoli et al. (2024) propose to improve the score at small diffusion timesteps, when the probability distribution of the samples is known. However, this is not directly compatible with coarse-graining, as the potential and the derived forces are not known. Similarly to us, Lai et al. (2023) also proposed a Fokker–Planck-inspired regularization; however, their method applies a higher-order regularization to the score itself to improve iid sample quality, rather than enforcing consistency through the potential to improve the learned score. Daras et al. (2023) propose a regularization to ensure average consistency of the score, wheras our objective aims to ensures concistency of the energy over complete trajectories through the Fokker–Planck equation. Relatedly, Hu et al. (2025) propose a score-based solver for high-dimensional Fokker–Planck equations, focusing on solving general SDE forward problems, and Du et al. (2024) use the Fokker–Planck equation to describe MD as a series of Gaussians that can be integrated but do not recover the full unconditional Boltzmann distribution.

## 5 Experiments

In this section, we evaluate the consistency of diffusion models by comparing samples obtained through classical denoising (*iid*) with those generated by sequential Langevin simulations (*sim*) using forces derived with the score–force relation from Equation (6). In other words, we verify the alignment of Equation (7) for energy-based models and show superior performance. We demonstrate our approach (*ScoreMD*) on three biomolecular systems—alanine dipeptide, Chignolin, and BBA—and introduce a transferable model that generalizes across dipeptides, improving over existing state-of-the-art Boltzmann generators (Klein and Noé, 2024). The code, model weights, and self-contained notebooks in JAX and PyTorch are publicly available at `https://github.com/noegroup/ScoreMD`.

**Metrics.** As the main metric of interest, we compare the 2D free energy surfaces of the equilibrium distributions obtained by different methods. For dipeptides, we project the data onto the dihedral angles $\varphi$ and $\psi$, while for proteins we perform time-lagged independent component analysis (TICA) (Pérez-Hernández et al., 2013) on bond distances and dihedral angles to recover two representative coordinates. To quantify differences between free energy surfaces, we report the potential of mean force (PMF) error (Durumeric et al., 2024), which measures the squared distance between the negative logarithms of the sampled and reference densities in the projected space. This metric places higher weight on low-density regions compared to alternatives such as the Jensen-Shannon (JS) divergence. Additional details are provided in Appendix B.2.

**Baselines.** We train a conservative *Diffusion* model and re-implement the *Two For One* method (Arts et al., 2023) within the same continuous-time diffusion framework to ensure a fair comparison. Both models use identical architectures and training procedures and hence produce the same iid samples (up to numerical errors); their only difference lies in the diffusion time used during evaluation. Specifically, *Two For One* performs simulation at a finite diffusion time $t \gg 0$ to enhance stability, whereas *Diffusion* is evaluated at $t = 10^{-5}$. For transferability, we re-train the *Transferable Boltzmann generator (BG)* model (Klein and Noé, 2024) with coarse-graining and use it without reweighting.

### 5.1 Alanine Dipeptide and Fast-Folding Proteins

**Dataset.** For alanine dipeptide, we use 50k samples from an MD simulation in implicit solvent (Köhler et al., 2021), coarse-grained to five atoms: [C, N, CA, C, N]. For Chignolin and BBA, we use the dataset from Lindorff-Larsen et al. (2011), coarse-grained to one bead per amino acid (10 and 28 residues, respectively), and use 80% of the samples for training. For the *iid* setting, we generate the same number of samples as in the training set. For *sim*, we initialize 100 parallel simulations from random training conformations, and simulate with a timestep of 2 fs for a total length of 1.2 µs for alanine dipeptide, 855.6 ns for Chignolin, and 200 ns for BBA. All trajectories are downsampled to match the number of training samples for consistency with the *iid* setting.

**(In-)consistent sampling.** Figure 2 compares the free energy landscapes in the projected space for *iid* sampling and Langevin simulation (*sim*). While all methods reproduce the training distribution under *iid* sampling, simulation quality varies and reveals inconsistencies. For the small system alanine dipeptide, the equilibrium distributions appear similar across methods. However, as system size

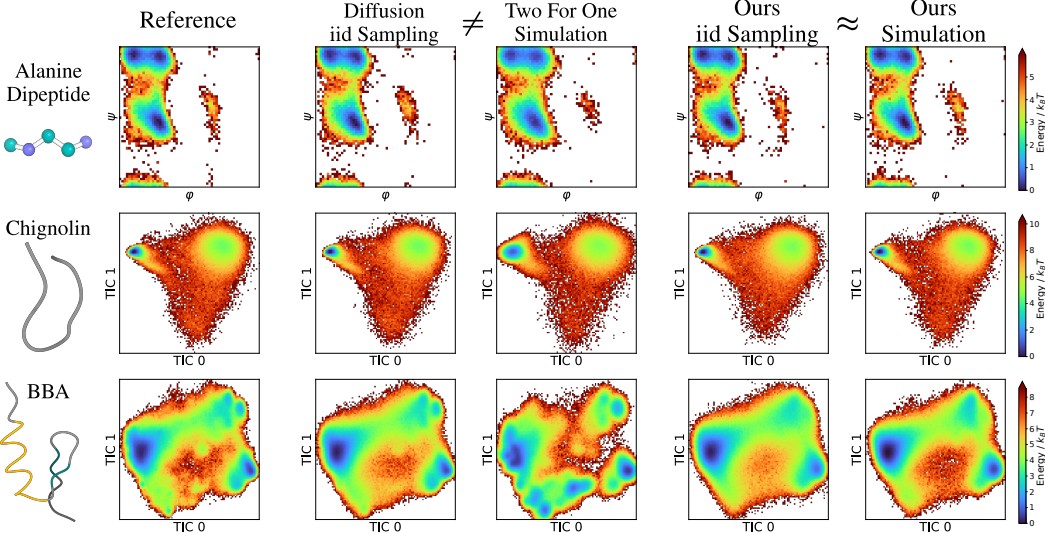

Figure 2: Comparison of equilibrium distributions obtained by *iid* sampling and Langevin simulation (*sim*) across different systems and methods. While classical *iid* sampling recovers the reference equilibrium distribution, performing simulation with the learned score reveals inconsistencies when models are not trained with Fokker–Planck regularization, i.e., $p(\boldsymbol{x}) \neq p_0(\boldsymbol{x})$. Regularized models achieve consistent behavior across systems.

| Method | | ALDP PMF ($\downarrow$) | Chignolin PMF ($\downarrow$) | BBA PMF ($\downarrow$) |
|---|---|---|---|---|
| Diffusion | iid | $0.098 \pm 0.006$ | $\mathbf{0.027 \pm 0.000}$ | $\mathbf{0.034 \pm 0.000}$ |
| Ours | iid | $\mathbf{0.097 \pm 0.008}$ | $0.035 \pm 0.001$ | $0.234 \pm 0.001$ |
| Two For One | sim | $0.206 \pm 0.004$ | $1.438 \pm 0.019$ | $1.624 \pm 0.107$ |
| Ours | sim | $\mathbf{0.091 \pm 0.004}$ | $\mathbf{0.038 \pm 0.006}$ | $\mathbf{0.254 \pm 0.005}$ |

Table 1: Quantitative comparison of sampling and simulation across different methods for alanine dipeptide (ALDP), Chignolin, and BBA. *Ours* yields significantly more consistent results between *iid* and *sim*.

Figure 3: Density of C–N bond length in Å for simulation of alanine dipeptide using different models.

increases, artifacts become more pronounced: for Chignolin, the *Two For One* simulation exhibits strong noise, and for BBA, the system separates into unphysical modes, yielding an incorrect equilibrium distribution. In contrast, our Fokker–Planck regularization enforces consistency, producing matching ensembles for *iid* sampling and Langevin simulation (*sim*), i.e., $p(\boldsymbol{x}) \approx p_0(\boldsymbol{x})$.

These trends can be seen in Table 1, which shows that *Ours* substantially improves simulation quality. We also observe a modest reduction in *iid* sampling performance, reflecting a tradeoff between generative fidelity and consistency, governed by the regularization strength $\alpha$. Larger values promote stronger alignment between sampling and simulation but may slightly degrade *iid* sampling accuracy.

**Noisy simulation.** *Two For One* improves consistency by evaluating the model at a larger diffusion time $t \gg 0$ during simulation, which introduces substantial noise that degrades structural fidelity. While *iid* sampling remains unaffected, simulations exhibit pronounced deviations, as seen for Chignolin in Figure 2. For alanine dipeptide, the free energy surface obtained with *Two For One* appears reasonable, but because the method relies on "noisy" forces, the observables are also affected by this noise. For instance, bond-length distributions will be incorrect, as depicted in Figure 3.

**Dynamics.** A key advantage of a *consistent* energy-based diffusion model is that it provides access not only to generative diffusion sampling but also to a physically meaningful energy function usable for downstream tasks such as MD simulation. As such, we can recover kinetic information and temporal behavior, as studied in Figure 4 for BBA, which is not accessible with classical diffusion sampling. Additional analysis of the dynamics, such as transition probabilities between states and the corresponding results for Chignolin, is provided in Appendix C.6.4.

**Force matching.** CG models are typically trained using force matching rather than diffusion-based objectives. In Appendix C.5.4, we benchmark our method against recent force-matching approaches for alanine dipeptide. Although these baselines often rely on explicit physical priors and force labels, our diffusion-based model achieves ensembles that more closely match the reference distribution.

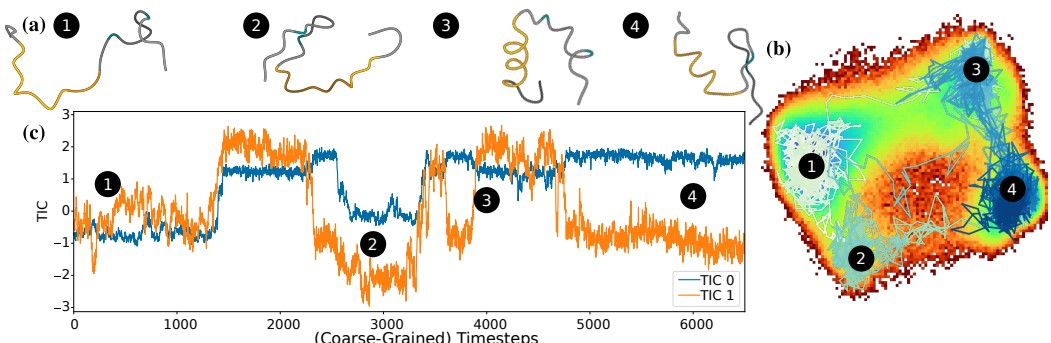

Figure 4: Recovering and analyzing the dynamic behavior of BBA. **(a)** 3D molecular structures from a single MD trajectory generated by our model. **(b)** Trajectory projected onto the first two TIC coordinates. **(c)** Illustrating the TIC coordinates over the coarse-grained (and subsampled) timesteps.

## 5.2 Transferability Across Dipeptides (Two Amino Acids)

**Dataset.** We use the dataset introduced by Klein et al. (2023a), which contains 49k samples from implicit-solvent simulations of $1\,\mu s$ covering all possible combinations of the 20 standard amino acids. From these, we use 175 dipeptides for training. Each dipeptide is coarse-grained by retaining the atoms [N, CA, CB, C, O] for each amino acid, a common CG resolution (Charron et al., 2025). With this, up to 10 atoms are retained per molecule, as seen in Figure 5 (a). For *iid* sampling, we draw 49k independent samples, while for *sim* we perform $30\,\mathrm{ns}$ of Langevin simulation per dipeptide, initialized from 10 random conformations with a timestep of $0.5\,\mathrm{fs}$ downsampled to 49k frames.

**Models.** We ablate the individual contributions of the Fokker–Planck regularization, the impact of MoE, and the effect of evaluating models at larger diffusion timesteps $t \gg 0$. Similar ablation studies for previous results are provided in Appendices C.5 and C.6. Specifically, we compare the *Diffusion* and *Two For One* baselines with three variants: *Mixture* denotes the MoE scheme, consisting of three experts trained on the intervals $(0, 0.1)$, $[0.1, 0.6)$, and $[0.6, 1.0)$. The models were robust to moderate changes in these ranges, provided the smallest-time expert was trained on a sufficiently large subinterval. All three experts are combined for *iid* sampling, while only the smallest-timescale model is used for simulation. The experts corresponding to larger timescales are reduced in size and complexity. *Fokker–Planck* refers to a single model trained with the loss from Equation (10). *Both* combines these two approaches, where the regularization is applied only to the smallest-timescale expert within the MoE setup. This corresponds to the model referred to previously as *Ours*.

**Overdispersion.** As shown in Figure 5 (b), all models can generate independent samples that closely match the reference distribution. However, samples from the *Transferable BG* model exhibit increased noise and over-explore low-probability states. In the original work, this behavior is corrected through sample reweighting, which we omit here and instead treat the model simply as a Boltzmann emulator. While the resulting differences are within standard deviations, the model yields a higher mean with comparable variance (see Table 2). Moreover, note that this model does not support simulation. When using the score for simulation, *Two For One* produces broader, overdispersed distributions, as shown in Figure 5 (b). This overdispersion also propagates to structural features such as inter-atomic distances, consistent with the behavior observed in previous systems (see Appendix C.8.2). More metrics and evaluation for more dipeptides can be found in Appendix C.8.

**Advantages of mixture.** The MoE architecture can significantly improve simulation quality compared to *Diffusion*. For the specific dipeptide shown in Figure 5 (c), *Both* further enhances performance over *Fokker–Planck*, improving both *iid* and simulated distributions. While the degree of improvement varies across dipeptides—with some cases showing comparable results (see Appendix C.8.3)—MoE generally increases simulation stability, as reflected in the lower PMF errors reported in Table 2.

Since *Mixture* and *Both* contain more parameters (see Appendix B.5), part of the improvement can be attributed to increased capacity. However, MoE also offers clear computational advantages: by applying regularization only at small diffusion times and using smaller, unconstrained models for larger timescales, it reduces training and inference cost, cutting sampling time by more than 50% in this case (see Appendix C.3) and prevents overregularization at large diffusion timesteps. Moreover, additional ablation studies in Appendix C.2 show that the benefits of MoE extend beyond parameter scaling, demonstrating that training on smaller temporal subranges improves robustness and that independently trained experts can be seamlessly combined into a coherent model.

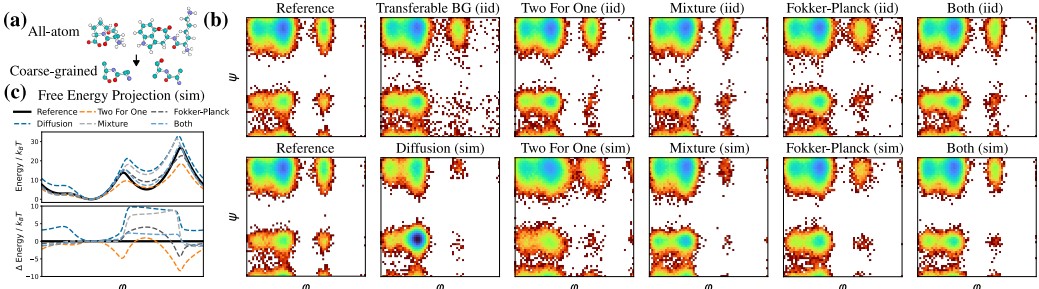

Figure 5: Comparison of methods on testset dipeptide KS. **(a)** The coarse-graining scheme. **(b)** Comparison of the Ramachandran plots of different methods for iid sampling and Langevin simulation. **(c)** The projection of the free energy surface and differences along the dihedral angle $\varphi$ for samples generated with simulation.

| Method | iid PMF ($\downarrow$) | sim. PMF ($\downarrow$) |
|---|---|---|
| Transferable BG | **0.230 ± 0.119** | - |
| Diffusion | **0.206 ± 0.159** | 6.515 ± 3.175 |
| Two For One | **0.203 ± 0.149** | 0.741 ± 0.319 |
| Mixture | **0.200 ± 0.127** | 0.658 ± 0.407 |
| Fokker–Planck | **0.241 ± 0.105** | **0.368 ± 0.267** |
| Both | **0.199 ± 0.127** | **0.203 ± 0.104** |

Table 2: Comparison of metrics across all testset dipeptides with PMF error. To compute the mean and standard deviation, we have averaged the results across all evaluated dipeptides. Lower values are better.

Figure 6: Comparing the Fokker–Planck error for $\log p^{\boldsymbol{\theta}}$ of multiple models.

**Fokker–Planck error.** Figure 6 shows the deviation from the Fokker–Planck equation, quantified as $\left\| \mathcal{F}_{p^{\theta}}(\boldsymbol{x}, t) - \partial_t \log p_t^{\boldsymbol{\theta}}(\boldsymbol{x}) \right\|_2$, plotted on a log scale. For models using MoE, this error is evaluated only up to $t = 0.1$, since only the small-timescale model is energy-based. Across all methods, the error is largest near $t = 0$. Applying the Fokker–Planck regularization significantly reduces this error, correlating with the improved sampling-simulation consistency observed earlier.

Interestingly, while *Mixture* improves consistency, its Fokker–Planck error at $t = 0$ remains comparable to that of unregularized models. This suggests that Fokker–Planck regularization and MoE improve consistency through different mechanisms, which explains why combining them outperforms either approach on its own, making *Both* again clearly the best model (compare Table 2).

## 6 Conclusion, Limitations and Future Work

In this work, we investigated energy-based diffusion models and analyzed the discrepancy between the density recovered through denoising and the learned energy at diffusion time $t = 0$. We showed that diffusion models are generally inconsistent, and these two densities do not coincide. To address this, we introduced a Fokker–Planck-based regularization on the model's energy, demonstrating that reducing deviations from the Fokker–Planck equation substantially improves model consistency. Furthermore, we found that focusing training on smaller diffusion times further enhances simulation quality. With these improvements, we obtain a physically consistent energy-based model that can generate independent equilibrium samples and recover realistic molecular dynamics from the same learned potential, providing access to both static and kinetic properties. This is particularly useful in coarse-grained settings where direct force information is not available. We validated these findings across multiple systems, trained on fast-folding proteins, and generalized across dipeptides.

Despite the theoretical motivation behind our approach, the results presented are primarily empirical. While our findings indicate that reducing the Fokker–Planck deviation improves consistency, this is unlikely to be the only source of simulation error. In fact, due to the fundamental differences between diffusion sampling and Langevin simulation, perfect alignment may not be achievable without limiting model expressivity. Moreover, evaluating the Fokker–Planck residual introduces computational overhead, which we mitigate through a weak residual formulation, though it still requires multiple forward passes during training and thus remains computationally demanding. Future work could extend this framework to larger molecular systems and explore transfer learning across biomolecular families. Another promising direction is to fine-tune existing diffusion models with the proposed regularization to explicitly correct Fokker–Planck deviations and enable simulation.

## Acknowledgments and Disclosure of Funding

The authors would like to thank Aleksander Durumeric, Yaoyi Chen, Maximilian Schebek, Klara Bonneau, Winfried Ripken, Marcel Kollovieh, Nicholas Gao, Matej Mezera, Yuanqi Du, Jiajun He, and Jungyoon Lee for the fruitful discussions and their helpful input. The work of Michael Plainer was supported by the Konrad Zuse School of Excellence in Learning and Intelligent Systems (ELIZA) through the DAAD program Konrad Zuse Schools of Excellence in Artificial Intelligence, sponsored by the Federal Ministry of Education and Research. The work of Hao Wu was supported by the National Natural Science Foundation of China, Grant No. 12171367. Moreover, we gratefully acknowledge support by the Deutsche Forschungsgemeinschaft (SFB1114, Projects No. A04 and No. B08) and the Berlin Mathematics center MATH+ (AA1-10). We gratefully acknowledge the computing time made available to them on the high-performance computer "Lise" at the NHR Center NHR@ZIB. This center is jointly supported by the Federal Ministry of Education and Research and the state governments participating in the NHR (`www.nhr-verein.de/unsere-partner`).

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

# A   Proofs, Derivations, and Formalisms

## A.1   Diffusion

We have opted to use VP diffusion (Song et al., 2021) throughout the paper, and thus, the drift and diffusion coefficient can be written as

$$\boldsymbol{f}(\boldsymbol{x}, t) = -\frac{1}{2}\beta(t)\boldsymbol{x}, \quad g(t) = \sqrt{\beta(t)}, \tag{17}$$

where

$$\beta(t) = \beta_{\min} + t \cdot (\beta_{\max} - \beta_{\min}), \tag{18}$$

with the hyperparameters from Song et al. (2021) such that $(\beta_{\min}, \beta_{\max}) = (0.1, 20)$. For this noise schedule to be suitable for molecules, it is important that we normalize the data to have unit variance.

With this specific choice for $\boldsymbol{f}$ and $g$, we can write the transition kernel as a Gaussian (Särkkä and Solin, 2019) with a moving mean and standard deviation such that

$$p_t(\boldsymbol{x}(t) \mid \boldsymbol{x}(0)) = \mathcal{N}(\boldsymbol{x}(t); \mu(\boldsymbol{x}(0), t), \sigma(t)I) \tag{19}$$

$$= \mathcal{N}(\boldsymbol{x}(t); e^{-\frac{1}{2}\int_0^t \beta(s)ds}\boldsymbol{x}(0), (1 - e^{-\int_0^t \beta(s)ds})I). \tag{20}$$

## A.2   Residual Loss

In this section, we prove Equation (12) and show that $\widetilde{R}(\boldsymbol{x}, t; \boldsymbol{v})$ can be used to get an unbiased estimation of $\widetilde{R}(\boldsymbol{x}, t)$. For simplicity of notation, let us express the log Fokker–Planck equations from Equation (8) as

$$\frac{1}{2}g^2(t)\operatorname{div}_{\boldsymbol{x}} \boldsymbol{s}^{\boldsymbol{\theta}}(\boldsymbol{x}, t) + \gamma_{\boldsymbol{\theta}}(\boldsymbol{x}, t) = 0, \tag{21}$$

where $\boldsymbol{s}^{\boldsymbol{\theta}}(\boldsymbol{x}, t) = \nabla_{\boldsymbol{x}} \log p_t^{\boldsymbol{\theta}}(\boldsymbol{x})$, and $\gamma_{\boldsymbol{\theta}}$ involves only the first-order gradient of $\log p_t^{\boldsymbol{\theta}}$. Here we define the "weak" residual (Guo et al., 2022) of the above equations for each $(\boldsymbol{x}, t)$

$$\widetilde{R}(\boldsymbol{x}, t) = \mathbb{E}_{\boldsymbol{v} \sim \mathcal{N}(0, \sigma^2 I)}\left[\frac{1}{2}g^2(t)\operatorname{div}_{\boldsymbol{x}} \boldsymbol{s}^{\boldsymbol{\theta}}(\boldsymbol{x} + \boldsymbol{v}, t) + \gamma_{\boldsymbol{\theta}}(\boldsymbol{x} + \boldsymbol{v}, t)\right], \tag{22}$$

where $\sigma > 0$ is a small number. It can be seen that residuals are zero if the two parts of the equations are exactly equal. We now aim to get the unbiased estimation of the above residual, without calculating high-order derivatives.

As such, we can show that for an arbitrary $t$,

$$
\begin{aligned}
\mathbb{E}_{\boldsymbol{v}}\left[\operatorname{div}_{\boldsymbol{x}} \boldsymbol{s}^{\boldsymbol{\theta}}(\boldsymbol{x} + \boldsymbol{v}, t)\right] &= \int \frac{\exp\left(-\frac{\boldsymbol{v}^\top \boldsymbol{v}}{2\sigma^2}\right)}{(2\pi\sigma^2)^{\frac{D}{2}}} \cdot \operatorname{div}_{\boldsymbol{x}} \boldsymbol{s}^{\boldsymbol{\theta}}(\boldsymbol{x} + \boldsymbol{v}, t)dv \tag{23}\\
&= -\frac{1}{(2\pi\sigma^2)^{\frac{D}{2}}} \int \left\langle \nabla_{\boldsymbol{v}} \exp\left(-\frac{\boldsymbol{v}^\top \boldsymbol{v}}{2\sigma^2}\right), \boldsymbol{s}^{\boldsymbol{\theta}}(\boldsymbol{x} + \boldsymbol{v}, t) \right\rangle dv \tag{24}\\
&= \frac{1}{(2\pi\sigma^2)^{\frac{D}{2}}} \int \exp\left(-\frac{\boldsymbol{v}^\top \boldsymbol{v}}{2\sigma^2}\right) \frac{\boldsymbol{v}^\top}{\sigma^2} \boldsymbol{s}^{\boldsymbol{\theta}}(\boldsymbol{x} + \boldsymbol{v}, t)dv \tag{25}\\
&= \mathbb{E}_{\boldsymbol{v}}\left[\frac{\boldsymbol{v}^\top \boldsymbol{s}^{\boldsymbol{\theta}}(\boldsymbol{x} + \boldsymbol{v}, t)}{\sigma^2}\right] \tag{26}\\
&= \frac{1}{2}\mathbb{E}_{\boldsymbol{v}}\left[\frac{\boldsymbol{v}^\top \boldsymbol{s}^{\boldsymbol{\theta}}(\boldsymbol{x} + \boldsymbol{v}, t)}{\sigma^2} - \frac{\boldsymbol{v}^\top \boldsymbol{s}^{\boldsymbol{\theta}}(\boldsymbol{x} - \boldsymbol{v}, t)}{\sigma^2}\right] \tag{27}\\
&= \mathbb{E}_{\boldsymbol{v}}\left[\left(\frac{\boldsymbol{v}}{\sigma}\right)^\top \frac{\boldsymbol{s}^{\boldsymbol{\theta}}(\boldsymbol{x} + \boldsymbol{v}, t) - \boldsymbol{s}^{\boldsymbol{\theta}}(\boldsymbol{x} - \boldsymbol{v}, t)}{2\sigma}\right], \tag{28}
\end{aligned}
$$

where $\boldsymbol{x} \in \mathbb{R}^D, \boldsymbol{v} \in \mathbb{R}^D$ and $t \in \mathbb{R}$.

Based on this, we can obtain

$$\widetilde{R}(\boldsymbol{x}, t) = \mathbb{E}_{\boldsymbol{v}} \left[ \widetilde{R}(\boldsymbol{x}, t; \boldsymbol{v}) \right] \tag{29}$$

$$= \mathbb{E}_{\boldsymbol{v}} \left[ \frac{1}{2} g^2(t) \left( \frac{\boldsymbol{v}}{\sigma} \right)^{\top} \frac{\boldsymbol{s}^{\boldsymbol{\theta}}(\boldsymbol{x} + \boldsymbol{v}, t) - \boldsymbol{s}^{\boldsymbol{\theta}}(\boldsymbol{x} - \boldsymbol{v}, t)}{2\sigma} + \frac{\gamma_{\boldsymbol{\theta}}(\boldsymbol{x} + \boldsymbol{v}, t) + \gamma_{\boldsymbol{\theta}}(\boldsymbol{x} - \boldsymbol{v}, t)}{2} \right].$$

Hence, $\widetilde{R}(\boldsymbol{x}, t; \boldsymbol{v})$ is an unbiased estimation of $\widetilde{R}(\boldsymbol{x}, t)$ by drawing a single sample $\boldsymbol{v} \sim \mathcal{N}(0, \sigma^2 I)$.

In practice, we can further reduce the computational overhead by only using a single approximation for $\gamma_{\boldsymbol{\theta}}$, and defining

$$\widetilde{R}(\boldsymbol{x}, t; \boldsymbol{v}) = \frac{1}{2} g^2(t) \left( \frac{\boldsymbol{v}}{\sigma} \right)^{\top} \frac{\boldsymbol{s}^{\boldsymbol{\theta}}(\boldsymbol{x} + \boldsymbol{v}, t) - \boldsymbol{s}^{\boldsymbol{\theta}}(\boldsymbol{x} - \boldsymbol{v}, t)}{2\sigma} + \gamma_{\boldsymbol{\theta}}(\boldsymbol{x} + \boldsymbol{v}, t). \tag{30}$$

We found $\sigma = 10^{-4}$ to be an effective choice throughout our experiments.

### A.2.1 Relation with Hutchinson Trace Estimation

In fact, our treatment of the divergence term in the weak residual formulation is closely related to the Hutchinson's trace estimator used by Albergo and Vanden-Eijnden (2025) for computing the PINN objective. Both approaches utilize Gaussian perturbations of $\boldsymbol{x}$ to obtain unbiased estimates of the divergence. However, it is important to note a key difference in how the residual is defined. As described in Equation (29), in our weak residual $\widetilde{R}(\boldsymbol{x}, t)$, we apply the same Gaussian perturbation not only to the divergence term but also to other terms in the residual. This means that when the strong residual $R(\boldsymbol{x}, t)$ is exactly zero, the weak residual $\widetilde{R}(\boldsymbol{x}, t) = \mathbb{E}[R(\boldsymbol{x} + \boldsymbol{v}, t)]$ also remains exactly zero, regardless of the perturbation variance $\sigma^2$, avoiding truncation errors. This distinction allows our weak residual formulation to maintain consistency even under large perturbations.

### A.3 Finite Difference Approximation

To approximate $\partial_t \log p^{\boldsymbol{\theta}}$, we relied on a finite difference approximation Fornberg (1988), as stated in Equation (14). For this estimation, we have followed the work of Lai et al. (2023), and used the hyperparameters that they suggested $(h_s, h_d) = (0.001, 0.0005)$.

### A.4 Connection with Flow Matching

The score–force relation introduced in Equation (6) is not specific to diffusion models. It also applies to other generative models, such as flow matching (Lipman et al., 2023), and in general to any model that can estimate $\nabla_{\boldsymbol{x}} \log p_{t=0}(\boldsymbol{x})$. In standard Gaussian flow matching, the learned vector field $v^{\boldsymbol{\theta}}$ relates to the score through

$$\nabla_{\boldsymbol{x}} \log p_t(\boldsymbol{x}) = \frac{1}{1-t} \left( t, v^{\boldsymbol{\theta}}(\boldsymbol{x}, t) - \boldsymbol{x} \right), \tag{31}$$

as discussed by Lipman et al. (2024). Thus, forces can also be obtained from flow-matching models via reparameterization. However, in our experiments, we found that flow-based models perform worse near $t \approx 0$, likely because the stochasticity inherent to diffusion models improves generalization in this regime.

### A.5 Formalization Coarse-Craining

In coarse-graining, we aim to reduce the number of dimensions of our system by combining multiple atoms into individual beads. Given non-CG samples $\boldsymbol{x}$, the Boltzmann distribution of CG samples $\boldsymbol{z}$ can be recovered by

$$p(\boldsymbol{z}) \propto \int \exp \left( -\frac{U(\boldsymbol{x})}{k_B T} \right) \delta(\Xi(\boldsymbol{x}) - \boldsymbol{z}) \, \mathrm{d}\boldsymbol{x}, \tag{32}$$

which defines the CG potential up to a constant. $\delta$ is the Dirac delta function.

## B Implementation Details and Experimental Setup

In this section, we will discuss additional details for the main experiments presented in the paper.

## B.1 Langevin Integrator

We perform NVT dynamics with the Langevin integrator as implemented in `openMM` (Eastman et al., 2017) version 8.2.0. The update reads

$$\boldsymbol{v}^{t+\Delta t} = \alpha\boldsymbol{v}^t - \frac{1-\alpha}{\gamma}\nabla_{\boldsymbol{x}}U(\boldsymbol{x})\boldsymbol{M}^{-1} + \sqrt{k_B T(1-\alpha)^2\boldsymbol{M}^{-1}}\boldsymbol{R}, \tag{33}$$

$$\boldsymbol{x}^{t+\Delta t} = \boldsymbol{x}^t + \Delta t \cdot \boldsymbol{v}^{t+\Delta t},$$

with $\alpha = \exp(-\gamma\Delta t)$. Here, $t$ and $\Delta t$ index the simulation timesteps (not the diffusion time), $\gamma$ is the friction coefficient, $\boldsymbol{M}$ the mass matrix, $\boldsymbol{R} \sim \mathcal{N}(\boldsymbol{0}, \boldsymbol{1})$ a standard normal random vector, $k_B$ the Boltzmann constant, and $T$ the temperature.

## B.2 Metrics

To compute the JS divergence and the PMF error, we first project our data onto either the dihedral angles or the first two TIC coordinates (Pérez-Hernández et al., 2013) and then discretize the observed free energy into binned histograms. For the JS divergence, we then compute the JS distance between the two probability vectors (we flatten the 2D histograms). To prevent discontinuities, we assume that in each bin there is at least one observation by adding 1.

As for the PMF error, we discretize into 64 bins for the dipeptides and into 100 bins for the proteins and compute the proportion of samples in each bin. These are then transformed by taking the $\log$ in each bin and then computing the square loss, which is averaged over all bins. Similarly, we have ensured that each bin contains some data and have added $10^{-6}$ as a baseline proportion. The approach and implementation are analogous to Durumeric et al. (2024).

## B.3 Alanine Dipeptide

**Dataset.** The alanine dipeptide datasets is available as part of the public `bgmol` (MIT licence) repository here: `https://github.com/noegroup/bgmol`. The dataset was generated with an MD simulation, using the classical *Amber ff99SBildn* force-field at 300K for implicit solvent for a duration of 1 ms Köhler et al. (2021) with the `openMM` library (Eastman et al., 2017). For training, we have selected 50k random samples from this simulation.

**Architecture.** For alanine dipeptide we have used quite a small architecture, where the hyperparameters are listed in Table 3. When multiple parameters are listed for the same model, this means that they are used for the corresponding MoE model. Note that when using MoE, we have mostly used the same model architecture, except that only the Fokker–Planck regularized model is conservative. As for the optimizer, we have used AdamW (Loshchilov and Hutter, 2019).

| Parameter | Diffusion | Mixture | Fokker–Planck | Both |
|---|---|---|---|---|
| Epochs | 10000 | 7000, 2000, 1000 | 10000 | 7000, 2000, 1000 |
| Max Learning Rate | $3 \cdot 10^{-4}$ | $3 \cdot 10^{-4}$ | $3 \cdot 10^{-4}$ | $3 \cdot 10^{-4}$ |
| Min Learning Rate | $10^{-5}$ | $10^{-5}$ | $10^{-5}$ | $10^{-5}$ |
| BS | 1024 | 1024 | 1024 | 1024 |
| Attention Heads | 8 | 8 | 8 | 8 |
| Feature Dim | 16 | 16 | 16 | 16 |
| Model-Ranges | $(0, 1)$ | $(0, 0.1), [0.1, 0.6), [0.6, 1.0)$ | $(0, 1)$ | $(0, 0.1), [0.1, 0.6), [0.6, 1.0)$ |
| Conservative | Yes | Yes, No, No | Yes | Yes, No, No |
| $\alpha$ | 0 | 0, 0, 0 | $5 \cdot 10^{-4}$ | $10^{-3}, 0, 0$ |
| Hidden Dimension | 96 | 96, 96, 96 | 96 | 96, 96, 96 |
| Layers | 2 | 2, 2, 2 | 2 | 2, 2, 2 |

Table 3: **Alanine dipeptide.** This table contains the hyperparameters for the different models shown.

**Simulation.** To perform Langevin simulation, we have extracted the forces from the model via Equation (6) at $t = 10^{-5}$ for all models except for *Two For One*, where we chose $t = 0.02$, which is the same value as presented in Arts et al. (2023) for the same number of training samples.

**Evaluation.** To compute the standard deviation in the experiments, we have trained the same model with three different seeds.

## B.4  Fast-Folding Proteins

**Dataset.** We evaluate our model on the proteins Chignolin (10 amino acids) and BBA (28 amino acids) using the data from Lindorff-Larsen et al. (2011). The reference data was simulated in explicit solvent with a step size of $2\,\mathrm{fs}$ at 340K for Chignolin and 325K for BBA. In total $106\,\mu s$ and $223\,\mu s$ were simulated for Chignolin and BBA. For training, we have randomly selected 80% of the samples which accounts to $427,794$ and $1,300,156$ samples for Chignolin and BBA respectively.

**Architecture.** We have used the same architecture as for alanine dipeptide, with the only difference being that we use larger networks. Compare the hyperparameters in Table 4 for Chignolin and Table 5 for BBA. Note that for *Two For One* we used the same model as *Diffusion* but evaluated it at a different $t$ for simulation. The hyperparameters slightly deviate from the choices in (Arts et al., 2023) to produce consistent results with our method. As optimizer, we used AdamW (Loshchilov and Hutter, 2019).

| Parameter | Diffusion | Mixture | Fokker–Planck | Both |
|---|---|---|---|---|
| Epochs | 2000 | 1700, 200, 100 | 2000 | 1700, 200, 100 |
| Max Learning Rate | $3 \cdot 10^{-4}$ | $3 \cdot 10^{-4}$ | $10^{-4}$ | $3 \cdot 10^{-4}$ |
| Min Learning Rate | $10^{-5}$ | $10^{-5}$ | $10^{-5}$ | $10^{-5}$ |
| BS | 1024 | 1024 | 1024 | 1024 |
| Attention Heads | 8 | 8 | 8 | 8 |
| Feature Dim | 16 | 16 | 16 | 16 |
| Model-Ranges | $(0,1)$ | $(0,0.1), [0.1,0.6), [0.6,1.0)$ | $(0,1)$ | $(0,0.1), [0.1,0.6), [0.6,1.0)$ |
| Conservative | Yes | Yes, No, No | Yes | Yes, No, No |
| $\alpha$ | 0 | 0, 0, 0 | $2 \cdot 10^{-3}$ | $2 \cdot 10^{-3}, 0, 0$ |
| Hidden Dimension | 128 | 128, 96, 96 | 128 | 128, 96, 96 |
| Layers | 3 | 3, 2, 2 | 2 | 3, 2, 2 |

Table 4: **Chignolin.** This table contains the hyperparameters for the different models shown.

| Parameter | Diffusion | Mixture | Fokker–Planck | Both |
|---|---|---|---|---|
| Epochs | 1500 | 1200, 400, 200 | 1500 | 1200, 400, 200 |
| Max Learning Rate | $3 \cdot 10^{-4}$ | $3 \cdot 10^{-4}$ | $10^{-4}$ | $10^{-4}$ |
| Min Learning Rate | $10^{-5}$ | $10^{-5}$ | $10^{-5}$ | $10^{-5}$ |
| BS | 1024 | 1024 | 1024 | 1024 |
| Attention Heads | 8 | 8 | 8 | 8 |
| Feature Dim | 16 | 16 | 16 | 16 |
| Model-Ranges | $(0,1)$ | $(0,0.1), [0.1,0.6), [0.6,1.0)$ | $(0,1)$ | $(0,0.1), [0.1,0.6), [0.6,1.0)$ |
| Conservative | Yes | Yes, No, No | Yes | Yes, No, No |
| $\alpha$ | 0 | 0, 0, 0 | $10^{-2}$ | $2 \cdot 10^{-2}, 0, 0$ |
| Hidden Dimension | 128 | 128, 128, 128 | 128 | 128, 128, 128 |
| Layers | 3 | 3, 3, 3 | 3 | 3, 3, 3 |

Table 5: **BBA.** This table contains the hyperparameters for the different models shown.

**Simulation.** To perform Langevin simulation, we have extracted the forces from the model via Equation (6) at $t = 10^{-5}$ for all models except for *Two For One*, where we chose $t = 0.02$ and $t = 0.005$ for Chignolin and BBA respectively, which is the same value as presented in Arts et al. (2023) for the same number of training samples.

**Evaluation.** To compute the standard deviation in the experiments, we have used the same models but three different seeds for inference.

## B.5  Dipeptides (2AA)

**Dataset.** The original dipeptide dataset (2AA) was introduced in Klein et al. (2023a) (MIT License) and is available here: `https://huggingface.co/datasets/microsoft/timewarp`. As this includes a lot of intermediate saved states and quantities, like energies, there is a smaller version made available by Klein and Noé (2024) (CC BY 4.0): `https://osf.io/n8vz3/?view_only= 1052300a21bd43c08f700016728aa96e`. For a comprehensive overview of the simulation details, refer to Klein et al. (2023a). All dipeptides were simulated in implicit solvent with a classical *amber-14* force-field at $T = 310$K. The simulation of the training and validation peptides were run

for 50ns, while the test peptides were simulated for $1\mu s$. All simulation were performed with the openMM library (Eastman et al., 2017).

Note that we have removed dipeptides containing Glycine from our dataset to ensure that all dipeptides have the same number of (coarse-grained) atoms. This made it easier to handle it in the code, but it is not a technical limitation of our architecture. It is split into 175 train, 75 validation, and 92 test dipeptides, out of which we have used 15 for the results presented in the paper (also the metrics) to reduce inference time. The fifteen test dipeptides are: AC, AP, AT, ET, HP, HT, IM, KQ, KS, LW, NF, NY, RL, RV, TD.

**Architecture.** The hyperparameters are listed in Table 6. When multiple parameters are listed for the same model, this means that they are used for the corresponding MoE model. Note that when using MoE, we have used smaller networks for larger diffusion times, and only the Fokker–Planck regularized model is conservative. As optimizer, we used AdamW (Loshchilov and Hutter, 2019).

| Parameter | Diffusion | Mixture | Fokker–Planck | Both |
|---|---|---|---|---|
| Epochs | 120 | 100, 20, 10 | 120 | 100, 20, 10 |
| Max Learning Rate | $3 \cdot 10^{-4}$ | $3 \cdot 10^{-4}$ | $3 \cdot 10^{-4}$ | $3 \cdot 10^{-4}$ |
| Min Learning Rate | $10^{-5}$ | $10^{-5}$ | $10^{-5}$ | $10^{-5}$ |
| BS | 1024 | 1024 | 1024 | 1024 |
| Attention Heads | 8 | 8 | 8 | 8 |
| Feature Dim | 16 | 16 | 16 | 16 |
| Model-Ranges | $(0, 1)$ | $(0, 0.1), [0.1, 0.6), [0.6, 1.0)$ | $(0, 1)$ | $(0, 0.1), [0.1, 0.6), [0.6, 1.0)$ |
| Conservative | Yes | Yes, No, No | Yes | Yes, No, No |
| $\alpha$ | 0 | 0, 0, 0 | $5 \cdot 10^{-4}$ | $10^{-4}, 0, 0$ |
| Hidden Dimension | 128 | 128, 96, 96 | 128 | 128, 96, 96 |
| Layers | 3 | 3, 2, 2 | 3 | 3, 2, 2 |

Table 6: **Dipeptides.** This table contains the hyperparameters for the different models shown.

**Simulation.** To perform Langevin simulation, we have extracted the forces from the model via Equation (6) at $t = 10^{-5}$ for all models except for *Two For One*. As this system has not been tested by Arts et al. (2023), we opted to use the same $t$ as for alanine dipeptide, namely $t = 0.02$.

**Evaluation.** To compute the standard deviation in the experiments, we compute the values across all testset dipeptides.

## B.6 Compute Infrastructure

We have used a single RTX 3090 GPU for the toy systems, an A100 with 80GB memory for alanine dipeptide, two A100 80GB GPUs for the dipeptide dataset and Chignolin, and four for BBA. Depending on availability, we have also used H100 for some experiments. For inference, we use a single GPU.

## B.7 Software Licences

In our code, we have used `jax` (Bradbury et al., 2018) (Apache-2.0) and the accompanying machine learning library `flax` (Heek et al., 2024) (Apache-2.0). For the graph transformer architecture, we have extended code from Arts et al. (2023) (MIT) and have re-implemented the code from `https://github.com/lucidrains/graph-transformer-pytorch` (MIT) in `jax`.

For the free-energy plots of the Müller-Brown potential, we used Hoffmann et al. (2021) (LGPL-3.0). For trajectories and simulations, we have used `openMM` (Eastman et al., 2017) (MIT) and `mdtraj` (McGibbon et al., 2015) (LGPL-2.1).

## C Further Studies and More Experiments

### C.1 Comparing Conservative and Score-based Models

Previous work (Arts et al., 2023) suggested that conservative models improve the quality of the diffusion process. However, this effect was not observed for image data (Salimans and Ho, 2021). In Figure 7, we compare these approaches in practice. For iid sampling, conservative models provide a

slight improvement. In contrast, for simulation, we were unable to train stable score-based models without a conservative parameterization, aligning with knowledge from the force-field literature (Schütt et al., 2017; Batzner et al., 2022). Using a conservative model yields much more stable forces, making simulation feasible. We attribute this to the smoother behavior of the conservative parameterization, which prevents sudden changes in the score. Since the impact on iid sampling is negligible, we consider the conservative parameterization most relevant for small timescales, where model training is more sensitive. Consequently, in our MoE architecture, we apply the conservative parameterization only to the small-time diffusion model, achieving comparable or even superior iid sampling performance.

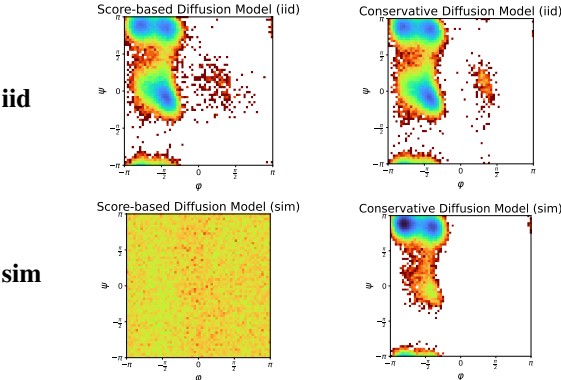

Figure 7: **Alanine dipeptide.** We compare a conservative diffusion model with a score-based model. We can see that around the low-density regions, the conservative parameterization generates better iid samples. As for simulation, a score-based model exhibits stability issues, and the simulation becomes unstable after a few thousand steps.

## C.2 Mixture vs. More Parameters

In our experiments, using the MoE scheme naturally increases the total number of parameters. To assess whether the observed improvements stem from this additional capacity or from the mixture design itself, we perform ablation studies on alanine dipeptide. Specifically, we compare the original results from the paper against three variants: one with wider layers (*Wide*), one with more layers (*Deep*), and one using three instances of the conservative diffusion model combined into a *Mixture* of experts. The results in Table 7 show that *Mixture* outperforms both larger single models and achieves more stable simulation behavior, and requires the least time for training and inference (see also Appendix C.3). This demonstrates that the advantages of the MoE design extend beyond simple parameter scaling and can yield runtime improvements.

| Method | Parameters | Training (↓) | iid JS (↓) | sim. JS (↓) | iid PMF (↓) | sim. PMF (↓) |
|---|---|---|---|---|---|---|
| Diffusion | 647585 | **50min** | **0.0081 ± 0.0003** | 0.0695 ± 0.0517 | 0.095 ± 0.003 | 1.047 ± 0.924 |
| Wide | 1964321 | 67min | 0.0082 ± 0.0003 | 0.0406 ± 0.0236 | 0.096 ± 0.003 | 0.467 ± 0.245 |
| Deep | 1968562 | 111min | **0.0078 ± 0.0003** | 0.0376 ± 0.0087 | **0.091 ± 0.002** | 0.478 ± 0.038 |
| Mixture | 1957567 | **50min** | **0.0079 ± 0.0003** | **0.0264 ± 0.0085** | 0.093 ± 0.007 | **0.325 ± 0.113** |

Table 7: **Alanine dipeptide.** Ablation study comparing the MoE scheme with parameter scaling. Reported are the JS divergence and PMF error for *iid* sampling and simulation (*sim*). Lower values indicate better performance. The MoE achieves the most stable and accurate simulations while maintaining the same training cost as the baseline.

## C.3 Runtime Comparison

We compare the training and inference runtimes of different approaches in Table 8. While the MoE setup could, in principle, be trained in parallel, our current implementation does not distribute the experts across devices and is therefore not fully optimized. For the low-dimensional systems, this

can even introduce a small additional overhead. As a result, runtime improvements are only visible for larger systems. In contrast, models trained with Fokker–Planck regularization show a clear computational overhead due to the additional residual evaluation, which requires multiple forward passes per training step. During inference, however, they perform the same as their counterparts.

| Dataset | Task | Diffusion | Mixture | Fokker–Planck | Both |
|---------|------|-----------|---------|---------------|------|
| Alanine Dipeptide | Training | 50min | 50min | 4h 39min | 3h 59min |
| | iid Sampling | 3min | 4min | 3min | 4min |
| Chignolin | Training | 4h 58min* | 3h 0min | 22h 36min | 21h 33min |
| | iid Sampling | 43min | 14min | 43min | 14min |
| BBA | Training | 1d 20h* | 1d 19h* | 7d 2h | 6d 15h |
| | iid Sampling | 12h 57min | 5h 56min | 12h 57min | 5h 56min |
| Dipeptides | Training | 4h 5min | 3h 50min | 28h 39min | 27h 5min |
| | iid Sampling | 8min | 4min | 8min | 4min |

Table 8: Training and inference runtimes for all models. Methods marked with * were trained using half the number of GPUs (one for Chignolin and two for BBA). The MoE model reduces inference time substantially due to its modular design, while the Fokker–Planck regularization introduces additional computational overhead from multiple forward evaluations during training.

In terms of simulation, the main computational advantage of our approach arises from its ability to perform simulations in parallel and to leverage coarse-graining. In classical molecular dynamics, simulating a protein typically requires modeling the solvent explicitly, which adds thousands of water molecules and substantially increases computational cost. Coarse-graining eliminates these degrees of freedom, thereby accelerating computation and improving parallel scalability.

For instance, in the dipeptide dataset, where each molecule contains only a few atoms and no solvent to coarse-grain, our model achieves 150 parallel simulations at 830 steps/s each, corresponding to roughly 125k steps/s on a single NVIDIA A100. In comparison, a conventional force-field simulation on an NVIDIA V40 reaches around 10k steps/s. Even larger gains can be expected for systems with explicit solvent, where coarse-graining removes many more particles. As an example, our BBA protein simulation (100 parallel runs of 1M steps each) completes in approximately one hour.

### C.4 Toy System

**Dataset.** We have used the Müller-Brown potential (Müller and Brown, 1979) to demonstrate the capabilities of our approach in two dimensions. For this, we have used the following potential

$$
\begin{aligned}
U(x,y) = & - 200 \cdot \exp\left(-(x-1)^2 - 10y^2\right) \\
& - 100 \cdot \exp\left(-x^2 - 10 \cdot (y-0.5)^2\right) \\
& - 170 \cdot \exp\left(-6.5 \cdot (0.5+x)^2 + 11 \cdot (x+0.5) \cdot (y-1.5) - 6.5 \cdot (y-1.5)^2\right) \\
& + \ \ 15 \cdot \exp\left(0.7 \cdot (1+x)^2 + 0.6 \cdot (x+1) \cdot (y-1) + 0.7 \cdot (y-1)^2\right) \ .
\end{aligned}
\tag{34}
$$

To generate training samples from this potential, we have performed a Langevin simulation (compare Equation (5)). For this, we have performed 5M steps with $k_B T = 23, dt = 0.005, M = 0.5 \cdot I$, where we only store every 50th sample to generate 100k training samples.

**Architecture and training.** For the toy systems, we have used a simple multi-layer perception with the hyperparameters presented in Table 9.

| Parameter | Diffusion | Mixture | Fokker–Planck | Both |
|-----------|-----------|---------|---------------|------|
| # Parameters | 17849 | 17263 | 17849 | 17263 |
| BS | 128 | 128 | 128 | 128 |
| Model-Ranges | $(0,1)$ | $(0,0.1), [0.1,0.6), [0.6,1.0)$ | $(0,1)$ | $(0,0.1), [0.1,0.6), [0.6,1.0)$ |
| Epochs | 180 | 120, 30, 30 | 180 | 120, 30, 30 |
| Hidden Dimension | 92 | 64, 64, 54 | 92 | 64, 64, 54 |
| Layers | 3 | 3, 2, 2 | 3 | 3, 2, 2 |
| $\alpha$ | 0 | 0 | $5 \cdot 10^{-4}$ | $5 \cdot 10^{-4}, 0, 0$ |

Table 9: **Müller-Brown.** This table contains the hyperparameters for the different models shown.

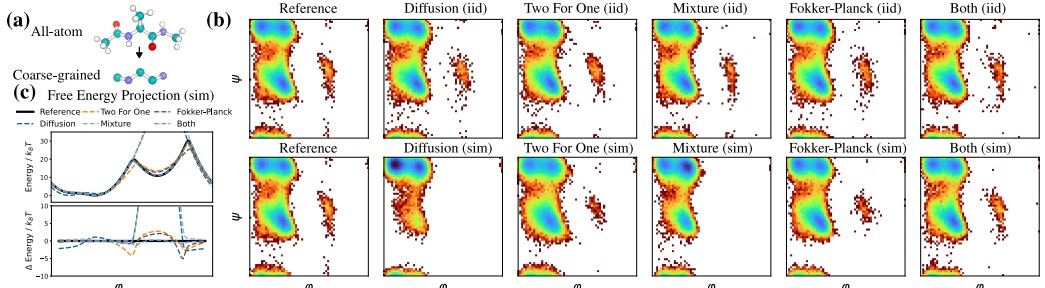

Figure 9: **Alanine dipeptide.** **(a)** The coarse-graining scheme. **(b)** Comparison of the Ramachandran plots of different methods for iid sampling and Langevin simulation. **(c)** The projection of the free energy surface and differences along the dihedral angle $\varphi$ for samples generated with simulation.

**Evaluation.** In Figure 8, we can see the free energy plots of the different methods. All methods produce iid samples that match the *Reference*. However, when using the learned score for Langevin simulation, the standard *Diffusion* model fails to reproduce the correct distribution and undersamples the low-probability state, highlighting the inconsistency between sampling and simulation. Although having roughly the same number of parameters, the *Mixture* model partially improves this, but consistency is only achieved with *Fokker–Planck* regularization. Combining *Both* approaches further improves performance. Numerical results can be seen in Table 10.

| Method | iid JS ($\downarrow$) | sim JS ($\downarrow$) | iid PMF ($\downarrow$) | sim PMF ($\downarrow$) |
|---|---|---|---|---|
| Reference | $0.0119 \pm 0.0004$ | | $0.087 \pm 0.002$ | |
| Diffusion | $\mathbf{0.0122 \pm 0.0013}$ | $0.0448 \pm 0.0125$ | $0.111 \pm 0.006$ | $0.504 \pm 0.150$ |
| Mixture | $\mathbf{0.0109 \pm 0.0007}$ | $0.0254 \pm 0.0109$ | $\mathbf{0.097 \pm 0.004}$ | $0.247 \pm 0.113$ |
| Fokker–Planck | $0.0130 \pm 0.0010$ | $0.0166 \pm 0.0009$ | $0.122 \pm 0.006$ | $0.163 \pm 0.008$ |
| Both | $\mathbf{0.0110 \pm 0.0007}$ | $\mathbf{0.0108 \pm 0.0008}$ | $\mathbf{0.098 \pm 0.003}$ | $\mathbf{0.099 \pm 0.004}$ |

Table 10: **Müller-Brown.** Comparison of methods based on JS Divergence and the PMF error. Lower values are better. To compute the standard deviation, we have trained ten different models and performed sampling/simulation with them. As for the reference, we have started multiple simulations with a different seed on the same ground-truth potential. This serves as a reference of what could optimally be achieved.

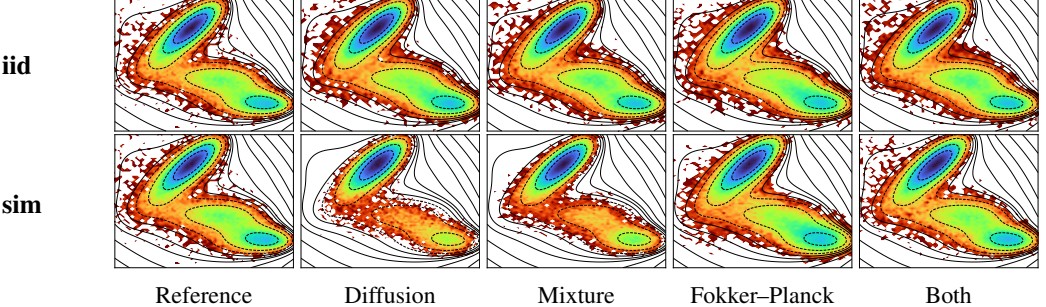

Figure 8: **Müller-Brown.** Comparing free energy plots of different models for classical diffusion sampling (iid) and Langevin simulation (sim). The energies should align with the training data (Reference).

## C.5 Alanine Dipeptide

In this section, we report some further results and plots on alanine dipeptide.

### C.5.1 Free Energies

**Inconsistent sampling.** Figure 9 (b) compares the free energies of the sampled dihedral angles for iid sampling and Langevin simulation (sim). While all methods can match the training distribution under iid sampling, simulation quality varies, and existing models show inconsistencies. Standard *Diffusion* fails to recover the low-probability mode (i.e., $\varphi > 0$) completely, even when starting a simulation from these regions. *Mixture* generally improves the results, but still does not find the other

| Method | iid JS ($\downarrow$) | sim. JS ($\downarrow$) | iid PMF ($\downarrow$) | sim. PMF ($\downarrow$) |
|---|---|---|---|---|
| Diffusion | $\mathbf{0.0081 \pm 0.0003}$ | $0.0695 \pm 0.0517$ | $\mathbf{0.095 \pm 0.003}$ | $1.047 \pm 0.924$ |
| Two For One | $\mathbf{0.0081 \pm 0.0003}$ | $0.0158 \pm 0.0002$ | $\mathbf{0.098 \pm 0.006}$ | $0.206 \pm 0.004$ |
| Mixture | $\mathbf{0.0080 \pm 0.0004}$ | $0.0353 \pm 0.0117$ | $\mathbf{0.092 \pm 0.007}$ | $0.388 \pm 0.109$ |
| Fokker–Planck | $0.0082 \pm 0.0002$ | $0.0090 \pm 0.0006$ | $0.098 \pm 0.003$ | $0.104 \pm 0.004$ |
| Both | $0.0082 \pm 0.0004$ | $\mathbf{0.0080 \pm 0.0002}$ | $\mathbf{0.097 \pm 0.008}$ | $\mathbf{0.091 \pm 0.004}$ |

Table 11: **Alanine dipeptide.** Comparison of JS divergence and PMF error. To compute the mean and the standard deviation, we have trained and evaluated three models with three different seeds.

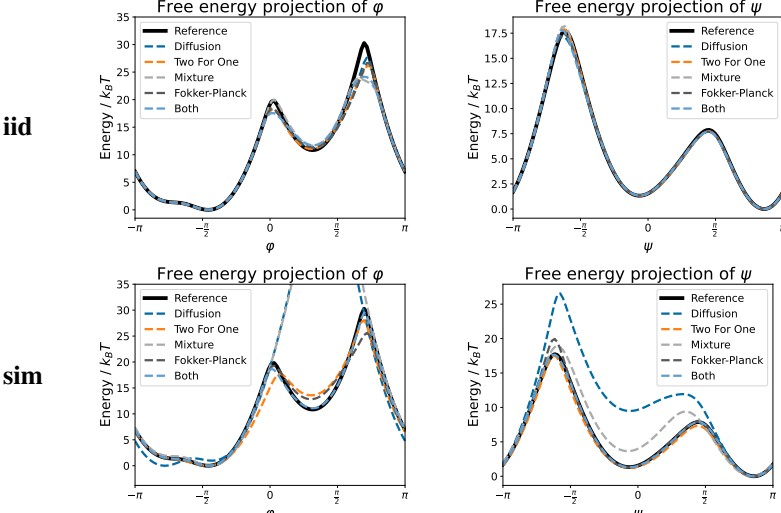

Figure 10: **Alanine dipeptide.** Comparing the free energy of along the dihedral angles $\varphi, \psi$ for iid sampling and simulation across different models.

mode. This is reflected in the numerical results in Table 11, where *Mixture* achieves lower means and smaller variance, but simulation errors remain noticeable. We attribute this behavior to the smaller time range, which focuses the model's attention, allowing it to learn a better, more stable optimum.

**Consistent models.** *Fokker–Planck* regularization enables the model to recover the missing states without modifying the diffusion time, and thus preserving structural accuracy. Table 11 shows that the regularization substantially improves consistency between iid and simulation, although iid performance slightly declines in favor of improved simulation performance. Combining MoE with Fokker–Planck regularization for *Both*, enhances simulation quality further while barely mitigating the drop in iid performance. The resulting model achieves close alignment between iid and simulation, and captures the free energy landscape in simulations accurately (see Figure 9 (c)). The similar iid performance of *Both* and *Fokker–Planck* suggests that applying regularization introduces additional constraints and restrictions on the model and hence can degrade generative quality.

**Detailed free energy.** In Figure 10 we compare the free energies along the dihedral angles $\varphi, \psi$ for iid sampling and simulation. We can see that the results from the main paper persist.

### C.5.2 Fokker–Planck Residual

In Figure 11 the Fokker–Planck residual error $\left\| \mathcal{F}_{p^\theta}(\boldsymbol{x}, t) - \partial_t \log p_t^{\boldsymbol{\theta}}(\boldsymbol{x}) \right\|_2$ is reported. Overall, the results are similar to what was reported in Figure 6. We can note that the Fokker–Planck error of *Mixture* is lower than *Diffusion*, indicating that MoE can improve the model's consistency.

### C.5.3 Bonds

Figure 12 shows the immediate bond lengths of the coarse-grained molecule for iid sampling and Langevin simulation. Since *Two For One* does not evaluate the model at $t = 0$, it introduces noise across all bonds. We can also see this behavior by looking at the Wasserstein distance of the bond-lengths to the reference data as seen in Table 12. In this table, *Mixture* provides the best bonds during simulation, but does not recover all modes.

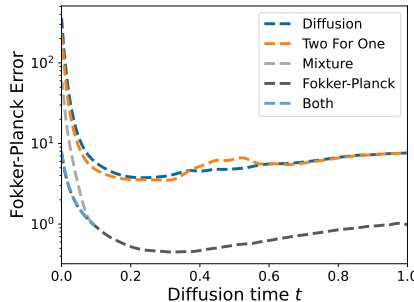

Figure 11: **Alanine dipeptide.** Comparing the Fokker–Planck error for $\log p^{\boldsymbol{\theta}}$ of multiple models.

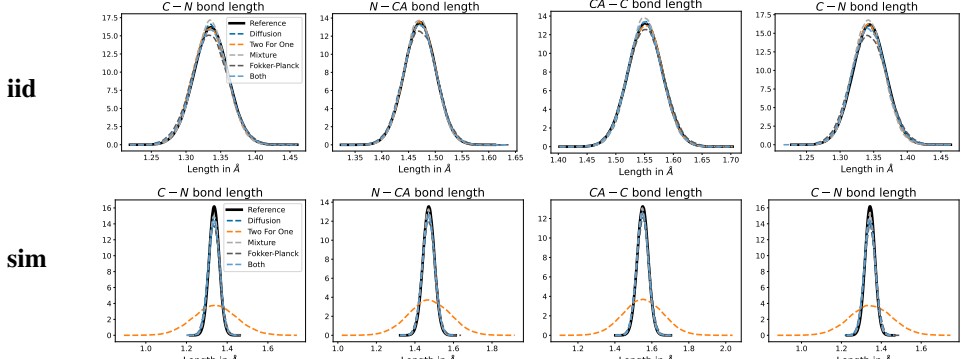

Figure 12: **Alanine dipeptide.** This illustration shows all direct bond lengths sampled by the different methods.

| Method | iid relative W1 ($\downarrow$) | sim relative W1 ($\downarrow$) |
|---|---|---|
| Diffusion | **0.88 ± 0.70** | 0.83 ± 0.15 |
| Two For One | **0.88 ± 0.58** | 23.51 ± 1.91 |
| Mixture | **1.13 ± 0.46** | **0.47 ± 0.12** |
| Fokker–Planck | 1.65 ± 0.37 | 1.23 ± 0.16 |
| Both | **1.00 ± 0.00** | 1.00 ± 0.00 |

Table 12: Comparison of methods on alanine dipeptide based on the Wasserstein 1 distance of the C-N bond lengths to the reference data. We have divided all entries by the Wasserstein 1 distance of *Both* so that the numbers are easier to compare. In other words, numbers larger than 1 mean that the bonds are worse than *Both*.

### C.5.4 Force Matching

We further compare our model against prior work on force matching, which either employs explicit forces from the training set (Husic et al., 2020) or infers forces implicitly from noise in the data (Durumeric et al., 2024; Klein et al., 2025). All these approaches share the same underlying architecture, the CGSchNet model (Husic et al., 2020), which combines a trainable SchNet component (Schütt et al., 2017) with fixed energy terms accounting for bonded interactions such as bonds, angles, and dihedrals. In contrast, our models operate without any such energy priors, and thus require significantly less prior knowledge about the investigated systems. As these previous studies commonly evaluated their models on a six-bead representation of alanine dipeptide (Wang et al., 2019; Husic et al., 2020; Klein et al., 2025), we retrain our models as well as the *Two for One* model on the same coarse-grained system. The corresponding training and evaluation trajectory was generated via MD for $1\,\mu$s in explicit solvent (Wang et al., 2019), and the training set for all methods consists of 100k samples. For the force-matching baselines, we report the results from Klein et al. (2025). The comparison, shown in Table 13, evaluates again the PMF between the simulated and target trajectories. We only evaluate the simulations, as the force-matching methods do not permit independent sampling. Overall, our models achieve superior performance compared to all force matching baselines on this dataset, despite not using any force information, unlike Husic et al. (2020).

| Method | sim. JS ($\downarrow$) | sim PMF ($\downarrow$) |
|---|---|---|
| Force Matching | $0.0243 \pm 0.0015$ | $0.322 \pm 0.005$ |
| Noise and Force Matching | $0.0402 \pm 0.0066$ | $0.864 \pm 0.146$ |
| Operator Forces | $0.0214 \pm 0.0041$ | $0.301 \pm 0.043$ |
| Operator Forces | $0.0166 \pm 0.0009$ | $0.282 \pm 0.031$ |
| Diffusion | $0.0928 \pm 0.0165$ | $1.135 \pm 0.205$ |
| Two for One | $0.0151 \pm 0.0006$ | $0.168 \pm 0.009$ |
| Mixture | $0.0506 \pm 0.0370$ | $0.545 \pm 0.429$ |
| Fokker–Planck | $\mathbf{0.0092 \pm 0.0017}$ | $\mathbf{0.107 \pm 0.028}$ |
| Both | $\mathbf{0.0075 \pm 0.0024}$ | $\mathbf{0.090 \pm 0.049}$ |

Table 13: **Six-bead alanine dipeptide.** Comparison of different models for learning forces. The table reports the JS divergence and the PMF error between simulated and target distributions (lower is better). Our Fokker–Planck-regularized models outperform all force-matching methods despite being trained without any force labels.

## C.6 Proteins

We now provide additional comparisons and ablation studies for the independent contributions of MoE and Fokker-Planck regularization for Chignolin and BBA as well as analysis of dynamics.

### C.6.1 Free Energy Surfaces

We present the same free energy surfaces as in the main text, but now distinguish between *Mixture*, *Fokker–Planck*, and *Both* (referred to as *Ours* in the main text). In Figure 13, we show the free energy along the first two TIC coordinates by projecting the data into 100 bins, and in Figure 14, we show the corresponding kernel density estimates along each axis. Beyond the results discussed in the main text, we observe that simulations using *Diffusion* (i.e., evaluating the model at $t = 0$) are unstable. The MoE variant in *Mixture* improves stability but does not fully resolve these issues. In contrast, introducing the Fokker–Planck regularization (with or without mixture) yields stable simulations, and the resulting free energy surfaces from diffusion sampling and simulation become consistent. In Figure 14, the models appear nearly identical for *iid* sampling but differ for *sim*. The qualitative metrics are similar between Fokker–Planck with and without MoE, showing only minor differences.

The same trends are observed for BBA in Figures 15 and 16. The *Diffusion* model fails to recover a meaningful free energy landscape, while *Mixture* using MoE substantially improves the results but remains insufficient. Once the Fokker–Planck regularization is applied, the simulations become consistent with diffusion sampling. As shown in Figure 16, the *iid* results of several models are nearly indistinguishable in the one-dimensional projections, making it difficult to determine which performs best. For *sim*, it is also unclear whether *Fokker–Planck* or *Both* yields better performance. A quantitative comparison is provided in Appendix C.6.3.

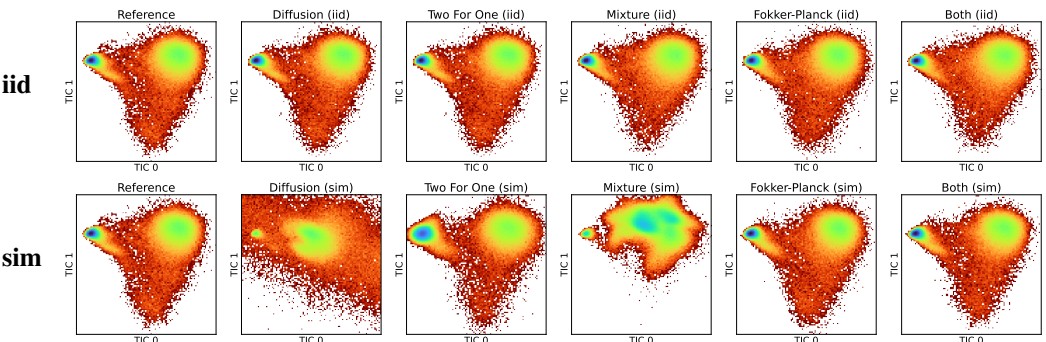

Figure 13: **Chignolin.** We compare the free energy projected onto the first two TIC axes for all presented methods for *iid* sampling and Langevin simulation (*sim*). The aim is to match the *Reference* (left) and have a self-consistent model where *iid* and *sim* closely aligns.

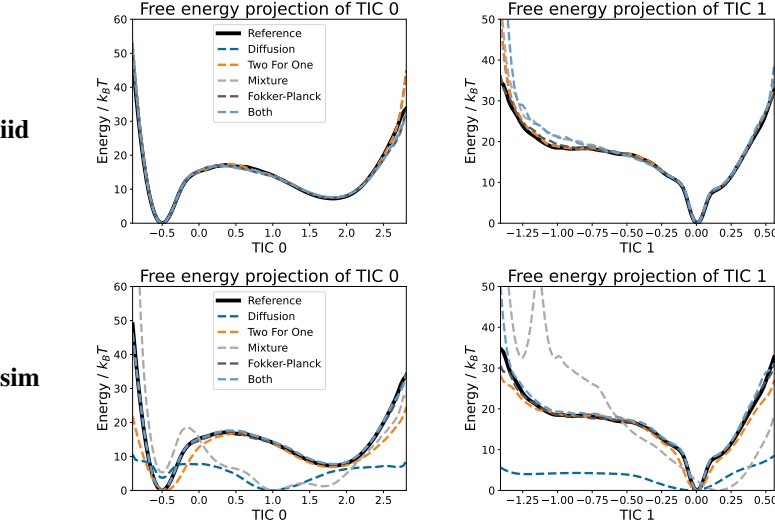

Figure 14: **Chignolin.** Comparing the free energy along the first two TIC axes for iid sampling and simulation across different models with the reference in black.

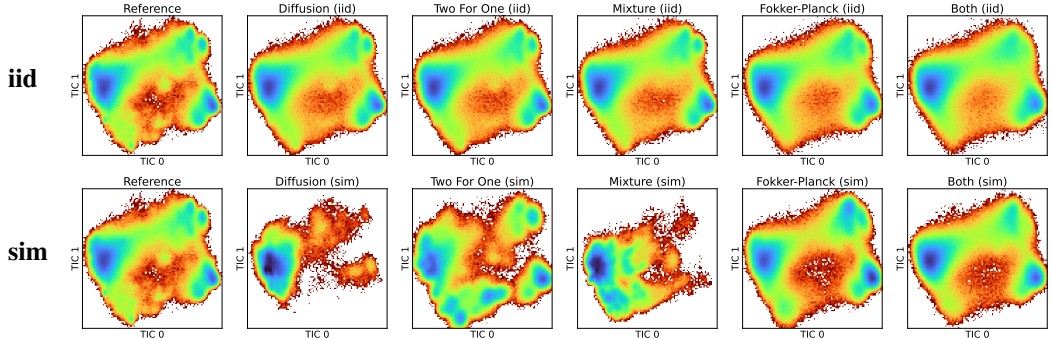

Figure 15: **BBA.** We compare the free energy projected onto the first two TIC axes for all presented methods for *iid* sampling and Langevin simulation (*sim*). The aim is to match the *Reference* (left) and have a self-consistent model where *iid* and *sim* closely aligns.

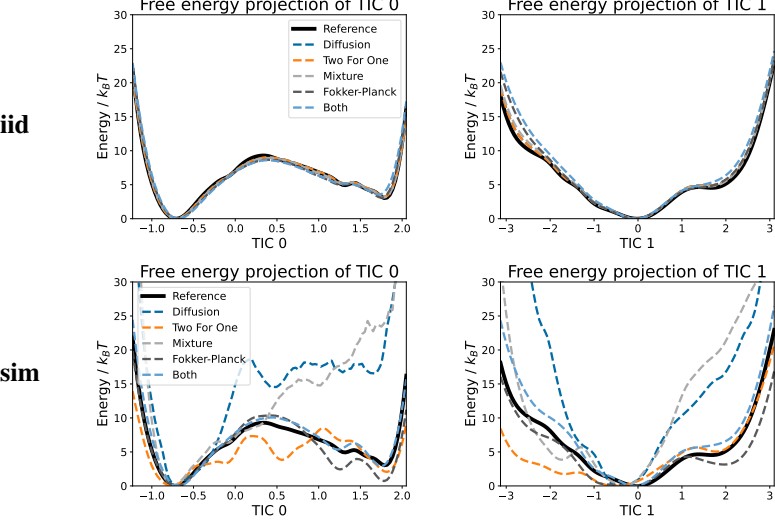

Figure 16: **BBA.** Comparing the free energy along the first two TIC axes for iid sampling and simulation across different models with the reference in black.

### C.6.2 Contact Maps

To analyze structural consistency beyond local distances, we compute pairwise contact maps between all atoms. As such, they provide a compact way of representing molecular structure and indicate which atom pairs are in spatial proximity. Specifically, for each generated ensemble, we compute the distances between all atom pairs and record whether they fall below a threshold of 10 Å. Averaging these contacts over all generated samples yields a probability map indicating how frequently each atom pair interacts. We then take the log of these to smooth out the range and improve visualization.

This is visualized for Chignolin and BBA in Figures 17 and 18. This representation captures the overall spatial organization of a molecule, such as which regions are folded together or form stable secondary structures. Comparing contact maps from different models provides an intuitive way to assess whether the generated ensembles reproduce realistic atomic arrangements.

These figures align with the overall trends in the paper showing that: *iid* works well across all models, *Diffusion* and *Mixture* is unstable for simulation, and *Fokker–Planck/Both* match the reference and the sampling methods are consistent. However, when examining these specific metrics, the additional noise introduced by *Two For One* is less apparent. We attribute this to the binary cutoff used to determine whether two atoms are in contact, which reduces sensitivity to small structural perturbations. This effect is not present in the pairwise distance matrix analysis in Tables 14 and 15.

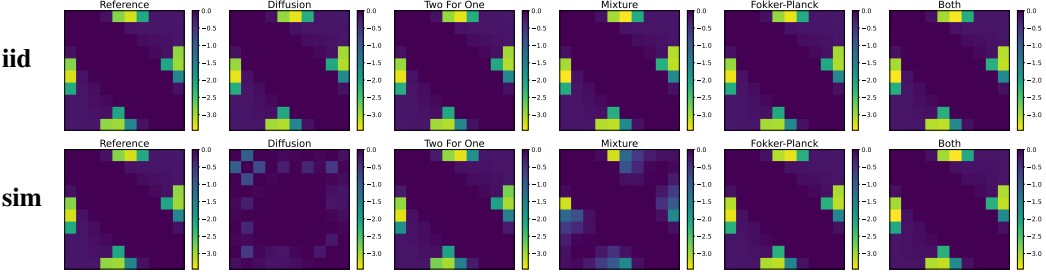

Figure 17: **Chignolin.** Pairwise contact maps comparing different models and the two sampling methods. Two atoms are considered in contact if their distance is below 10 Å. Bright colors (corresponding to small values) indicate a low probability of close proximity. The maps are symmetric by construction.

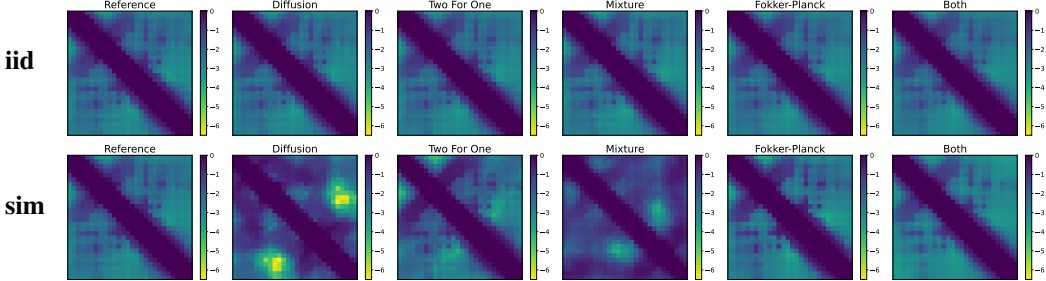

Figure 18: **BBA.** Pairwise contact maps comparing different models and the two sampling methods. Two atoms are considered in contact if their distance is below 10 Å. Bright colors (corresponding to small values) indicate a low probability of close proximity. The maps are symmetric by construction.

### C.6.3 Quantiative Results

In addition to the results presented in the main paper, we provide further quantitative analyses in this section. Alongside the PMF error, we also report the JS divergence. While the PMF error evaluates energy differences, the JS divergence compares probability densities, which can make model differences less pronounced. Both metrics are computed in 2D TICA space.

We additionally report the JS divergence computed on the pairwise distance (PWD) matrix. For this, we first calculate all interatomic distances and extract the upper triangular part of the full PWD matrix to avoid redundancy. We then remove the diagonal elements and exclude pairs separated by fewer than three bonds to focus on nonlocal structural correlations. In other words, we compare distances between atoms that are not close neighbors along the chain. For each atom pair, we then construct normalized histograms of their distance distributions for both the generated and reference ensembles and compute the JS divergence between them. Averaging these divergences across all pairs yields a

single scalar value that quantifies how well the generated ensemble reproduces the overall spatial organization of the molecule. Lower values correspond to higher structural fidelity.

The resulting metrics are summarized in Tables 14 and 15 for Chignolin and BBA, respectively. The trends are consistent with those discussed in the main text: for Chignolin, we observe only a slight decrease in *iid* sampling quality when introducing Fokker–Planck regularization or the MoE model. However, for the larger and more complex BBA system, it becomes evident that achieving a consistent energy-based model requires trade-offs in *iid* sampling quality. In contrast, our models excel in the *sim* setting, achieving the best results in both energetic and structural metrics, particularly for the PWD-based measure of structural fidelity.

| Method | iid JS ($\downarrow$) | sim. JS ($\downarrow$) | iid PMF ($\downarrow$) | sim. PMF ($\downarrow$) | iid PWD JS ($\downarrow$) | sim. PWD JS ($\downarrow$) |
|---|---|---|---|---|---|---|
| Diffusion | **0.0036 ± 0.0001** | 0.4351 ± 0.0141 | **0.027 ± 0.000** | 63.804 ± 0.372 | **0.0001 ± 0.0000** | 0.3817 ± 0.0009 |
| Two For One | **0.0036 ± 0.0001** | 0.1023 ± 0.0008 | **0.027 ± 0.000** | 1.438 ± 0.019 | **0.0001 ± 0.0000** | 0.0082 ± 0.0000 |
| Mixture | 0.0042 ± 0.0001 | 0.4336 ± 0.0075 | 0.033 ± 0.000 | 11.185 ± 0.430 | 0.0003 ± 0.0000 | 0.2045 ± 0.0004 |
| Fokker–Planck | 0.0048 ± 0.0001 | **0.0050 ± 0.0001** | 0.037 ± 0.000 | **0.039 ± 0.001** | 0.0004 ± 0.0000 | **0.0008 ± 0.0000** |
| Both | 0.0045 ± 0.0001 | **0.0050 ± 0.0008** | 0.035 ± 0.001 | **0.038 ± 0.006** | 0.0003 ± 0.0000 | **0.0012 ± 0.0005** |

Table 14: **Chignolin.** Quantitative comparison of all methods based on JS divergence, PMF error (both computed in TICA space), and the mean JS divergence of the pairwise distance (PWD) distributions of atoms. Lower values indicate better agreement with the reference data. The *iid* metrics assess equilibrium sampling quality, while the *sim* metrics evaluate the consistency of the simulated dynamics.

| Method | iid JS ($\downarrow$) | sim. JS ($\downarrow$) | iid PMF ($\downarrow$) | sim. PMF ($\downarrow$) | iid PWD JS ($\downarrow$) | sim. PWD JS ($\downarrow$) |
|---|---|---|---|---|---|---|
| Diffusion | **0.0043 ± 0.0000** | 0.2014 ± 0.0043 | **0.034 ± 0.000** | 5.387 ± 0.144 | **0.0006 ± 0.0000** | 0.1003 ± 0.0050 |
| Two For One | **0.0043 ± 0.0000** | 0.1162 ± 0.0021 | **0.034 ± 0.000** | 1.624 ± 0.107 | **0.0006 ± 0.0000** | 0.0240 ± 0.0008 |
| Mixture | 0.0091 ± 0.0000 | 0.1964 ± 0.0113 | 0.081 ± 0.000 | 5.970 ± 0.278 | 0.0012 ± 0.0000 | 0.1016 ± 0.0020 |
| Fokker–Planck | 0.0130 ± 0.0000 | 0.0292 ± 0.0035 | 0.117 ± 0.001 | **0.275 ± 0.033** | 0.0018 ± 0.0000 | 0.0078 ± 0.0014 |
| Both | 0.0238 ± 0.0001 | **0.0238 ± 0.0001** | 0.234 ± 0.001 | **0.254 ± 0.005** | 0.0023 ± 0.0000 | **0.0042 ± 0.0001** |

Table 15: **BBA.** Quantitative comparison of all methods based on JS divergence, PMF error (both computed in TICA space), and the mean JS divergence of the pairwise distance (PWD) distributions of atoms. Lower values indicate better agreement with the reference data. The *iid* metrics assess equilibrium sampling quality, while the *sim* metrics evaluate the consistency of the simulated dynamics.

### C.6.4 Dynamics

The main advantage of our method is that we have access to a *consistent* and physically meaningful energy function that allows us to perform simulation and recover dynamic properties. To illustrate this benefit, we analyze the recovered dynamics and provide further evaluations.

As in the main text for BBA, we visualize in Figure 19 a single simulated trajectory for Chignolin. The system exhibits folding and unfolding transitions between metastable states, showing realistic conformational changes over time. For this, we employ the *Both* model, which combines Fokker–Planck regularization with the MoE scheme.

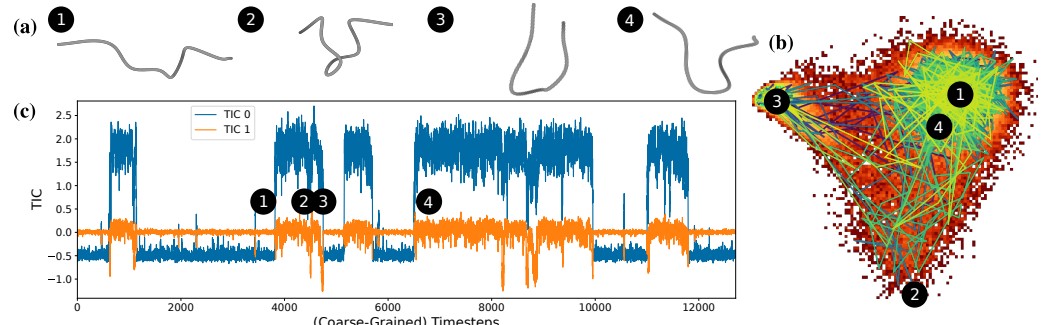

Figure 19: **Chignolin.** Recovering and analyzing the dynamic behavior. **(a)** 3D molecular structures from a single MD trajectory generated by our model. **(b)** Trajectory projected onto the first two TIC coordinates. **(c)** Illustrating the TIC coordinates over the coarse-grained (and subsampled) timesteps.

To further assess the long-time MD simulations, we follow the work of Arts et al. (2023) and cluster the conformations into three metastable states for Chignolin and four for BBA. State assignments are obtained via *k*-means clustering on the TIC coordinates, from which we estimate a Markov model and compute the transition probabilities between states. The state definitions and transition matrices are visualized in Figures 20 and 21. Again, this analysis is only applicable to models that recover continuous dynamics rather than independent samples.

Although the visual differences between transition matrices are subtle, the Fokker–Planck–regularized models clearly provide the most accurate estimates of transition probabilities when compared to the reference simulation. To quantify this, we compute the mean JS divergence between the transition probabilities of each model and the reference, reported in Table 16. These results confirm that Fokker–Planck regularization improves kinetic consistency, while the MoE scheme leads to slightly less accurate transitions; both when used independently and in combination with Fokker–Planck.

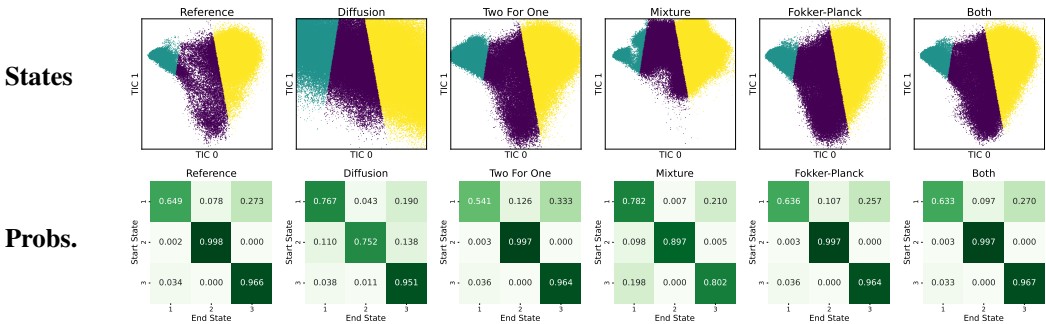

Figure 20: **Chignolin** Each conformation is assigned to one of three states based on the TIC coordinates (top row), shown as State 1 (purple), State 2 (green), and State 3 (yellow). Using these assignments, we estimate the corresponding transition probabilities between states.

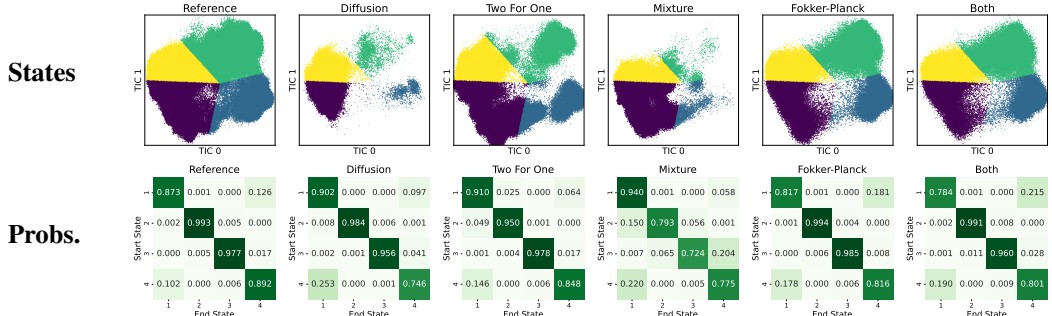

Figure 21: **BBA.** Each conformation is assigned to one of four states based on the TIC coordinates (top row), shown as State 1 (purple), State 2 (blue), State 3 (green), State 4 (yellow). Using these assignments, we estimate the corresponding transition probabilities between states.

| Method | Chignolin JS ($\downarrow$) | BBA JS ($\downarrow$) |
|---|---|---|
| Diffusion | $3.4 \cdot 10^{-2}$ | $6.8 \cdot 10^{-3}$ |
| Two for One | $2.3 \cdot 10^{-3}$ | $7.1 \cdot 10^{-3}$ |
| Mixture | $3.0 \cdot 10^{-2}$ | $4.0 \cdot 10^{-2}$ |
| Fokker–Planck | $4.6 \cdot 10^{-4}$ | $\mathbf{2.5 \cdot 10^{-3}}$ |
| Both | $\mathbf{2.1 \cdot 10^{-4}}$ | $4.2 \cdot 10^{-3}$ |

Table 16: Mean JS divergence between the transition probabilities estimated from simulation and those of the reference trajectories (compare Figures 20 and 21). Lower values indicate better agreement.

## C.7 Umbrella Sampling

Similarly to Appendix C.6.4, access to an energy-based diffusion model enables further applications such as molecular simulation and *umbrella sampling* (Torrie and Valleau, 1977). In umbrella sampling, several simulations are carried out with restraints applied to specific regions of configuration space (e.g., along selected dihedral angles), and the resulting biased ensembles are reweighted by the bias potential to recover unbiased free energy estimates.

While such enhanced sampling techniques lie beyond the main scope of this work, we conducted preliminary experiments with our model. We found that, in its current form, umbrella sampling remains challenging. The model tends to perform poorly when forced into high-energy or previously unseen regions of configuration space. For example, in Chignolin, umbrella sampling along the first TIC coordinate failed to adequately explore the extremes of the coordinate, leading to unreliable free-energy estimates in those regions. Nonetheless, the overall shape of the free energy profile could still be recovered with significantly fewer simulation steps. We believe that the proposed Fokker–Planck regularization could help address this limitation. Since the loss can be evaluated for arbitrary configurations $x$, it may encourage the model to generalize better to out-of-distribution or noisier samples, thereby improving its robustness in biased or enhanced sampling settings.

## C.8 Transferability Across Dipeptides (two amino acids)

### C.8.1 Complete Metrics

In the main text, we reported the PMF error across the test set, and in Table 17, we additionally include the JS divergence for the same experiments. Both metrics show consistent trends, although JS divergence is less sensitive to small differences, as it measures discrepancies between densities rather than energies.

| Method | iid JS ($\downarrow$) | sim. JS ($\downarrow$) | iid PMF ($\downarrow$) | sim. PMF ($\downarrow$) |
|---|---|---|---|---|
| Transferable BG | **0.0183 ± 0.0070** | - | **0.230 ± 0.119** | - |
| Diffusion | **0.0155 ± 0.0083** | 0.2256 ± 0.1304 | **0.206 ± 0.159** | 6.515 ± 3.175 |
| Two For One | **0.0153 ± 0.0080** | 0.0466 ± 0.0114 | **0.203 ± 0.149** | 0.741 ± 0.319 |
| Mixture | **0.0155 ± 0.0078** | 0.0444 ± 0.0237 | **0.200 ± 0.127** | 0.658 ± 0.407 |
| Fokker–Planck | **0.0200 ± 0.0071** | 0.0254 ± 0.0119 | **0.241 ± 0.105** | **0.368 ± 0.267** |
| Both | **0.0157 ± 0.0078** | **0.0177 ± 0.0084** | **0.199 ± 0.127** | **0.203 ± 0.104** |

Table 17: **Dipeptides.** Comparison of metrics across all testset dipeptides with JS divergence and PMF error. To compute the mean and standard deviation, we have averaged the metrics across the dipeptides from the test set.

### C.8.2 Lysine-Serine (KS)

In Figure 22, we present extended results for the dipeptide analyzed in the main text (KS). The trends remain consistent with previous findings: using *Two For One* for simulation leads to incorrect observables and distorted equilibrium distributions. In addition, we show the free energy surfaces along the dihedral angles $\varphi$ and $\psi$ separately. The Fokker–Planck regularization improves these free energy profiles, and the *Both* model produces particularly accurate and well-aligned surfaces.

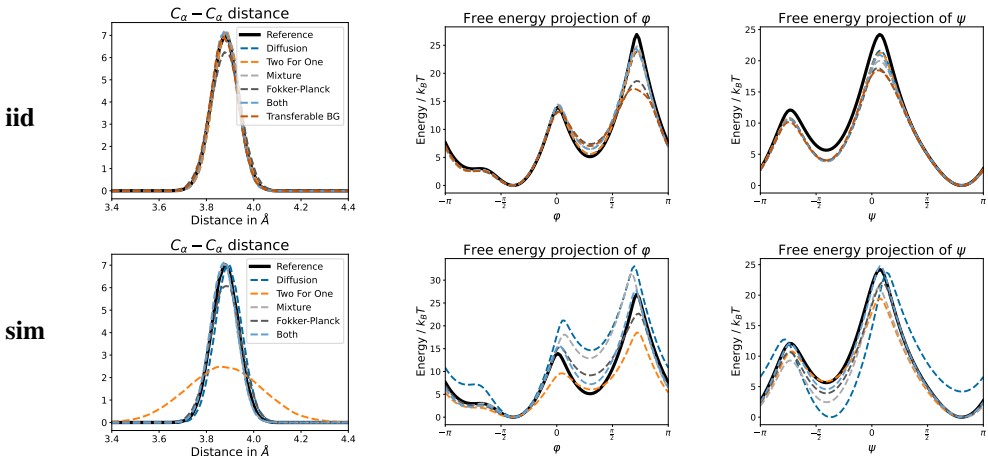

Figure 22: **KS:** We compare further metrics between iid sampling and Langevin simulation. We compare the $C_\alpha$–$C_\alpha$ distance for the dipeptides and also the free energy projections along the dihedral angles $\varphi, \psi$.

### C.8.3 Transferability: Results on More Dipeptides

In this section, we show additional dipeptides from the test set to evaluate the performance of all models and their combinations. While individual systems exhibit minor variations, the overall trends remain consistent. We present the following dipeptides: AC Figures 23 and 24, AP Figures 25 and 26, ET Figures 27 and 28, HP Figures 29 and 30, NY Figures 31 and 32, RV Figures 33 and 34, and TD Figures 35 and 36.

Notably, in Figures 25, 29 and 35, the *Diffusion* model fails to produce stable simulations, resulting in divergence and visible artifacts. This instability appears across several systems, not only those containing proline. While we found in our experiments that introducing dropout can help to stabilize the simulation, we chose not to include it since all other methods remain stable without it. In particular, these cases highlight that restricting training to a smaller diffusion-time subrange, as is done with *Mixture*, improves simulation stability.

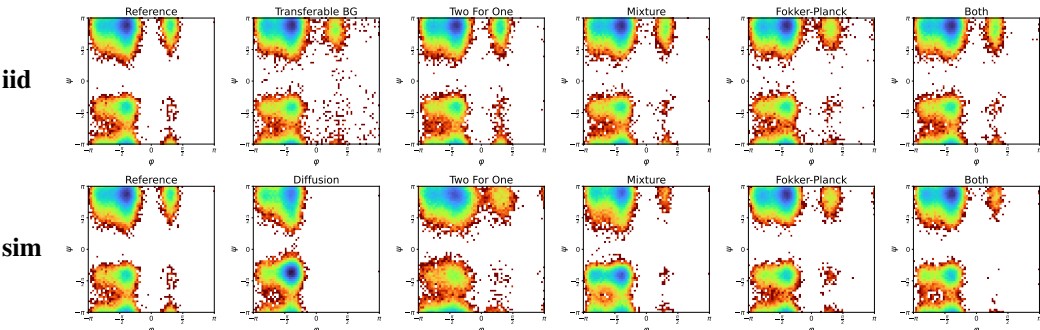

Figure 23: **AC:** We compare the free energy plot on the dihedral angles $\varphi, \psi$ for all presented methods for iid sampling and Langevin simulation.

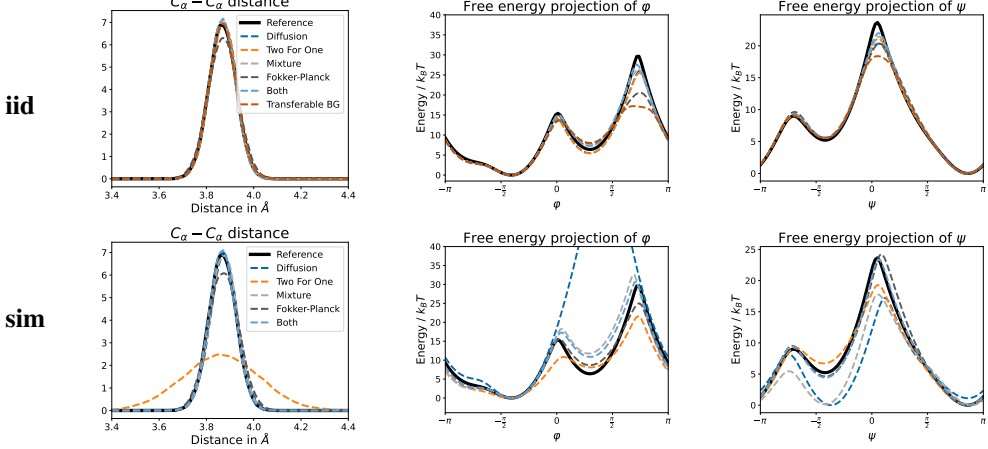

Figure 24: **AC:** We compare further metrics between iid sampling and Langevin simulation. We compare the $C_\alpha$–$C_\alpha$ distance for the dipeptides and also the free energy projections along the dihedral angles $\varphi, \psi$.

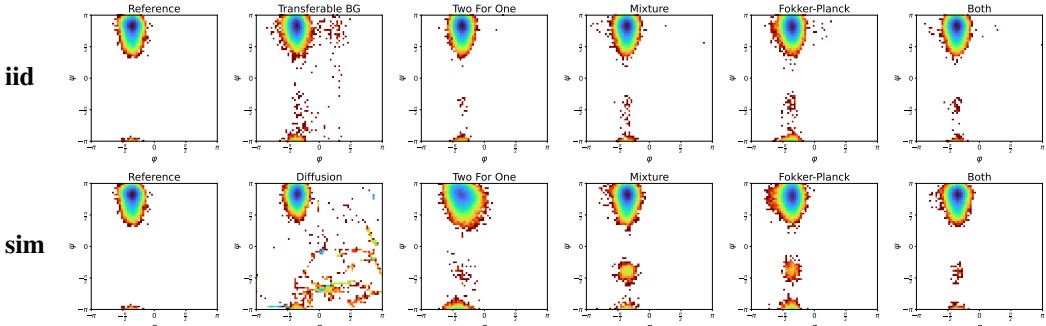

Figure 25: **AP:** We compare the free energy plot on the dihedral angles $\varphi, \psi$ for all presented methods for iid sampling and Langevin simulation.

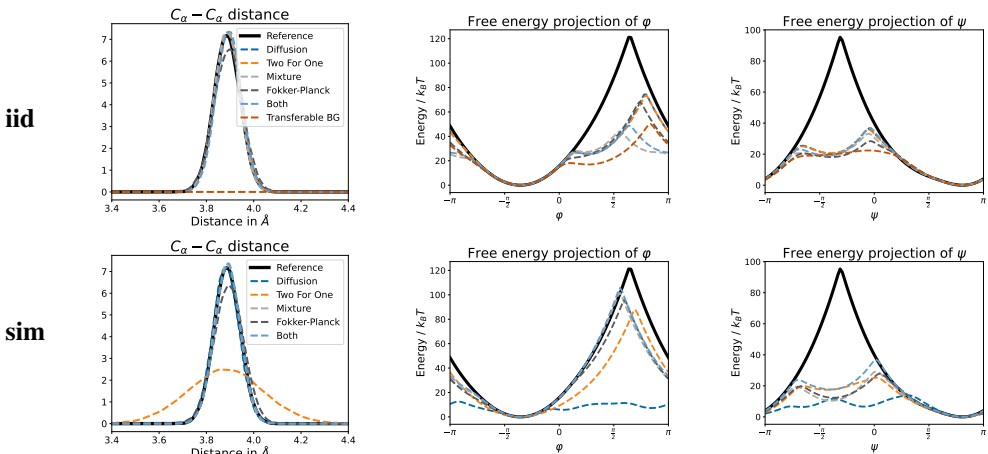

Figure 26: **AP:** We compare further metrics between iid sampling and Langevin simulation. We compare the $C_\alpha$–$C_\alpha$ distance for the dipeptides and also the free energy projections along the dihedral angles $\varphi, \psi$.

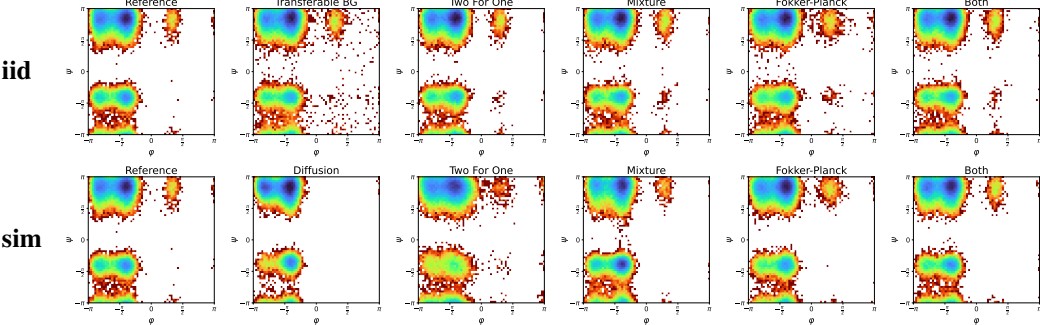

Figure 27: **ET:** We compare the free energy plot on the dihedral angles $\varphi, \psi$ for all presented methods for iid sampling and Langevin simulation.

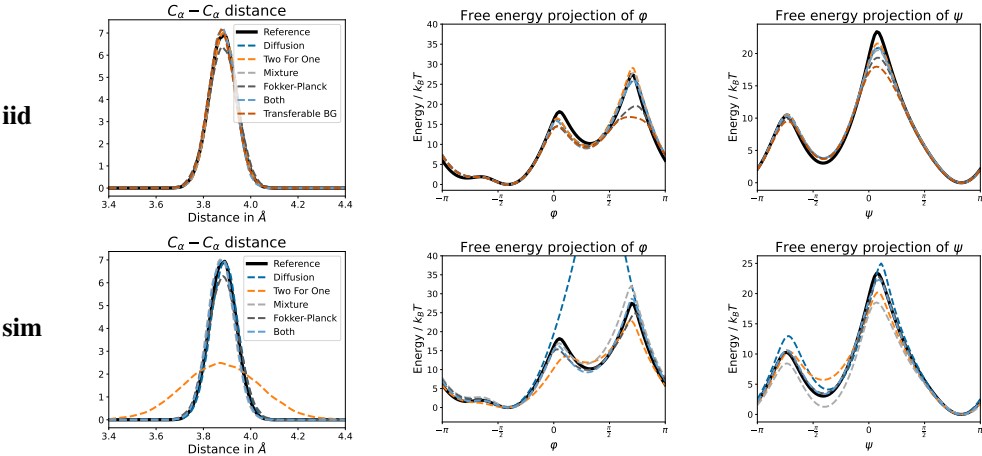

Figure 28: **ET:** We compare further metrics between iid sampling and Langevin simulation. We compare the $C_\alpha$–$C_\alpha$ distance for the dipeptides and also the free energy projections along the dihedral angles $\varphi, \psi$.

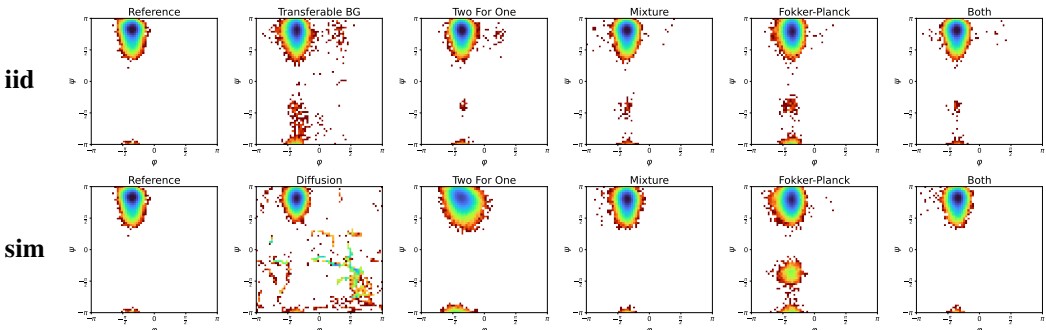

Figure 29: **HP:** We compare the free energy plot on the dihedral angles $\varphi, \psi$ for all presented methods for iid sampling and Langevin simulation.

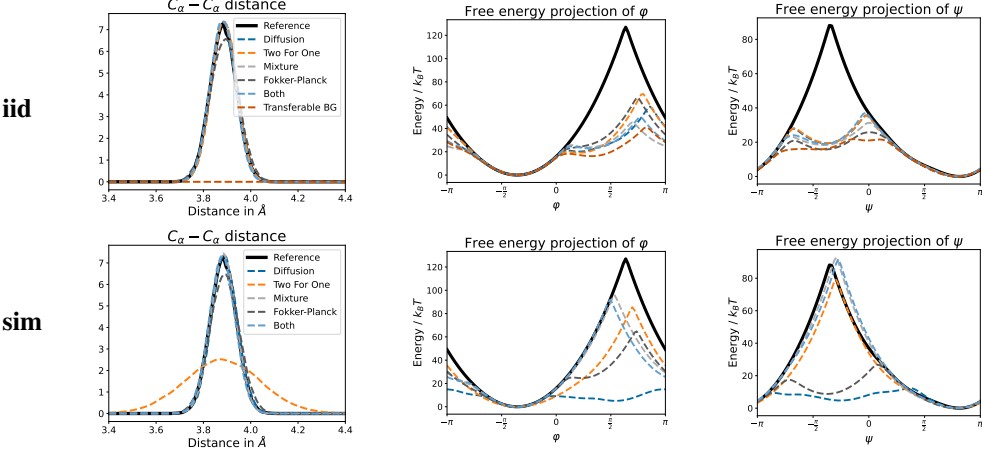

Figure 30: **HP:** We compare further metrics between iid sampling and Langevin simulation. We compare the $C_\alpha$–$C_\alpha$ distance for the dipeptides and also the free energy projections along the dihedral angles $\varphi, \psi$.

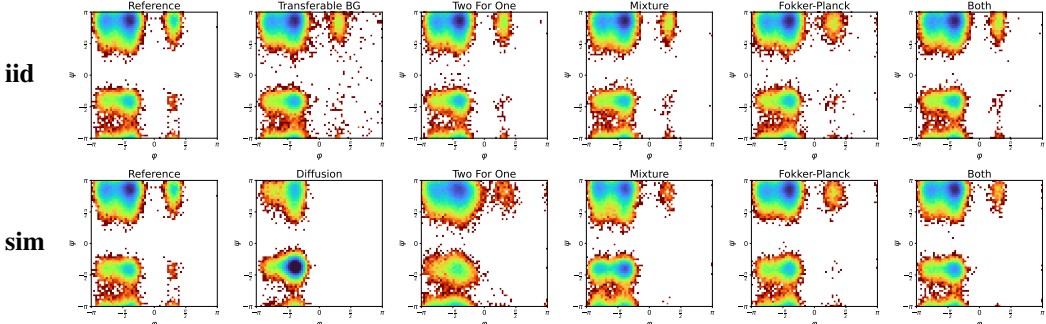

Figure 31: **NY:** We compare the free energy plot on the dihedral angles $\varphi, \psi$ for all presented methods for iid sampling and Langevin simulation.

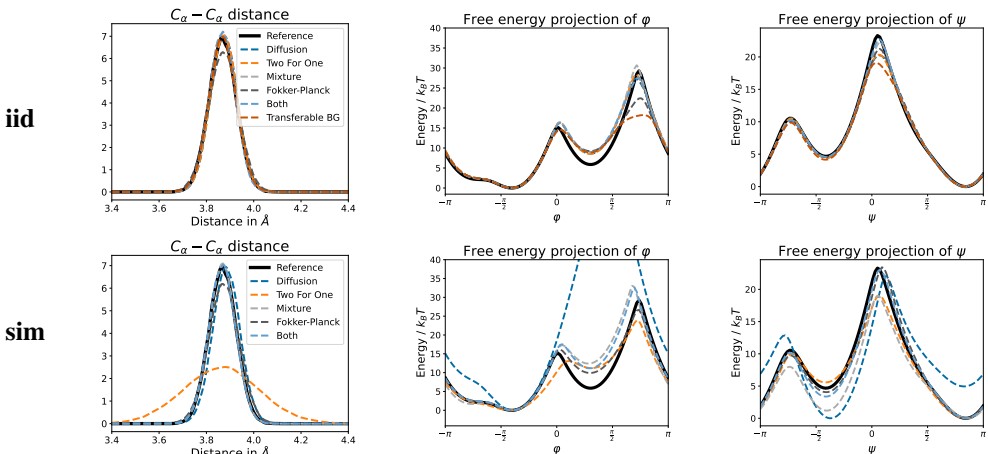

Figure 32: **NY:** We compare further metrics between iid sampling and Langevin simulation. We compare the $C_\alpha$–$C_\alpha$ distance for the dipeptides and also the free energy projections along the dihedral angles $\varphi, \psi$.

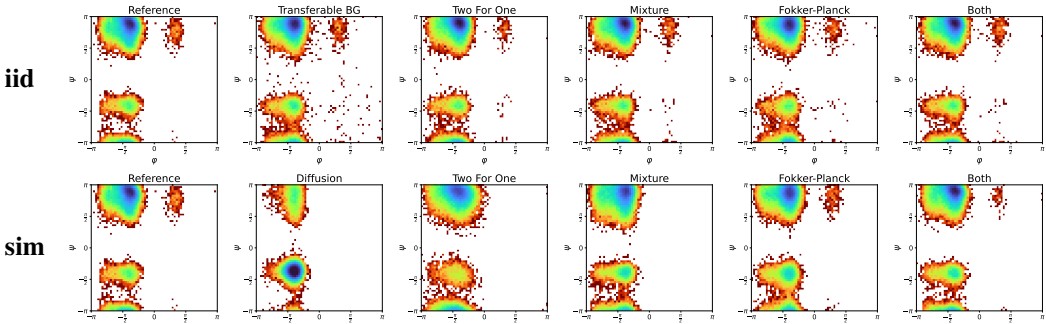

Figure 33: **RV:** We compare the free energy plot on the dihedral angles $\varphi, \psi$ for all presented methods for iid sampling and Langevin simulation.

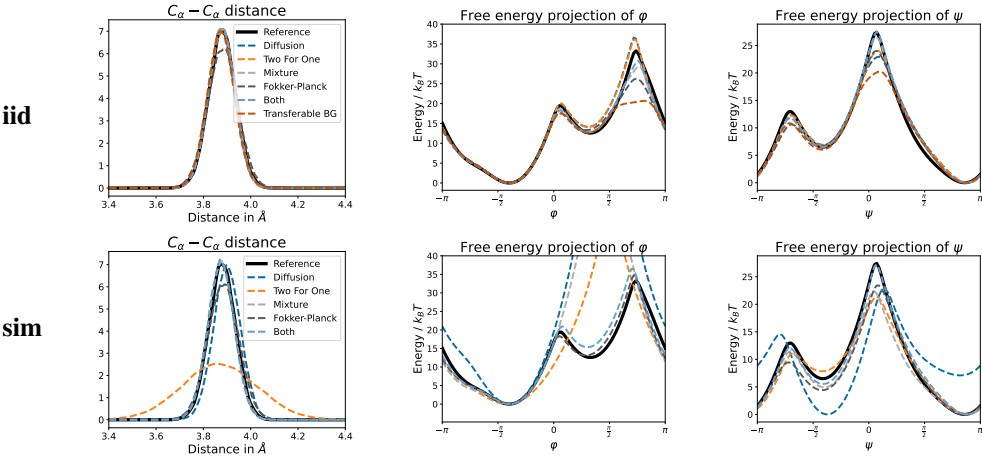

Figure 34: **RV:** We compare further metrics between iid sampling and Langevin simulation. We compare the $C_\alpha$–$C_\alpha$ distance for the dipeptides and also the free energy projections along the dihedral angles $\varphi, \psi$.

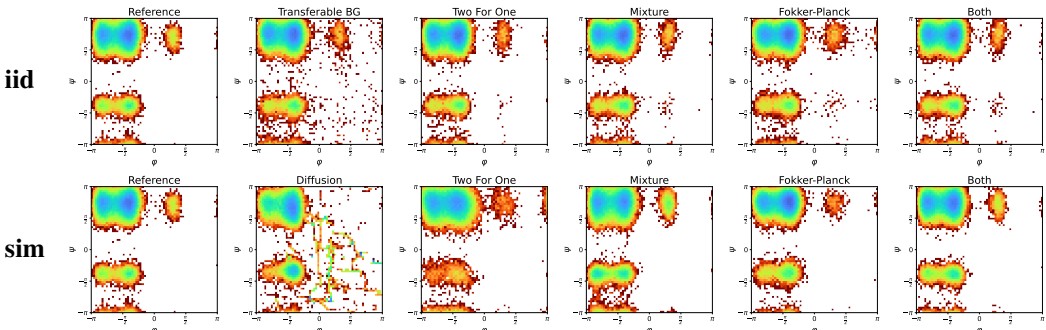

Figure 35: **TD:** We compare the free energy plot on the dihedral angles $\varphi, \psi$ for all presented methods for iid sampling and Langevin simulation.

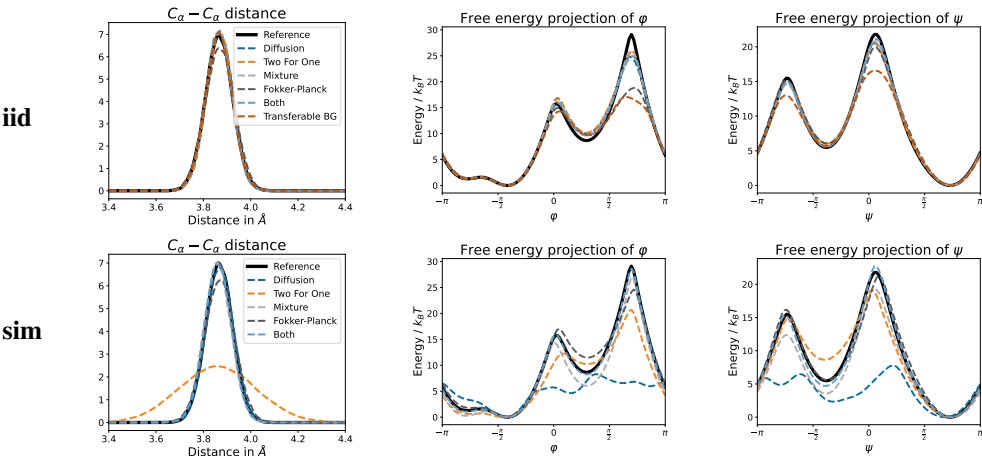

Figure 36: **TD:** We compare further metrics between iid sampling and Langevin simulation. We compare the $C_\alpha$–$C_\alpha$ distance for the dipeptides and also the free energy projections along the dihedral angles $\varphi, \psi$.

# D  Societal Impact

Our work focuses on improving the efficiency of molecular sampling and simulation. We consider this research foundational, with the potential to accelerate applications such as drug and material discovery. While we do not identify any immediate risks, the technology could be misused, for example, in the development of biological weapons. Furthermore, our method currently does not provide formal guarantees, which poses a risk of misleading downstream research if the method produces incorrect or biased results.

