# OpenReview forum: "Consistent Sampling and Simulation: Molecular Dynamics with Energy-Based Diffusion Models"
_NeurIPS.cc/2025/Conference — NeurIPS 2025 poster_

### Official Review · Reviewer_ad79 · 2025-06-24

**Clarity:** 3
**Significance:** 2
**Originality:** 3
**Rating:** 3
**Confidence:** 3

**Summary:**

The paper proposes a regularization loss for diffusion model that is based on Fokker–Planck-equation (FPE) to enforce consistency between sampling and simulation modes in molecular dynamics (MD) simulations. The authors identify that diffusion models, when trained on equilibrium molecular distributions, exhibit inconsistencies between diffusion reverse process (backward inference) and use the first score function for Langevin simulation. To address this, they introduce a computationally efficient regularization method that ensures consistency. They validate their approach on a simple toy system, alanine dipeptide, and a transferable Boltzmann emulator for dipeptides demonstrating improved consistency and efficient sampling.

**Questions:**

1. For the definition of $L_{FP}$ (line 132), which does the D^{-2} denote or indicate?
2. How do you obtain the $\log p^{\theta}(x, t)$ in equation 13 as your practical estimation? This term is not trivial for most of the diffusion models. Which is the concrete form of residual error (eq (8))?
3. In equation (6), i do not quite understand the derivation in the following sense: the EBM that correlates p(x) and U(x) has its support over the whole configuration space of x (exp is non-negative). However, the learned score at t=0 $s^{\theta}(x)$ does not necessarily capture the gradient field with arbitrary x, especially the region far from the data manifold. I am hesitant about the correctness of such derivation. Following such derivation, one can even simply train the network via denoising score matching to estimate the "force" around the data manifold. Training diffusion models on multi-tasking across multiple time steps yet throwing away the score network for time t>0 seems weird for sampling. Maybe you can elaborate or justify it furthermore.
4. What does it mean by $NNET(x,t)$ in line 155 and line 156?
5. In practical concern, why do we actually need to do “sim” instead of just “iid” for diffusion sampling? ("sim", "iid" are following the notation in the experimental section)

**Ethical Concerns:**

["NO or VERY MINOR ethics concerns only"]

**Final Justification:**

Summary. The work is well-motivated and original albeit the limited scale of systems studied. The major concerns hover around 1) the motivation of using "simulation" manner for sampling while the generative models (diffusion) are trained by i.i.d samples (via the score matching objective). It is hard to straight conclude whether the learned score function s(x, t) actually reflects the underlying physical kinetic like the physical force fields rather than hypothetical hallucination; 2) The experiments only involves toy peptides systems and BBA while the main baseline in this study ("two for one") had tested for several fast folding proteins.

Regarding evaluation. A good motivation of using diffusion model in replace of traditional MD simulation is the prohibitive scaling law of system size with force field and exhaustive simulation. For the small system, the MD simulation (+enhanced sampling methods) can already work well, while the authors do not show promise of this method to work for larger systems. That being said, the regularization according to FP equations is very interesting and of good significance. After taking rebuttal and responses into consideration, I lean toward borderline rejecting this paper mainly due to the limited evaluation for properly justification of the proposed regularization.

**Limitations:**

Yes.

**Paper Formatting Concerns:**

No major formatting issues.

**Quality:**

3

**Strengths And Weaknesses:**

Strength:
- Introducing Fokker-Planck equation (FPE) based regularization loss for diffusion models to learn to sample consistently with the energy-based model formulation.
- The experiments demonstrate that with the proposed loss, the diffusion works well by generating consistent sample using langevin dynamics (LD) simulator, on MB potential, ALDP and dipeptide tasks.

Weakness:
- While the paper empirically demonstrates improvements on toy systems, the theoretical underpinning connecting regularization to broader implications of diffusion models is insufficiently developed, which may not support the main claim of diffusion models in the paper.
- The writing is not clear enough and seems to lack of necessary explanation (see equations).
- The significance of the methodological advance seems unclear. I am not convinced how incorporating the FP loss improves the diffusion sampling from the experimental results.

---

> ### Author Rebuttal · Authors · 2025-07-30
>
> We thank the reviewer for their feedback and concrete questions, which help us clarify the contributions of the paper. It seems that some aspects of the application context, particularly the importance of simulation, may not have been introduced with sufficient clarity. We therefore briefly revisit the motivation and have reordered the questions to address them more coherently.
>
>
> > In practical concern, why do we actually need to do "sim" instead of just "iid" for diffusion sampling? ("sim", "iid" are following the notation in the experimental section)
>
> The goal of our work is to build diffusion models that support not only independent (iid) sampling but also molecular dynamics (MD) simulation (sim) by leveraging the learned score through Langevin dynamics. Previous work [1] has failed to recover the underlying forces, and the simulation produced different results from sampling.
>
> While iid sampling allows us to generate equilibrium configurations, simulation enables access to kinetic properties and dynamical information, which is crucial for many scientific applications. For example, to study how a molecule binds to a protein, we need temporally coherent trajectories; something that iid samples cannot provide. Whole fields of biochemistry (e.g., Transition Path Sampling) are interested in studying such dynamics. Furthermore, sim enables targeted exploration of rare states. Starting from a single low-probability conformation, we can simulate the system and uncover long-timescale behavior without the inefficiency of repeatedly sampling iid until the desired state reappears.
>
> In addition, the ability to estimate forces and the potential from the score opens up opportunities for advanced sampling techniques (e.g., metadynamics, umbrella sampling), which would otherwise be inaccessible.
>
> We agree that our introduction could better highlight these practical motivations, and we thank the reviewer for pointing this out. We will revise the introduction to make these use cases more explicit and will add supporting literature.
>
> > In equation (6), i do not quite understand the derivation in the following sense: [...] the learned score at t=0 $s^{\theta}(x)$ does not necessarily capture the gradient field with arbitrary x, especially the region far from the data manifold. [...]
>
> We would like to clarify that Eq 6 is a well-established identity that has been discussed in prior work [1] and follows directly from the Boltzmann distribution under the assumption that the model accurately learns the score (see Sec 2.2).
>
> The more relevant question, that we explicitly address, is whether this identity can be realized in practice. As discussed in Sec 2.1, the score function at $t=0$ can be non-smooth, and training via denoising score matching (DSM) becomes numerically unstable. This is a known limitation of diffusion models and has been documented in the literature [1,2,3].
>
> We improve the accuracy of the learned score at small $t$ with a Fokker-Planck regularization. Importantly, the introduced loss can be evaluated even on samples farther from the data distribution, and offers a mechanism to improve generalization in regions where DSM alone provides limited guidance. While no model can be expected to yield perfect gradients far from the training data, our method helps mitigate this limitation.
>
> We will introduce a discussion of this in a revision.
>
> > While the paper empirically demonstrates improvements on toy systems, the theoretical underpinning connecting regularization to broader implications of diffusion models is insufficiently developed [...]
>
> Our work specifically targets diffusion models trained on molecular systems. In this setting, the connection between the model's score and the physical force field is well established, and simulation via Langevin dynamics becomes meaningful. While the Fokker-Planck regularization is derived from first principles, our claims remain scoped to molecular systems where these conditions apply.
>
> Empirically, we show that our approach improves consistency between diffusion sampling (iid) and energy-based simulation (sim). These improvements are not limited to low-dimensional toy systems: we have extended our evaluation to larger molecules, including proteins such as BBA (28 amino acids). These systems go well beyond the dipeptide examples and exhibit the same trends:
>
> **BBA Results:**
> |Method|iid JS|sim JS|iid PMF|sim PMF|
> |---|---|---|---|---|
> |Diffusion|**0.0043 ± 0.0000**|0.2014 ± 0.0043|**0.034 ± 0.000**|5.387 ± 0.144|
> |Two For One|**0.0043 ± 0.0000**|0.1162 ± 0.0021|**0.034 ± 0.000**|1.624 ± 0.107|
> |Fokker-Planck|0.0070 ± 0.0000|**0.0427 ± 0.0053**|0.060 ± 0.000|**0.438 ± 0.074**|
>
> To our knowledge, we are the first to provide consistent sampling and dynamics for such high-dimensional systems, where previous attempts [1] failed. These results reinforce our main claim: our regularization improves the consistency between sampling and simulation for molecules, even in high-dimensional settings.
>
> > The significance of the methodological advance seems unclear. I am not convinced how incorporating the FP loss improves the diffusion sampling from the experimental results.
>
> We want to clarify that our goal is not to improve standard iid sampling but to enable consistent simulation for diffusion models by learning accurate scores for small $t$. This distinction is important: while we maintain comparable performance to state-of-the-art methods for iid generation, our focus is on enabling *simulation*, which traditional diffusion models do not support reliably [1].
>
> The core idea behind the Fokker-Planck regularization is to improve the score estimation at small diffusion times $t$, where DSM becomes unstable. As discussed in Sec 2.1, these scores can be non-smooth and ill-conditioned, making accurate learning difficult. In contrast, at larger $t$, the score is smoother and easier to learn.
>
> The Fokker-Planck equation provides a consistent relation between scores across time. By minimizing the Fokker-Planck residual, we leverage the more stable learning at large $t$ and *propagate this information back* to smaller $t$, leading to improved estimates near the data manifold. This can be seen in Figure 6, where the deviation from the Fokker-Planck equation at $t=0$ is reduced by nearly two orders of magnitude with our regularization.
>
> We will revise the paper to make this distinction and motivation clearer.
>
> > [...] one can even simply train the network via denoising score matching to estimate the "force" around the data manifold. Training diffusion models on multi-tasking across multiple time steps yet throwing away the score network for time t>0 seems weird for sampling. Maybe you can elaborate or justify it furthermore.
>
> We do not discard any of the models. For iid sampling, we use all expert models across the full timeline, recovering standard diffusion behavior. For simulation, we only use the expert trained on small $t$, which provides the most accurate estimate of the score near the data manifold. In the case where we only need simulation, we can train only this expert.
>
> Note that we cannot train a model purely at $t=0$, as the DSM loss is ill-defined there (see Sec 2.1). We need to train over an interval (which we do in MoE), and even then, this region is difficult to learn accurately. We show in Sec 5 that the regularization improves performance in this regime and enables MD simulations that standard diffusion training cannot support.
>
> We will improve the paper by more clearly distinguishing the objectives of sampling and simulation, and by better motivating the role of MoE and regularization in each.
>
> > The writing is not clear enough and seems to lack of necessary explanation (see equations).
>
> We thank the reviewer for their feedback. We agree that the paper would benefit from clearer motivation to reach a wider audience. We will revise the introduction to better explain why learning forces from data is important (see first answer). To improve clarity, we will include an algorithm outlining training and inference steps.
>
> Next, we will answer the questions about the equations.
>
> > For the definition of $L_{FP}$ (line 132), which does the D^{-2} denote or indicate?
>
> As denoted in line 132: "and define the corresponding loss as $L_{FP}[\log p^{\theta}](x, t) = \lambda_{FP}(t) D^{-2} \left\Vert R(x, t) \right\Vert^2_2$, where $x \in \mathbb{R}^D$", $D$ is the dimension of the data.
>
> > How do you obtain the $\log p^{\theta} (x, t)$ in equation 13 as your practical estimation? This term is not trivial for most of the diffusion models. Which is the concrete form of residual error (eq (8))?
>
> As discussed in Section 3.2, computing $\log p^{\theta}(x,t)$ is non-trivial for standard diffusion models. However, we use an energy-based parameterization where the score is defined as the gradient of a scalar neural network output $s^{\theta}(x,t) = \nabla_x \log p^{\theta}(x,t)$, which gives us access to the energy.
>
> For the concrete form of the residual used in training, we refer to Eq 11.
>
> > What does it mean by $NNET(x,t)$ in line 155 and line 156?
>
> NNET denotes any neural network. In this context, the question likely refers to the distinction between two parameterizations: modeling the score directly with a neural network, versus modeling the score as the gradient of an energy function.
>
> We adopt the latter, energy-based formulation (see Sec 3.2), which gives us access to the unnormalized log-density required for computing the Fokker-Planck residual. We will try to make the notation clearer and will revise it to better highlight this distinction.
>
> -------
>
> [1] Arts et al. (2023). Two for one: Diffusion models and force fields for coarse-grained molecular dynamics
>
> [2] Kim et al. (2022). Soft truncation: A universal training technique of score-based diffusion model for high precision score estimation
>
> [3] Koehler et al. (2023). Statistical efficiency of score matching: The view from isoperimetry

---

> > ### Comment · Reviewer_ad79 · 2025-08-05
> >
> > Thanks the authors for their detailed response in addressing my questions and concerns. I appreciate any revisions on improving the readability of this manuscript by properly explaining any notation.  I have the following concerns and want to discuss:
> >
> > 1. Motivation-wise. I still cannot be convinced why we need to go "sim" instead of "iid" in the sense of generative learning. First of all, the score or "force" is learned along with the pre-defined (mostly gaussian) process of diffusion and the model simply is forced to imitate that during training. Surely you can draw relationships between physical force and score function and this is the motivation of denoising diffusion. However, the learned kinetic behavior is largely dependent upon the forward process used to train the diffusion model, in my opinion. If you use rectified flow (flow matching) or schrodinger bridge, the behavior could be different. Why (or How) does such imitation reveal the intractable kinetic behavior of real molecular systems?
> >
> > 2. According to the authors:
> > > In addition, the ability to estimate forces and the potential from the score opens up opportunities for advanced sampling techniques (e.g., metadynamics, umbrella sampling), which would otherwise be inaccessible.
> >
> > I do not agree with this is proper claim which attempts to reinforce the "significance" of the proposed method. Enhanced sampling methods have been standing long before this work and they can work well without the need of estimating neural force field or something. "otherwise be inaccessible" seems misleading to me.
> >
> > 3. The justification of the proposed regularization. From my understanding (let me know if I go wrong), the paper did not introduce brand new model or learning paradigm but a "key" term in training of diffusion to make the learned force $s(x, t\approx 0)$ (near t=0) more consistent, and orthogonal to the specific form of diffusion models. I believe we should be careful about introducing this since it may be misleading without strong experiments support. From the experimental results, the visualization and quantitative evaluations on the peptides (or BBA) are not convincing enough to believe this term can be so significant to support it. An advice for improving this is applying this for a broader family of diffusion models and benchmark across different scales of systems (eg. twelve fast folding proteins, or larger protein like GPCR) , which however might be out of the scope of rebuttal and discussion phase.

---

> ### Author Response · Authors · 2025-08-05
> **Enhanced sampling with diffusion models**
>
> 1/3
>
> We thank the reviewer for taking the time to write the follow-up questions. We have reordered the questions and will address them separately.
>
> > I do not agree with this is proper claim which attempts to reinforce the "significance" of the proposed method. Enhanced sampling methods have been standing long before this work and they can work well without the need of estimating neural force field or something. "otherwise be inaccessible" seems misleading to me
>
> To clarify, these are two different things. MD needs enhanced sampling because it has a sampling problem that includes (but isn't limited to) a barrier-crossing problem. Enhanced sampling methods like thermodynamic integration, REMD, Metadynamics, Umbrella Sampling etc have been developed decades ago, actually starting with Zwanzig in the 1950s - that clearly isn't novel.
>
> Diffusion models do no have the same sampling problem because they generate samples iid - they don't have to cross barriers. But they still do have a sampling problem in that states that have a low probability in the distribution the diffusion model is trained on (e.g., the equilibrium distribution of a molecule) are sampled rarely. For example, a free energy difference of 5 kcal/mol corresponds to having to draw about 4000 samples to get one sample in the rare state, and of course these are 4000 denoising trajectories that aren't that cheap - so sampling the equilibrium distribution isn't efficient when you want to compute large free energy differences.
>
> Therefore enabling diffusion models to access all of the biasing/unbiasing tricks that have been developed for rare-event sampling for MD would be very impactful, and is in our opinion a prerequisite to make diffusion samplers for biomolecular structures useful for practical applications. The most natural way to do this is indeed to achieve consistency between the diffusion model sampling probability p(x) and an energy u(x), such that p(x) ~ exp(-u(x)), in other words, ensure that the diffusion model is a proper energy-based model. Once you have that equivalence, all of the rare-event sampling tricks that have been developed for MD in the last 7 decades become available for diffusion model samplers. And that's the point of this paper.

---

> ### Author Response · Authors · 2025-08-06
>
> 2/3
>
> > Motivation-wise. I still cannot be convinced why we need to go "sim" instead of "iid" in the sense of generative learning.
>
> We respectfully disagree with the framing that simulation (sim) is unnecessary for generative modeling. We believe that the comment above gives additional insight into why this consistency is needed, and we will add more motivation for these practical reasons.
>
> To make it more specific, we also provide results on the temporal evolution, where we estimate the transition probability from the underlying Markov chain to compare the probability of transitioning to a different state (e.g., folding and unfolding) and compare it with reference simulations. Again, we can show consistently better results (see table below). These results are only possible with sim and not with iid.
>
> **Chignolin**
> | Method | Transition Probability JS |
> |---|---|
> | Diffusion | $3.4 \cdot 10^{-2}$ |
> | Two For One | $2.3 \cdot 10^{-3}$ |
> | Mixture | $3.0 \cdot 10^{-2}$ |
> | Fokker-Planck | $4.6 \cdot 10^{-4}$ |
> | Both | $\mathbf{2.1 \cdot 10^{-4}}$ |
>
> **BBA**
> | Method | Transition Probability JS |
> |---|---|
> | Diffusion | $6.8 \cdot 10^{-3}$ |
> | Two For One | $7.1\cdot 10^{-3}$ |
> | Fokker-Planck | $\mathbf{1.4 \cdot 10^{-3}}$ |
>
>
> Further, an extended list of reasons can also be found in the comment to reviewer c2SZ.
>
> We will additionally include a figure visualizing the time evolution in the TIC space, showing how folding and unfolding transitions occur throughout the simulations, including renderings of the proteins. We thank the reviewer for this suggestion and believe that these additional results further strengthen the claims of the paper.
>
> > First of all, the score or "force" is learned along with the pre-defined (mostly gaussian) process of diffusion and the model simply is forced to imitate that during training [...]
>
> We want to restate, that the identity we provide in Eq. 6, relating the score  $\nabla_x \log p$ to molecular forces is a known property [1] that holds for any Boltzmann-distributed data, regardless of the generative model. Diffusion models are uniquely useful here because they learn the score explicitly, giving us access to the forces without requiring labeled force data.
>
> > However, the learned kinetic behavior is largely dependent upon the forward process used to train the diffusion model, in my opinion. If you use rectified flow (flow matching) or schrodinger bridge, the behavior could be different. Why (or How) does such imitation reveal the intractable kinetic behavior of real molecular systems?
>
> We agree that the identity $\nabla_x \log p$ ≈ force holds only if the model accurately learns the data score and the samples reflect the correct distribution. It does not hold universally across all generative schemes unless they recover the same score.
>
> However, this identity is not specific to the forward process used in diffusion models and holds for any method that can accurately estimate $\nabla_x \log p$, including flow matching and Schrödinger bridges. For instance, in standard Gaussian flow matching, the learned vector field $v^\theta$ relates to the score as [4]:
> $$
> \nabla_x \log p_t(x) = \frac{1}{1- t} (t \cdot v^\theta(x, t) - x).
> $$
> Thus, one can also extract forces from flow-matching models via reparameterization. In our experiments, however, we found that flow-based models performed worse near $t \approx 0$, likely due to higher stochasticity. Similarly, Schrödinger Bridges also provide access to scores $\nabla_x \log p$ [5] though the object often coincides with denoising score matching [6].
>
> Still, while the identity enables force extraction, our proposed regularization is based on the Fokker-Planck equation specific to diffusion-based SDEs and cannot be directly applied to other generative frameworks without modification (e.g., replacing it with the Liouville equation for deterministic flows), which are out of scope here but potentially promising directions. We will clarify this distinction further in the appendix.
>
> ------
>
> [4] Lipman, Y., Havasi, M., Holderrieth, P., Shaul, N., Le, M., Karrer, B., ... & Gat, I. (2024). Flow matching guide and code
>
> [5] De Bortoli, V., Thornton, J., Heng, J., & Doucet, A. (2021). Diffusion schrödinger bridge with applications to score-based generative modeling
>
> [6] Tong, A., Malkin, N., Fatras, K., Atanackovic, L., Zhang, Y., Huguet, G., ... & Bengio, Y. (2023). Simulation-free schr\" odinger bridges via score and flow matching

---

> > ### Author Response · Authors · 2025-08-06
> >
> > 3/3
> >
> > > The justification of the proposed regularization. From my understanding (let me know if I go wrong), the paper did not introduce brand new model or learning paradigm but a "key" term in training of diffusion to make the learned force $s(x, t \approx 0)$ (near t=0) more consistent, and orthogonal to the specific form of diffusion models. I believe we should be careful about introducing this since it may be misleading without strong experiments support. From the experimental results, the visualization and quantitative evaluations on the peptides (or BBA) are not convincing enough to believe this term can be so significant to support it. An advice for improving this is applying this for a broader family of diffusion models and benchmark across different scales of systems (eg. twelve fast folding proteins, or larger protein like GPCR) , which however might be out of the scope of rebuttal and discussion phase.
> >
> > We agree with the reviewer that the proposed regularization that enforces the Fokker-Planck equation does not change the architecture (instead of an energy-based parameterization), and we do not claim otherwise. The contribution lies in identifying and correcting a *specific, practically important* failure mode in existing diffusion models for molecules. Namely, that the score violates the Fokker-Planck equation.
> >
> > As we discuss in Sec 3.1, the Fokker-Planck equation is not an additional assumption but a fundamental property of the diffusion process. In fact, minimizing the denoising score matching loss already implicitly minimizes the residual of the Fokker-Planck equation [7]. Therefore, satisfying this equation is not an optional constraint but a requirement that any well-trained diffusion model should fulfill, hence it is *not an orthogonal training objective*. A violation of this relation indicates a model that fails to accurately represent the score. Our contribution is to make this constraint explicit during training, which we show leads to improved consistency.
> >
> > We demonstrate our results on BBA and generalization across 400 dipeptides, which goes well beyond toy settings. Prior work [1] failed to accurately model these systems and our benchmarks show **improvements by more than one magnitude** compared to classical diffusion. The improvements are robust across systems of varying size and complexity, and show that our method scales beyond synthetic or idealized examples. We also provide a wide range of experiments, including dynamic analysis for high-dimensional proteins (see above), beating state of the art diffusion models.
> >
> >
> > ----
> >
> >
> > [7] Albergo, M. S., Boffi, N. M., & Vanden-Eijnden, E. (2023). Stochastic interpolants: A unifying framework for flows and diffusions

---

> > > ### Comment · Reviewer_ad79 · 2025-08-06
> > >
> > > Thank a lot to the authors for the responses that basically answers my questions. Interestingly to see the vanilla diffusion objective does not make the score function adhere to the FP equations very well while the authors proposed to add a new regularization term to push this. The work is well-motivated albeit the limited scale of systems studied. Overall, I will increase my score accordingly.

---

> > > > ### Author Response · Authors · 2025-08-08
> > > >
> > > > We are glad that our comments could resolve all questions of the reviewer and would like to thank them for the constructive discussions. We agree that it is surprising to see that the diffusion objective does not lead to stable scores, and we attribute this to numerical instabilities close to the data distribution. We believe that including further discussion about the motivation and intuition of the Fokker-Planck equation and its relation to other generative paradigms will improve the manuscript.

---

### Official Review · Reviewer_TwQZ · 2025-07-03

**Clarity:** 3
**Significance:** 2
**Originality:** 3
**Rating:** 4
**Confidence:** 2

**Summary:**

The paper addresses the inconsistency between molecular dynamics (MD) simulations and traditional sampling in diffusion models by introducing a Fokker–Planck-based regularization to the diffusion loss. To reduce computational costs, the authors use efficient approximations and a time-based mixture of experts architecture. Experiments on toy systems, alanine dipeptide, and transferable dipeptide settings demonstrate that their approach improves consistency between sampling and simulation. In short, the contribution of this paper lies in solving the inconsistency, i.e., mismatch between how we generate samples from a model and how we use it to simulate time-dependent behavior.

**Questions:**

1. Mixture intervals

A simple question in Section 5. The Mixture of Experts (MoE) model is trained on diffusion time intervals (0, 0.1), [0.1, 0.6), and [0.6, 1.0). Could the authors clarify the intuition or practical rationale behind choosing these specific cutoffs? Were they based on empirical findings, model behavior (e.g., Fokker–Planck error), or heuristic choices?

2. High variance of Mixture of experts in simulations results

In Tables 1 and 2, the Mixture of Experts model shows relatively high variance in the simulation metrics compared to other methods. This seems like an interesting and potentially important observation. Could the authors offer an explanation or intuition for this behavior?

3. Experiment metric (JS)

From section 5, it seems that the JS divergence is computed for the energy distribution. Could the authors state why this is done, instead of JS divergence on dihedral angles / TIC(time-lagged independent component) or pairwise distance as in Two For One? A simple results on JS divergence as in Two For One or why they have chosen energy would be good be enough to resolve my question.

**Ethical Concerns:**

["NO or VERY MINOR ethics concerns only"]

**Final Justification:**

The authors have well resolved almost all of my concerns and questions, therefore I modify the score accordingly.

**Limitations:**

yes, the authors adequately addressed the limitations and potential negative societal impact of their work.

**Quality:**

2

**Strengths And Weaknesses:**

**Strengths**

1. (Clarity) clear and organized presentation

The paper is clearly written and well organized, making it easy to follow. Most experiments are well designed and support the authors’ claims across various settings.

**Weakness**

1. (Quality) Quantitative transferable dipeptides results

For the transferable dipeptide, quantitative and qualitative results only for dipeptide AC has been presented in the main paper. Though there are qualitative results on other test dipeptides in the appendix, I do not see any quantitative results on other peptides.

(Minor) Moreover, additional qualitative results for dipeptides could be helpful. Specially, most dipeptide show similar landscape to the Ramachandran plot in Figure 5 and 12, while dipeptides containing proline and glycine shows some different Ramachandran plot trends. Though there is one in Figure 14, one or two additional test peptides results with different Ramachandran plot trends could strengthen the claim for the generalization of the proposed method in the transferable dipeptide settings.

2. (Quality) Efficiency of training and inference

While the authors have emphasized that the proposed method achieves efficient training and inference, I feel that the experiment results fall short on this. The author have claimed efficiency in many aspects, such as

- line 58 (contribution 2) achieve efficient training and inference
- line 80 efficiently evaluate the loss
- line 150 efficient approximation of the loss
- line 183 MOE approach to allocate model capacity more efficiently

There are not results supporting this claim in the main paper, while there is table 7 in the appendix reports the training and inference time for the proposed method. However, methods including Fokker-Planck regularization terms shows several longer training time than the original model.

3. (Significance) Dynamic results

The authors have experimented only to dipeptide system scales, while a major baseline in the paper “Two For One” has done experiments to fast folding protein system scales (chignolin, BBA, etc). Experiments on systems with this scales allows dynamics analysis as in Figure 5 of the Two For One paper. Testing two or three fast folding protein could strengthen the paper’s claim a lot, since this dynamics analysis is closely related to why we run simulations.

---

> ### Author Rebuttal · Authors · 2025-07-30
>
> We thank the reviewer for their thoughtful review and agree that results on larger systems would strengthen the paper. We now include fast-folding proteins, which further support our main claims. In the following, we address all points raised by the reviewer.
>
> > For the transferable dipeptide, quantitative and qualitative results only for dipeptide AC has been presented in the main paper. [...] I do not see any quantitative results on other peptides.
>
> The quantitative results in Table 2 are in fact computed across the *entire test set*, not just the dipeptide AC. The table caption is mislabeled in this regard. The rest of the caption is accurate "To compute the mean and standard deviation, we have averaged the metrics across the dipeptides from the test set." We thank the reviewer for pointing this out and we will fix this.
>
> > (Minor) Moreover, additional qualitative results for dipeptides could be helpful. [...]. Though there is one in Figure 14, one or two additional test peptides results with different Ramachandran plot trends could strengthen the claim for the generalization [...]
>
> The dipeptides shown in App. C5 were selected randomly from the test set, as we found them to be representative in terms of model performance. We agree that including additional examples, particularly those that exhibit more diverse behavior, could better support our generalization claims. We are happy to add such results to the appendix.
>
> > While the authors have emphasized that the proposed method achieves efficient training and inference, I feel that the experiment results fall short on this. The author have claimed efficiency in many aspects, [...]
>
> > There are not results supporting this claim in the main paper, while there is table 7 in the appendix reports the training and inference time for the proposed method. However, methods including Fokker-Planck regularization terms shows several longer training time than the original model.
>
> We thank the reviewer for this observation and agree that our use of the term efficient could be more consistent and better contextualized. To clarify, we do *not* claim efficiency in terms of wall-clock training time for models with Fokker-Planck regularization, which we acknowledge introduces overhead (see conclusion/limitations in Section 6 and Appendix C.3). Below, we provide additional context for each instance and will revise the manuscript to make these distinctions clearer:
> > line 58 (contribution 2) achieve efficient training and inference
>
> This is found in our second contribution, where we introduce the mixture of experts (MoE) scheme as an independent contribution. MoE reduces cost by training smaller models for most of the diffusion timeline.
> > line 80 efficiently evaluate the loss
>
> This statement appears in the background section and refers to the standard denoising score matching (DSM) loss, which has a closed-form expression for the VP SDE and can be computed efficiently.
> > line 150 efficient approximation of the loss
>
> Here, we refer to the *approximations* to make the Fokker-Planck regularization tractable. The original loss involves second-order derivatives, which would be intractable in high dimensions. Here we refer to the weak formulation and finite-difference approximation, which make the regularization feasible. That said, we agree that calling this "efficient" without qualification may be misleading, and we will revise the wording accordingly.
> > line 183 MOE approach to allocate model capacity more efficiently
>
> Here, efficiency refers to how model capacity is used, not training speed. When using a single model across all timesteps, capacity must be shared uniformly. In contrast, the MoE approach allows us to tailor model complexity to the difficulty of the subtask, improving how the model capacity is allocated.
>
> We appreciate the reviewer's attention to this detail and will clarify these aspects in the final version.
> > [...]  a major baseline in the paper "Two For One" has done experiments to fast folding protein system scales (chignolin, BBA, etc). Experiments on systems with this scales allows dynamics analysis as in Figure 5 of the Two For One paper. Testing two or three fast folding protein could strengthen the paper's claim a lot, since this dynamics analysis is closely related to why we run simulations.
>
> We thank the reviewer for this valuable suggestion. We agree that demonstrating scalability to larger systems would further strengthen our claims, especially in the context of dynamical consistency.
>
> Following this suggestion, we prepared results on **Chignolin** and **BBA**. As with smaller systems, we observe that diffusion models produce unstable simulations, and that "Two For One" introduces notable noise in the trajectories, or in the case of BBA, tends to fragment the distribution across multiple modes. Our Fokker-Planck-regularized models yield significantly improved simulation results, with better alignment between sampling and simulation.
>
> **Chignolin**
> |Method|iid JS|sim JS|iid PMF|sim PMF|
> |---|---|---|---|---|
> |Diffusion|**0.0036 ± 0.0001**|0.4351 ± 0.0141|**0.027 ± 0.000**|63.804 ± 0.372|
> |Two For One|**0.0036 ± 0.0001**|0.1023 ± 0.0008|**0.027 ± 0.000**|1.438 ± 0.019|
> |Mixture|0.0042 ± 0.0001|0.4336 ± 0.0075|0.033 ± 0.000|11.185 ± 0.430|
> |Fokker-Planck|0.0048 ± 0.0001|**0.0050 ± 0.0001**|0.037 ± 0.000|0.040 ± 0.001|
> |Both|0.0045 ± 0.0001|**0.0050 ± 0.0008**|0.035 ± 0.001|**0.038 ± 0.006**|
>
> **BBA**
> |Method|iid JS|sim JS|iid PMF|sim PMF|
> |---|---|---|---|---|
> |Diffusion|**0.0043 ± 0.0000**|0.2014 ± 0.0043|**0.034 ± 0.000**|5.387 ± 0.144|
> |Two For One|**0.0043 ± 0.0000**|0.1162 ± 0.0021|**0.034 ± 0.000**|1.624 ± 0.107|
> |Fokker-Planck|0.0070 ± 0.0000|**0.0427 ± 0.0053**|0.060 ± 0.000|**0.438 ± 0.074**|
>
> We will include these results in the revised manuscript, along with a dynamic analysis akin to Figure 5 in Two For One, to support the evaluation of kinetic consistency in larger systems.
>
> > [...] The Mixture of Experts (MoE) model is trained on diffusion time intervals (0, 0.1), [0.1, 0.6), and [0.6, 1.0). Could the authors clarify the intuition or practical rationale behind choosing these specific cutoffs? Were they based on empirical findings, model behavior (e.g., Fokker-Planck error), or heuristic choices?
>
> The choice of intervals was based primarily on empirical findings. We experimented with different numbers and placements of intervals and found that performance was generally robust; especially for the larger diffusion times, where the data is more heavily corrupted and the model behavior is less sensitive to the exact range.
>
> That said, for the smallest interval used during simulation, we observed that setting the upper bound too low (e.g., < 0.1) made the model less robust to noise and led to poor generalization. Conversely, setting it too high introduces high variability in the data, making it harder for the model to focus on the detailed structures required for simulation.
>
> The chosen partition seems to strike a balance between specialization and stability, and also regularizes the region with the highest Fokker-Planck error. We will include a discussion of this design choice in the revised manuscript to better motivate the interval selection.
>
> > In Tables 1 and 2, the Mixture of Experts model shows relatively high variance in the simulation metrics compared to other methods. This seems like an interesting and potentially important observation. Could the authors offer an explanation or intuition for this behavior?
>
> The higher variance of the MoE model can be best explained with the alanine dipeptide example. Neither the MoE nor the single-model baselines consistently recover the low-probability region in simulation. However, while the single model tends to systematically miss this mode, the MoE model occasionally captures it.
>
> We attribute this to the increased capacity of MoE, which allows it to "remember" more of the training distribution. However, since it does not explicitly address the underlying inconsistency at small diffusion times, the model may or may not recover rare modes depending on initialization or training noise, leading to the observed variance.
>
> This further motivates our combination of MoE with Fokker-Planck regularization. We thank the reviewer for highlighting this observation, and we will clarify this point in the revised manuscript.
>
> > From section 5, it seems that the JS divergence is computed for the energy distribution. Could the authors state why this is done, instead of JS divergence on dihedral angles / TIC(time-lagged independent component) or pairwise distance as in Two For One? A simple results on JS divergence as in Two For One or why they have chosen energy would be good be enough to resolve my question.
>
> We thank the reviewer for raising this point. We would like to clarify that we do *not* compute JS divergence on energy distributions as this would not be feasible in our coarse-grained setting, where the true energy function is unknown. Instead, all reported metrics are based on 2D projections such as dihedral angles or TICA coordinates, as suggested by the reviewer, consistent with the approach taken in Two For One.
>
> We agree that pairwise distance (PWD)-based metrics provide a useful complementary view. Following the suggestion, we have computed JS divergence on the PWD histograms which can be found below. We will include these results in the revised manuscript and clarify the basis of all reported metrics.
>
> **Chignolin**
> |Method|iid PWD JS|sim PWD JS|
> |---|---|---|
> |Diffusion|**0.0001 ± 0.0000**|0.3817 ± 0.0009|
> |Two For One|**0.0001 ± 0.0000**|0.0082 ± 0.0000|
> |Mixture|0.0003 ± 0.0000|0.2045 ± 0.0004|
> |Fokker-Planck|0.0004 ± 0.0000|**0.0008 ± 0.0000**|
> |Both|0.0003 ± 0.0000|**0.0012 ± 0.0005**|
>
> **BBA**
> |Method|iid PWD JS|sim PWD JS|
> |---|---|---|
> |Diffusion|**0.0006 ± 0.0000**|0.1003 ± 0.0050|
> |Two For One|**0.0006 ± 0.0000**|0.0240 ± 0.0008|
> |Fokker-Planck|0.0011 ± 0.0000|**0.0104 ± 0.0010**|

---

> > ### Comment · Reviewer_TwQZ · 2025-08-05
> >
> > I thank the authors for clarifying my questions, and presenting additional experiments results on larger systems including Chignolin and BBA, with additional PWD metrics. Most of my concerns has been resolved, and I have modified my score accordingly.
> >
> > ### Q3 JS metric
> >
> > I thank the authors for the clarification. From Appendix B.1., I understood that the JS divergence was simply computed against the ground truth energy histogram. Updating the details on the revised manuscript would be great.
> >
> > ### Q2 High variance of MOE
> >
> > While I am still not entirely certain about the underlying cause and intuitions of the high variance observed in the MOE metric, I appreciate the authors’ efforts to address this issue. Considering the overall explanation and new experimental results, I have positively adjusted my score.

---

> > > ### Author Response · Authors · 2025-08-05
> > >
> > > We thank the reviewer for their continued feedback and for looking at our responses in detail. We will update the occurrences in the manuscript accordingly and will clarify how we compute the JS divergence in the appendix.
> > >
> > >
> > > Regarding Q2, we would like to restate our explanation using different words and will also add more context and visualizations to the paper to make this clearer.
> > >
> > > In short: both the standard diffusion and the MoE model have limitations when used for simulation. To understand the variance of the MoE model, consider a system with two modes. A single diffusion model may consistently miss one mode, always predicting the same (incomplete) result. This leads to high error but low variance. In contrast, the MoE model has more capacity and can sometimes capture both modes. However, it doesn’t resolve the core inconsistency at small diffusion times, so whether both modes are recovered can depend on randomness in training. This leads to lower average error but higher variance, as we average the results over multiple runs.
> > >
> > > We observe this in the alanine dipeptide example: the rare mode on the right is usually missed, but the MoE occasionally captures it, unlike the single model which never does. This motivates our final model, which combines MoE with Fokker–Planck regularization to reduce both error and variance.

---

### Official Review · Reviewer_c2SZ · 2025-07-03

**Clarity:** 3
**Significance:** 1
**Originality:** 1
**Rating:** 5
**Confidence:** 4

**Summary:**

The authors look at the problem of molecular dynamics, in particular sampling from boltzmann distributions, through the use of diffusion models and langevin sampling. They highlight that current score-based diffusion models, while immensely successful at modeling images, are not as efficient at modeling MD simulations which can be consequently improved with a combination of (a) energy parameterization as opposed to score parameterization, (b) mixture-of-experts formulation for the architecture, and (c) Fokker-planck regularization to enable consistency. Their experiments indicate that the combination of these approaches leads to better simulation results, while still maintaining good iid sampling.

**Questions:**

- It was unclear how the authors successfully train energy-based parameterization since it would require taking the gradient of the model w.r.t the state as well as consequent gradient w.r.t the parameters of the model. Is this operation efficient or are there any approximations leveraged?
- Are equations 10-12 somehow related to the hutchinson trace estimator or the estimator used in [1]?
- What is the purpose of using a mixture-of-experts system instead of just increasing the model size/capacity with number of parameters?

[1] Albergo, Michael S., and Eric Vanden-Eijnden. "Nets: A non-equilibrium transport sampler." arXiv preprint arXiv:2410.02711 (2024).

**Ethical Concerns:**

["NO or VERY MINOR ethics concerns only"]

**Final Justification:**

The authors have resolved all my concerns regarding both the use case as well as the importance of their work. Their additional experiments demonstrate the use case of simulation in scientific applications and their proposed method of including a regularization from the FPE directly leads to better results. I am happy to recommend accept.

**Limitations:**

No additional limitations.

**Paper Formatting Concerns:**

No concerns.

**Quality:**

2

**Strengths And Weaknesses:**

**Strengths**

- The authors provide tractable means of leveraging fokker-planck regularization through finite-difference based estimation of divergence and derivative terms.
- They showcase that such regularization leads to improved performance for both simulation and iid sampling, when combined with mixture-of-experts architecture.

**Weaknesses**

- The authors conjecture that learning a score function through denoising score matching leads to unreliable estimates closer to the data distribution. However, this has been a well known property in literature, and is in particular showed in detail in [1].
- The authors aim to solve the main problem of simulation which requires score at $t=0$ for which they leverage the learned score. However, typically for these applications energy, and thus forces, are available at $t=0$ so what is the aim for using the learned score from denoising score matching? I believe an insight into this was lacking in the current draft.
- While it is evident that enforcing the fokker-planck equation at intermediate time points would lead to consistent sampling, what is the reason for why it would lead to better estimation of score at $t=0$ since it is only the boundary point?

[1] De Bortoli, Valentin, et al. "Target score matching." arXiv preprint arXiv:2402.08667 (2024).

---

> ### Author Rebuttal · Authors · 2025-07-30
>
> We thank the reviewer for their comments and will use the feedback to further clarify the motivation behind our approach. Below, we address the questions in detail.
>
> > The authors conjecture that learning a score function through denoising score matching leads to unreliable estimates closer to the data distribution. However, this has been a well known property in literature, and is in particular showed in detail in [1].
>
> We agree that the limitations is well known, and we reference multiple works in our manuscript [A,B,C] that highlight this issue. We do not present this as a novel observation, but address it empirically by showing that reducing deviation from the Fokker-Planck equation improves the accuracy of the learned score, particularly near the data distribution.
>
> Importantly, approaches like target score matching [1] are not applicable here, as they require access to force labels, which we explicitly avoid (see next answer). We will gladly cite the suggested literature [1] as further support for the known limitations.
>
> > The authors aim to solve the main problem of simulation which requires score at $t=0$ for which they leverage the learned score. However, typically for these applications energy, and thus forces, are available at $t=0$ so what is the aim for using the learned score from denoising score matching? I believe an insight into this was lacking in the current draft.
>
> Our work specifically focuses on settings where force labels at $t=0$ are not available and forces cannot be computed, and hence are not available for simulation or training.
>
> This situation arises frequently in coarse-grained modeling, where the atomistic resolution is reduced and the underlying all-atom fine-grained potential (and thus the forces) are no longer well-defined. In our approach, we learn from samples alone, which allows for arbitrary levels of system abstraction (see Sec 1), similar to prior work [C].
>
> Even in systems where forces could be computed, practical limitations often prevent this. E.g., obtaining accurate forces using quantum methods such as DFT can be prohibitively expensive, while collecting configurations is often much cheaper. Also, in many experimental settings, one can measure samples, but not the forces acting on the system.
>
> As our approach does not require forces, it can also be used in mixed-data settings where we want to train e.g., using simulation and real-world data.
>
> We agree that this distinction could have been made clearer in the manuscript and will revise the introduction to better motivate the broader applicability of learning from samples alone.
>
> > While it is evident that enforcing the fokker-planck equation at intermediate time points would lead to consistent sampling, what is the reason for why it would lead to better estimation of score at $t=0$ since it is only the boundary point?
>
> While the Fokker-Planck equation is indeed enforced over intermediate time points, its inclusion improves the estimation of the score at $t=0$ through the following: When $t$ is small, the target energy (i.e., $-\log p_t$) or score function can be non-smooth, and denoising score matching (DSM) becomes numerically unstable (as discussed in Sec 2.1). In contrast, for larger $t$, the energy function becomes smoother, and DSM tends to yield more stable and accurate estimates.
>
> The Fokker-Planck equation provides a principled way to relate energy functions across time. By minimizing the Fokker-Planck residual during training, accurate estimates at large $t$ can be *propagated back* toward $t=0$, improving the estimation near the data distribution. Indeed, we observe in our numerical experiments that this leads to improved consistency and accuracy of the energy and score functions around $t=0$. This is reflected in our experiments in Sec 5. Fig 6 shows that our approach reduces the deviation from the Fokker-Planck equation at $t=0$ by nearly two orders of magnitude.
>
> Technically, our models are evaluated at a small $t > 0$ to ensure numerical stability and define gradients cleanly. In practice, we observe no difference when using $t=0$ during simulation, suggesting the learned score remains well-behaved at the boundary.
>
> Empirically, we can show that this regularization leads to better learned scores at $t = 0$ (see Sec 5). We also applied this approach to the protein BBA (28 amino acids), where to our knowledge, we are the first to provide consistent sampling and simulation [C] (see table below). We believe that this further strengthens our point that Fokker-Planck regularization improves the consistency even at the boundary.
>
> **BBA**
> |Method|iid JS|sim JS|iid PMF|sim PMF|
> |---|---|---|---|---|
> |Diffusion|**0.0043 ± 0.0000**|0.2014 ± 0.0043|**0.034 ± 0.000**|5.387 ± 0.144|
> |Two For One|**0.0043 ± 0.0000**|0.1162 ± 0.0021|**0.034 ± 0.000**|1.624 ± 0.107|
> |Fokker-Planck|0.0070 ± 0.0000|**0.0427 ± 0.0053**|0.060 ± 0.000|**0.438 ± 0.074**|
>
> We thank the reviewer for raising this point and will incorporate a more detailed explanation of this effect in the manuscript, including a discussion of the boundary behavior and its implications for score estimation.
>
> > It was unclear how the authors successfully train energy-based parameterization since it would require taking the gradient of the model w.r.t the state as well as consequent gradient w.r.t the parameters of the model. Is this operation efficient or are there any approximations leveraged?
>
> In Sec. 3.2, we note that an energy-based parameterization requires computing gradients of the model output with respect to the input. This is implemented by performing a backward pass through the network. While this introduces additional computational overhead, roughly a factor of 2, it remains tractable and has been used before [C,D].
>
> As shown in App C.2, the energy-based parameterization improves stability and simulation quality and is crucial for obtaining reliable forces for MD simulation. Similar behavior has been reported in [C].
>
> We will revise the manuscript to point to App C.2 and include the computational overhead introduced by the energy-based parameterization.
>
> > What is the purpose of using a mixture-of-experts system instead of just increasing the model size/capacity with number of parameters?
>
> Rather than increasing the size of a single model, we chose a time-based mixture-of-experts (MoE) approach for several practical reasons (see Sec 3.3).
>
> 1. Increasing model capacity increases training time, memory consumption, and inference cost. In contrast, our MoE setup trains separate, smaller models on disjoint time intervals where only one expert is active at a time. We effectively gain the benefits of a larger model without the overhead.
>
> 2. MoE allows us to match model complexity to the specific demands of each diffusion time range. Near $t=0$, the score function must resolve fine-grained structure, motivating the use of larger models. At higher $t$, the data is noisy and can be modeled effectively with smaller networks.
>
> 3. Dividing a time-dependent learning task into subproblems helps avoid optimization issues due to sharp transitions or local minima, and allows each model to focus on a simpler subproblem. This approach is common in the numerical PDE literature [E], where a separate network is trained per time segment.
>
> 4. The performance benefits of MoE go beyond scaling. Below, we compare models trained on alanine dipeptide using either a larger model ("Wide" with more hidden units and "Deep" with more layers) or MoE with three experts of equivalent total parameter count. MoE outperforms all single-model baselines in simulation accuracy, while not introducing any overhead.
>
> |Method|Training|# Parameters|iid JS|sim JS|iid PMF|sim PMF|
> |---|---|---|---|---|---|---|
> |Diffusion|**50 min**|650k|**0.0081 ± 0.0003**|0.0695 ± 0.0517|0.095 ± 0.003|1.047 ± 0.924|
> |Wide|67 min|2M|0.0082 ± 0.0003|0.0406 ± 0.0236|0.096 ± 0.003|0.467 ± 0.245|
> |Deep|111 min|2M|**0.0078 ± 0.0003**|0.0376 ± 0.0087|**0.091 ± 0.002**|0.478 ± 0.038|
> |MoE|**50 min**|2M|**0.0079 ± 0.0003**|**0.0264 ± 0.0085**|**0.093 ± 0.007**|**0.325 ± 0.113**|
>
> We will add this ablation and further clarify the benefits of MoE in the revised manuscript, including a pointer to this connection with time-splitting approaches for solving time-dependent PDEs.
>
> > Are equations 10-12 somehow related to the hutchinson trace estimator or the estimator used in [1]?
>
> Our treatment of the divergence term in the weak residual formulation is closely related to the Hutchinson's trace estimator used by Albergo and Vanden-Eijnden [1] for computing the PINN objective. Both approaches utilize Gaussian perturbations of $x$ to obtain unbiased estimates of the divergence. However, it is important to note a key difference in how the residual is defined. As described in App. A.2, in our weak residual $\tilde R(x,t)$, we apply the same Gaussian perturbation not only to the divergence term but also to other terms in the residual (see Eq. (30)). This means that when the strong residual $R(x,t)$ is exactly zero, the weak residual $\tilde R(x,t) = \mathbb{E}[R(x+v,t)]$ also remains exactly zero, regardless of the perturbation variance $\sigma^2$, avoiding truncation errors. This distinction allows our weak residual formulation to maintain consistency even under large perturbations. We will clarify this point further in the revised manuscript and cite the work [1] appropriately.
>
> -------
>
> [A] Kim et al. (2022) Soft truncation: A universal training technique of score-based diffusion model for high precision score estimation
>
> [B] Koehler et al. (2023) Statistical efficiency of score matching: The view from isoperimetry
>
> [C] Arts et al. (2023) Two for one: Diffusion models and force fields for coarse-grained molecular dynamics
>
> [D] Song et al. (2019) Generative modeling by estimating gradients of the data distribution
>
> [E] Wight et al. (2020) Solving Allen-Cahn and Cahn-Hilliard equations using the adaptive physics informed neural networks

---

> > ### Comment · Reviewer_c2SZ · 2025-08-05
> > **Reviewer Response**
> >
> > Thanks to the authors for a detailed response and additional experiments! I would recommend the authors to provide a clearer description of the problem setup earlier on - i.e. where ground-truth forces are not known but data is provided. In particular, can the authors highlight a practical use-case where one may want to perform simulation using the inferred scores at $t=0$ as opposed to just doing inference on the diffusion model from noise to data? Both approaches aim to provide samples from the target distribution where the latter has been shown to be really good, so what is the use case for the former?

---

> > > ### Author Response · Authors · 2025-08-05
> > >
> > > We thank the reviewer for their continued engagement and feedback and agree that the practical motivation should be stated more clearly earlier in the paper. In the following, we will discuss applications where simulation is needed instead of independent samples.
> > >
> > > Classical diffusion sampling provides iid samples from the equilibrium distribution but lacks temporal structure, making it unsuitable for estimating **kinetic properties** [F, G]. In contrast, simulation using learned forces (sim) produces continuous trajectories, which are essential for such analyses.
> > >
> > > Temporal information is particularly important when studying **transition mechanism**. For instance, to understand how a drug binds to a protein [H] or how a protein folds [I], we need simulation to capture the time evolution, which cannot be inferred from static samples alone. Entire fields in biochemistry, such as transition path sampling [J] and minimum energy path estimation [K], rely on simulations to study dynamic behavior.
> > >
> > > Simulation also enables **targeted exploration** from specific initial states, including rare or experimentally inaccessible configurations. Rather than inefficiently sampling iid until a desired state reoccurs, simulation can explore the system’s evolution from that state directly to uncover long-timescale behavior [L].
> > >
> > > Finally, the ability to estimate forces and the potential from the score opens up opportunities to use well-established **enhanced sampling** methods such as metadynamics [M] and umbrella sampling [N], which require a dynamical framework.
> > >
> > > These considerations are especially relevant in coarse-grained systems that we study, where fine-grained structural details, energies, and forces are unavailable. This scenario is common in simulations involving solvents, which have very high dimensionality and are computationally expensive. Our model can implicitly learn solvent effects, significantly reducing dimensionality and computational cost. Typically, machine-learned force fields in coarse-grained modeling are developed primarily for simulation [O,P], and rarely support both simultaneously [C]. In contrast, our approach uniquely complements simulation with independent sampling within a single model.
> > >
> > >
> > > We thank the reviewer for this feedback and will incorporate these clarifications and examples into the introduction to better motivate the relevance of simulation beyond iid sampling. Additionally, we will also add a dynamic analysis for one of the proteins to show the time evolution of the system, which should further illustrate the benefits of simulation.
> > >
> > >
> > > ------
> > >
> > > [F] Wang, J., & Hou, T. (2011). Application of molecular dynamics simulations in molecular property prediction. 1. density and heat of vaporization
> > >
> > > [G] Wang, J., & Hou, T. (2011). Application of molecular dynamics simulations in molecular property prediction II: diffusion coefficient
> > >
> > > [H] Buch, I., Giorgino, T., & De Fabritiis, G. (2011). Complete reconstruction of an enzyme-inhibitor binding process by molecular dynamics simulations
> > >
> > > [I] Englander, S. W., & Mayne, L. (2014). The nature of protein folding pathways
> > >
> > > [J] Throwing ropes over rough mountain passes, in the dark
> > >
> > > [K] Henkelman, G., Uberuaga, B. P., & Jónsson, H. (2000). A climbing image nudged elastic band method for finding saddle points and minimum energy paths
> > >
> > > [L] Chong, L. T., Saglam, A. S., & Zuckerman, D. M. (2017). Path-sampling strategies for simulating rare events in biomolecular systems
> > >
> > > [M] Laio, A., & Parrinello, M. (2002). Escaping free-energy minima
> > >
> > > [N] Torrie, G. M., & Valleau, J. P. (1977). Nonphysical sampling distributions in Monte Carlo free-energy estimation: Umbrella sampling
> > >
> > > [O] Wang, J., Olsson, S., Wehmeyer, C., Pérez, A., Charron, N. E., De Fabritiis, G., ... & Clementi, C. (2019). Machine learning of coarse-grained molecular dynamics force fields
> > >
> > > [P] Charron, N. E., Bonneau, K., Pasos-Trejo, A. S., Guljas, A., Chen, Y., Musil, F., ... & Clementi, C. (2025). Navigating protein landscapes with a machine-learned transferable coarse-grained model

---

> > > > ### Comment · Reviewer_c2SZ · 2025-08-05
> > > > **Reviewer Response**
> > > >
> > > > Thanks to the authors for a clear illustration of the benefits of simulation and for clarifying the use-case of the proposed method towards problems beyond *iid sampling*. Are there experiments that the authors conduct in simulation that evaluate the different approaches in such **temporal** settings? In particular, Figures 3 and 5 seem to tackle the problem of iid sampling and simulation without any temporal structure (please correct me if I am wrong here).

---

> > > > > ### Author Response · Authors · 2025-08-06
> > > > >
> > > > > We thank the reviewer for their thoughtful followup question and for recognizing the importance of simulation-based use cases. Our primary goal in this work is to achieve consistency between the diffusion model sampling probability $p(x)$ and an energy $U(x)$ such that $p \propto \exp(-U(x))$, i.e., that the diffusion model is a proper energy-based model. Once we have this equivalence, all enhanced sampling, rare event sampling methods that have been studied in the last seven decades become usable from diffusion model samplers.
> > > > >
> > > > > Figure 3 and 5 precisely evaluate this consistency in a simulation setting, which requires an accurate energy function [Q]. The samples shown in Figure 3 and 5 (sim) are generated via long Langevin simulations (millions of sequential steps) using the learned energy and its gradients. These trajectories are temporally correlated and thus fundamentally different from iid samples. Even small inconsistencies in the score function can accumulate and degrade simulation quality, as can be observed in non-regularized models (e.g., Figure 14, “Diffusion (sim)”). The stability of our regularized models in these settings demonstrates improved temporal consistency.
> > > > >
> > > > > That said, we agree that explicitly evaluating temporal behavior can make the analysis clearer for readers interested in simulation dynamics. For Chignolin and BBA, we have conducted a dynamic analysis based on transition probabilities between metastable states (e.g., folded and unfolded states) of the underlying Markov model. This tells us how likely a transition to a different state (e.g., unfolding) is, which would not be possible with iid samples. The Jensen-Shannon (JS) divergence between these transition matrices is reported in the tables below, highlighting the benefits of our approach:
> > > > >
> > > > >
> > > > > **Chignolin**
> > > > > | Method | Transition Probability JS |
> > > > > |---|---|
> > > > > | Diffusion | $3.4 \cdot 10^{-2}$ |
> > > > > | Two For One | $2.3 \cdot 10^{-3}$ |
> > > > > | Mixture | $3.0 \cdot 10^{-2}$ |
> > > > > | Fokker-Planck | $4.6 \cdot 10^{-4}$ |
> > > > > | Both | $\mathbf{2.1 \cdot 10^{-4}}$ |
> > > > >
> > > > > **BBA**
> > > > > | Method | Transition Probability JS |
> > > > > |---|---|
> > > > > | Diffusion | $6.8 \cdot 10^{-3}$ |
> > > > > | Two For One | $7.1\cdot 10^{-3}$ |
> > > > > | Fokker-Planck | $\mathbf{1.4 \cdot 10^{-3}}$ |
> > > > >
> > > > >
> > > > > We will additionally include a figure visualizing the time evolution in the TIC space, showing how folding and unfolding transitions occur throughout the simulations, including renderings of the proteins. We thank the reviewer for this suggestion and believe that these additional results further strengthen the claims of the paper.
> > > > >
> > > > > --------
> > > > >
> > > > > [Q] Trendelkamp-Schroer, B., & Noé, F. (2016). Efficient estimation of rare-event kinetics

---

> > > > > > ### Comment · Reviewer_c2SZ · 2025-08-06
> > > > > > **Reviewer Response**
> > > > > >
> > > > > > Thanks to the authors for prompt and elaborate response and for the additional experiments. I am happy with the discussion and the authors have resolved all my concerns. I am happy to recommend acceptance of the paper.

---

> > > > > > > ### Author Response · Authors · 2025-08-08
> > > > > > >
> > > > > > > We would like to once again thank the reviewer for their constructive feedback and the productive rebuttal. We are glad, that we could address all their concerns about the need for simulation and training without forces, and we will make this clearer in a revised version of the manuscript. We believe that the additional temporal analysis significantly improves the paper and thank the reviewer for this suggestion.

---

### Official Review · Reviewer_GH1q · 2025-07-03

**Clarity:** 4
**Significance:** 3
**Originality:** 3
**Rating:** 5
**Confidence:** 3

**Summary:**

This paper explores the connection between the score of diffusion models and the potential energy surface of a molecular system. The authors show that there is a mismatch between samples generated via reverse diffusion and Langevin dynamics using the score as a force field. To fix this, the authors augment the standard denoising score matching loss with a Fokker–Planck residual penalty that biases the learned energy/time evolution to satisfy the Fokker–Planck equation. They further reduce computational cost by partitioning diffusion time into segments and training a Mixture-of-Experts. They show empirically that they can train consistent Boltzmann Generators.

**Questions:**

1.	Could you provide a more detailed analysis of the non-smoothness of the resulting PES? (see above for more detail on the question) How smooth are geometry optimisations for example?
2.	Could you test a more diverse dataset, potentially of small molecules? With more diversity in the functional groups present. Does the FP regularisation truly transferrable learn more consistent scores? This is of interest as there is no guarantee of the score satisfying the FP equation from an architectural perspective.

**Ethical Concerns:**

["NO or VERY MINOR ethics concerns only"]

**Final Justification:**

The rebuttal phase addressed many of my questions. It is clear that the paper has the potential for high impact in at least one sub-area of AI. I particularly appreciate the more direct comparisons to standard force-matching approaches. I would encourage the authors to further benchmark their simulation model as a force field, particularly by examining the smoothness of the potential energy surface. Their response that such investigations will be pursued in future work is reasonable.

**Limitations:**

-	Overall adequately addressed, however the problems associated with a non-smooth PES do not seem adequately investigated.

**Paper Formatting Concerns:**

- the hyper-ref seems to be reformatted.

**Quality:**

3

**Strengths And Weaknesses:**

-	The clarity of the paper is very strong as is the overall quality of the text.
-	Figure 2 nicely illustrates the need for the added FP regularisation.
-	The results are convincing and the free energy plots informative.
-	The transferability of the approach could be probed more. Is it transferrable across diverse molecular datasets? Rather than just similar dipeptides?
-	The authors mention that the MoE approach is more expensive, but a detailed cost comparison of multiple inferences is not provided.
-	The use of only a single MoE and the associated non smoothness of the potential energy surface is not thoroughly investigated and appears to me as a significant limitation of the approach. The artifacts may appear in kinetic observables and may produce artifacts in other relevant observables not investigated in this approach. Could the authors quantitatively investigate the non-smoothness of the potential energy surface, by probing the forces as the model changes between mixture of experts? How significant can these differences be?
-	How does the proposed method compare to recently proposed long-stride MD simulations such as FlashMD, BoostMD, TrajCast, Timewarp?

---

> ### Author Rebuttal · Authors · 2025-07-30
>
> We thank the reviewer for their insightful review and questions. We appreciate their judgment that our paper is a significant contribution to this conference. We now address their questions and concerns individually.
>
> > The authors mention that the MoE approach is more expensive, but a detailed cost comparison of multiple inferences is not provided.
>
> The MoE approach introduces essentially *no* computational overhead during training or inference, as only a *single* model is active at any given diffusion time $t$. In fact, Appendix C.3 shows that MoE can reduce training and significantly reduce inference time by up to 50% in our experiments. These performance gains are achieved by assigning simpler, smaller models to larger diffusion times with a lot of noise, where high precision is not required.
>
> This design differs from conventional MoE approaches that combine multiple experts per input or rely on overfitting specialized models. Our MoE is a time-partitioned scheme, optimized for computational efficiency, and ensures consistency across different $x$ at the same diffusion time.
>
> We will make this clearer in the next revision.
>
> > The use of only a single MoE and the associated non smoothness of the potential energy surface is not thoroughly investigated and appears to me as a significant limitation of the approach. The artifacts may appear in kinetic observables and may produce artifacts in other relevant observables not investigated in this approach. Could the authors quantitatively investigate the non-smoothness of the potential energy surface, by probing the forces as the model changes between mixture of experts? How significant can these differences be?
>
> > Could you provide a more detailed analysis of the non-smoothness of the resulting PES? (see above for more detail on the question) How smooth are geometry optimisations for example?
>
> We thank the reviewer for raising this important point. As noted above, for each diffusion time $t$, only a single model is trained and used. This means that at diffusion time $t=0$, which is required for simulation, only a *single* independent expert model is used. This model is essentially a small-time expert and was trained on the interval $(0, 0.1)$. Since the entire simulation trajectory remains within this interval, only one model is used throughout, and thus no model switching occurs during simulation or training. As a result, the potential energy surface (PES) and corresponding forces remain smooth within this regime, and the MoE architecture does not introduce discontinuities that could affect kinetic observables.
>
> Discontinuities in the score can occur during iid sampling, where samples are generated across the full diffusion timeline by stitching together different experts. In this case, the score is not guaranteed to be continuous at the expert boundaries. We experimented with smoothing transitions between models by interpolating scores or jointly training experts with overlapping time intervals, but observed no qualitative improvements in sample quality or training stability. Thus, we opted for the simpler approach of independently training each expert.
>
> We acknowledge that this distinction between simulation and sampling use cases may not have been made sufficiently clear in the current version of the manuscript and will revise the text accordingly.
>
> > How does the proposed method compare to recently proposed long-stride MD simulations such as FlashMD, BoostMD, TrajCast, Timewarp?
>
> We thank the reviewer for highlighting these related approaches. Methods like FlashMD, TrajCast, and TimeWarp all accelerate MD by directly predicting updated configurations at larger time steps. In contrast, our approach learns a (coarse-grained) force field, derived as the gradient of a learned energy function, and then uses that force field to drive simulations. By explicitly modeling the energy, our method can be combined with enhanced-sampling techniques such as umbrella sampling and metadynamics.
>
> BoostMD proposes a framework to speed-up simulations by evaluating ML force fields only every N simulations steps, and evaluating a cheaper surrogate model for the intervening updates. By contrast, our method requires no force-field information during training, only configuration data. In principle, this allows us to drive simulations directly with our learned model, obviating any explicit ML force field. Whether such purely data-driven trajectories achieve sufficient accuracy in these domains remains to be seen.
>
> Direct comparisons are challenging, since each method has been validated on different systems.
>
> We thank the reviewer for the suggestions and will add these in the related work section.
>
> > The transferability of the approach could be probed more. Is it transferrable across diverse molecular datasets? Rather than just similar dipeptides?
>
> > Could you test a more diverse dataset, potentially of small molecules? With more diversity in the functional groups present. Does the FP regularisation truly transferrable learn more consistent scores? This is of interest as there is no guarantee of the score satisfying the FP equation from an architectural perspective.
>
> We thank the reviewer for this suggestion. While evaluating broader chemical diversity, such as small molecules with diverse functional groups, is indeed of interest, the computational cost of extending to such a dataset is significant and unfortunately beyond the scope of what we can address during the rebuttal period.
>
> That said, we agree that demonstrating scalability beyond the dipeptide dataset is important. To this end, we extended our evaluation to two larger and more complex systems: **Chignolin** (10 amino acids) and **BBA** (28 amino acids), both of which are well-studied fast-folding proteins. These systems go significantly beyond the small coarse-grained dipeptides and serve as a step toward testing generalization to more diverse and larger molecular systems.
>
> **Chignolin Results:**
> |Method|iid JS|sim JS|iid PMF|sim PMF|
> |---|---|---|---|---|
> |Diffusion|**0.0036 ± 0.0001**|0.4351 ± 0.0141|**0.027 ± 0.000**|63.804 ± 0.372|
> |Two For One|**0.0036 ± 0.0001**|0.1023 ± 0.0008|**0.027 ± 0.000**|1.438 ± 0.019|
> |Mixture|0.0042 ± 0.0001|0.4336 ± 0.0075|0.033 ± 0.000|11.185 ± 0.430|
> |Fokker-Planck|0.0048 ± 0.0001|**0.0050 ± 0.0001**|0.037 ± 0.000|0.040 ± 0.001|
> |Both|0.0045 ± 0.0001|**0.0050 ± 0.0008**|0.035 ± 0.001|**0.038 ± 0.006**|
>
> **BBA Results:**
> |Method|iid JS|sim JS|iid PMF|sim PMF|
> |---|---|---|---|---|
> |Diffusion|**0.0043 ± 0.0000**|0.2014 ± 0.0043|**0.034 ± 0.000**|5.387 ± 0.144|
> |Two For One|**0.0043 ± 0.0000**|0.1162 ± 0.0021|**0.034 ± 0.000**|1.624 ± 0.107|
> |Fokker-Planck|0.0070 ± 0.0000|**0.0427 ± 0.0053**|0.060 ± 0.000|**0.438 ± 0.074**|
>
> In both systems, we observe a substantial improvement in sampling-simulation consistency when applying Fokker-Planck regularization. These effects are also visible in the corresponding free energy surfaces: while the Two For One [1] model tends to overdisperse and fragment the distribution across multiple modes, our method better preserves the structure of the equilibrium landscape and recovers the correct metastable states. To our knowledge, our method is the first that can accurately sample and simulate the dynamics of BBA.
>
> These results support the conclusion that Fokker-Planck regularization improves sampling-simulation consistency even in substantially larger systems, without relying on architectural constraints. While these proteins are still relatively small compared to the full range of molecular diversity in chemistry, they demonstrate the scalability and consistency benefits of our approach in higher dimensions. We will include these additional results and dynamic analyses in the revised manuscript.
>
> > ### Limitations:
>
> > Overall adequately addressed, however the problems associated with a non-smooth PES do not seem adequately investigated.
>
> We thank the reviewer for pointing this out. As clarified in our response above, the Mixture-of-Experts (MoE) design does not introduce non-smoothness in the potential energy surface (PES) during simulation, since only a single expert trained over a narrow diffusion time interval is used throughout. This ensures that the PES remains smooth along simulation trajectories. We acknowledge that this was not made sufficiently clear in the original submission and will revise the manuscript to better explain this point.
>
> -------
>
> [1] Arts et al. (2023). Two for one: Diffusion models and force fields for coarse-grained molecular dynamics

---

> ### Comment · Reviewer_GH1q · 2025-08-07
>
> Thank you very much for the detailed response. However, I have some further questions concerning the response.
>
> My concerns about the PES smoothness have been thoroughly addressed. Some more information such as dimer curves or other typical force field metrics/PES slices investigating smoothness would be interesting. General benchmarks typically applied to MLFFs would be interesting. Can you converge transition state searches? How good are torsion scans?
>
> The authors state that it is possible to run umbrella sampling and other enhanced sampling approaches. Could the authors explicitly show these results and hence improve the convergence of the free energy plots of Figure 2-3?
>
> Overall there seems to be very little comparison to existing corse graining approaches. Am I missing direct comparisons to force-matching approaches? What is the speedup over normal molecular dynamics compared to other approaches eg [1] or other cited in the MS? Is this mainly a theoretically interesting observation or beating existing SOTA methods?
>
> [1] Machine Learning of coarse-grained Molecular Dynamics Force Fields, Wang et al. 2018

---

> > ### Author Response · Authors · 2025-08-08
> > **Official Comment by Authors - Part 1/2**
> >
> > We thank the reviewer for their follow-up and are glad that our previous response has addressed their concerns regarding PES smoothness. Below, we provide additional results to further address the reviewer’s suggestions. Given the limited time before the discussion period ends (<48h), we could only run a subset of the additionally suggested experiments.
> >
> > > My concerns about the PES smoothness have been thoroughly addressed. Some more information such as dimer curves or other typical force field metrics/PES slices investigating smoothness would be interesting. General benchmarks typically applied to MLFFs would be interesting. Can you converge transition state searches? How good are torsion scans?
> >
> > Our main contribution of the paper is improving the consistency of the score, which we evaluate primarily through sequential MD simulations. Our model is not only a force field, but combines sampling and simulation capabilities, and we try to reflect both in our evaluation. We believe that simulating millions of sequential steps is a suitable test, as small force errors can accumulate over time, making high accuracy essential. Moreover, our energy-based parameterization guarantees smooth energies, since the model takes explicit gradients. This means that most metrics that we derive will be smooth as well. While benchmarks on small subsystems (e.g., dimers) can be informative, we focus on more representative metrics for biomolecular systems such as proteins.
> >
> > We agree that tracking the evolution of observables over time is important. To address this, we report additional results on the pairwise distances (PWD) between atoms. Below, we report the Jenson Shennon (JS) divergence between all pairwise distances. In the revised version of the manuscript, we will also include visualizations of the PWDs in the form of histograms.
> >
> > **Comparing PWD of Chignolin**
> > |Method|iid PWD JS|sim PWD JS|
> > |---|---|---|
> > |Diffusion|**0.0001 ± 0.0000**|0.3817 ± 0.0009|
> > |Two For One|**0.0001 ± 0.0000**|0.0082 ± 0.0000|
> > |Mixture|0.0003 ± 0.0000|0.2045 ± 0.0004|
> > |Fokker-Planck|0.0004 ± 0.0000|**0.0008 ± 0.0000**|
> > |Both|0.0003 ± 0.0000|**0.0012 ± 0.0005**|
> >
> > **Comparing PWD of BBA**
> > |Method|iid PWD JS|sim PWD JS|
> > |---|---|---|
> > |Diffusion|**0.0006 ± 0.0000**|0.1003 ± 0.0050|
> > |Two For One|**0.0006 ± 0.0000**|0.0240 ± 0.0008|
> > |Fokker-Planck|0.0011 ± 0.0000|**0.0104 ± 0.0010**|
> >
> > To further improve the evaluation of the simulation aspect, we defined meta-stable states, following [B], of Chignolin and BBA and estimated the transition probabilities of the underlying Markov chain (see table below). This analysis relies on sequential simulations and cannot be performed with independent samples, further highlighting the need for simulation. For evaluation, we report the JS divergence between the observed transition probabilities.
> >
> > **Transition Probability Analysis for Chignolin**
> > | Method | Transition Probability JS |
> > |---|---|
> > | Diffusion | $3.4 \cdot 10^{-2}$ |
> > | Two For One | $2.3 \cdot 10^{-3}$ |
> > | Mixture | $3.0 \cdot 10^{-2}$ |
> > | Fokker-Planck | $4.6 \cdot 10^{-4}$ |
> > | Both | $\mathbf{2.1 \cdot 10^{-4}}$ |
> >
> > **Transition Probability Analysis for BBA**
> > | Method | Transition Probability JS |
> > |---|---|
> > | Diffusion | $6.8 \cdot 10^{-3}$ |
> > | Two For One | $7.1\cdot 10^{-3}$ |
> > | Fokker-Planck | $\mathbf{1.4 \cdot 10^{-3}}$ |
> >
> > We will add these tables to the manuscript and include a figure illustrating conformational changes over time.
> >
> > Regarding torsion scans, these are implicitly covered by our reported dihedral angle distributions for alanine dipeptide and the transferable dipeptides setting (Figure 3 and 5). The PMF metric in Tables 1 and 2 directly quantifies differences in torsional behavior between reference simulations and our model. We will clarify in the text how the PMF is computed to make this connection explicit.
> >
> > > The authors state that it is possible to run umbrella sampling and other enhanced sampling approaches. Could the authors explicitly show these results and hence improve the convergence of the free energy plots of Figure 2-3?
> >
> > Our main claim is the improvement of sampling-simulation consistency, which we primarily assess by comparing free energy surfaces from long MD simulations with those from iid sampling. We see the exploration of enhanced sampling techniques, including umbrella sampling and OPES [A], as promising future work and beyond the scope of our current work.
> >
> > We have, however, prepared preliminary umbrella sampling experiments for Chignolin. While the results are less accurate than those from the long MD runs reported in the paper, they recover a qualitatively similar free energy surface (along TIC0) using less than 1% of the simulation data required by our other experiments.
> >
> > We thank the reviewer for the suggestion and will include more thorough umbrella sampling experiments and a discussion of these findings in the final manuscript.
> >
> > [A] Invernizzi, M. (2021). OPES: On-the-fly probability enhanced sampling method

---

> > > ### Author Response · Authors · 2025-08-08
> > > **Official Comment by Authors - Part 2/2**
> > >
> > > > Overall there seems to be very little comparison to existing corse graining approaches. Am I missing direct comparisons to force-matching approaches?
> > >
> > > We thank the reviewer for this suggestion. We compare our method against the state-of-the-art diffusion simulation method Two For One [B], which significantly improves upon prior work [C]. Notably, these methods, as well as ours, do not require force label for the training data, which is not always available. However, Two For One [B] also performs better than force-matching methods that require explicit force labels, such as the work suggested by the reviewer [1], hence we only included Two For One in benchmarks.
> > > However, we agree that a comparison against more methods strengths the contributions of this paper. We have prepared a comparison against very recent force-matching methods that do not require force labels, but derive them via a noising process or flow models from data [D,E]. In addition, we also compare with the subsequent work of [1], namely CGSchNet [F], which was also evaluated in [E] and relies on explicit force labels for training the ML model. Notably, our approach does not rely on physical priors to stabilize simulation, in contrast to all these force-matching methods (D-F). Despite this fact, we can observe in the table below, that our method is capable to accurately learn the forces of this system, beating all other methods.
> > >
> > > **Comparison for alanine dipeptide with other force matching approaches**
> > > | Method | sim JS | sim PMF |
> > > |---|---|---|
> > > | Force Matching [F] | 0.0243 ± 0.0015 | 0.322 ± 0.005 |
> > > | Noise Forces [D] | 0.0402 ± 0.0066 | 0.864 ± 0.146 |
> > > | Operator Forces * [E] | 0.0214 ± 0.0041 | 0.301 ± 0.043 |
> > > | Operator Forces * [E] | 0.0166 ± 0.0009 | 0.282 ± 0.031 |
> > > | Ours (Fokker Planck) | **0.0088 ± 0.0006** | **0.105 ± 0.011** |
> > > | Ours (Both) | **0.0086 ± 0.0004** | **0.099 ± 0.003** |
> > >
> > > *Using two different flow models to generate the training forces
> > >
> > > > What is the speedup over normal molecular dynamics compared to other approaches eg [1] or other cited in the MS?
> > >
> > > The main computational advantage of our approach comes from the ability to run simulations in parallel and to employ coarse-graining. In conventional molecular dynamics, simulating a protein typically requires also modeling the solvent, which can involve thousands of additional water atoms. Coarse-graining removes these degrees of freedom, reducing computational cost and improving parallelizability.
> > >
> > > For example, in the dipeptide dataset, which contains only a few atoms and no solvent to coarse-grain, our method achieves 150 parallel simulations at 830 steps/s each, totaling roughly 125k steps/s on a single NVIDIA A100. In comparison, a classical force-field simulation on an NVIDIA V40 reaches about 10k steps/s. We expect larger gains for systems with explicit solvent, where coarse-graining removes many more particles. For instance, our BBA protein simulation (100 parallel runs of 1M steps) completed in about one hour.
> > >
> > >
> > > We will extend the manuscript with a more detailed comparison and include these performance results in the appendix.
> > >
> > >
> > > > Is this mainly a theoretically interesting observation or beating existing SOTA methods?
> > >
> > > Our method establishes a theoretically motivated connection between the Fokker-Planck equation and the consistency of diffusion models, and we demonstrate that this connection leads to strong empirical gains. Across all tested systems, ranging from toy potentials to transferable biomolecular models, our approach consistently outperforms prior methods, often by a substantial margin. While we stop short of claiming definitive state-of-the-art across all possible molecular systems, our results show clear improvements over existing baselines in both sampling and simulation quality. Extending this evaluation to larger and more complex systems would be a natural next step and could enable the use of large-scale models for accurate simulation as well.
> > >
> > > We thank the reviewer for the constructive feedback and believe that the additional evaluations and experiments, in particular the comparison to force-matching approaches, will strengthen the main claims and improve the manuscript.
> > >
> > >
> > >
> > > [B] Arts et al. (2023). Two for one: Diffusion models and force fields for coarse-grained molecular dynamics
> > >
> > > [C] Kohler, J., Chen, Y., Kramer, A., Clementi, C., & Noé, F. (2023). Flow-matching: Efficient coarse-graining of molecular dynamics without forces
> > >
> > > [D] Durumeric, A. E., Chen, Y., Noé, F., & Clementi, C. (2024). Learning data efficient coarse-grained molecular dynamics from forces and noise
> > >
> > > [E] Klein, L., Kelkar, A., Durumeric, A., Chen, Y., & Noé, F. (2025). Operator Forces For Coarse-Grained Molecular Dynamics
> > >
> > > [F] B. E. Husic, N. E. Charron, D. Lemm, J. Wang, A. Pérez, M. Majewski, A. Krämer, Y. Chen, S. Olsson, G. De Fabritiis, et al. (2020). Coarse graining molecular dynamics with graph neural networks

---

> ### Comment · Reviewer_GH1q · 2025-08-08
>
> Thank you for the additional experiments. They reflect substantial effort to strengthen the empirical validation, and I appreciate the expanded comparison to force-matching approaches.
>
> To further substantiate the claims of transferability and accuracy, I recommend evaluating the learned PES on standard MLFF benchmarks. For example, torsion scans across a chemically diverse set such as TorsionNet-500 rather than focusing solely on dipeptides. This would enable clearer comparison to state-of-the-art MLFF methods and better contextualize the results.
>
> Overall, the new experiments have strengthened my assessment of the paper’s value.

---

> > ### Author Response · Authors · 2025-08-09
> >
> > We thank the reviewer for their time and effort they have put into the review and the discussion period. We agree that evaluation and training on chemically more diverse datasets, such as the TorsionNet-500 are a promising research direction and we will note it as future work. We are glad that the additional experiments and extended comparisons addressed the reviewer’s concerns, and we believe that, in particular, the broader set of baselines strengthens the claims of the paper.

---

### Note · Authors · 2025-08-12

We would like to once more thank all reviewers for their careful evaluations and constructive feedback and we are glad that we could address their concerns.

Since several reviewers raised questions about the motivation behind our work and the relevance of simulation when iid sampling is possible, we have substantially revised the introduction and added extensive literature references to better contextualize our approach. In particular, we emphasize that beyond generating high-quality samples, many scientific applications critically rely on accurate energy and score function estimates. Our work directly addresses this need by improving the consistency between sampling and simulation, yielding a model that can do both. We believe these clarifications make our goals clearer and the paper more accessible to a wider audience and easier to understand.

Following reviewer suggestions, we have added results on larger systems and proteins (Chignolin and BBA), demonstrating the scalability of our method. Notably, for the protein BBA we can now learn forces consistent with sampling, significantly improving accuracy over previous work. We also include an analysis of the folding and unfolding dynamics of these proteins, further illustrating the method’s capabilities.

Finally, we have included additional benchmarks comparing against different force-matching approaches.

Taken together, we believe that these and the other changes and additions made during the rebuttal significantly strengthen our work.

---

### Decision · Program_Chairs · 2025-09-17

**Decision:**

Accept (poster)

**Comment:**

The work introduces a Fokker–Planck-based regularization to the standard denoising score matching loss of a diffusion model. This improves the alignment between sampling from Boltzmann distributions using the score of diffusion and Langevin dynamics for molecular dynamics simulations.

The reviews have generally found the work potentially highly impactful. Particularly after the rebuttal, where the authors presented a broader experimental evaluation and stronger motivation for the approach, the consensus is that this paper provides a valuable contribution to the field.